# Integration of Airborne and Ground Observations of Nitryl Chloride in the Seoul Metropolitan Area and the Implications on Regional Oxidation Capacity During KORUS-AQ 2016

Daun Jeong[1], Roger Seco[1], Dasa Gu[1,a], Youngro Lee[2], Benjamin A. Nault[3,4], Christoph J. Knote[5], Tom Mcgee[6], John T. Sullivan[6], Jose L. Jimenez[3,4], Pedro Campuzano-Jost[3,4], Donald R. Blake[1], Dianne Sanchez[1], Alex B. Guenther[1], David Tanner[2], L. Gregory Huey[2], Russell Long[7], Bruce E. Anderson[8], Samuel R. Hall[9], Kirk Ullmann[9], Hye-jung Shin[10], Scott C. Herndon[11], YoungJae Lee[10], Danbi Kim[10], Joonyoung Ahn[10], and Saewung Kim[1]

[1]Department of Earth System Science, University of California, Irvine, Irvine, CA, USA
[2]School of Earth and Atmospheric Sciences, Georgia Institute of Technology, Atlanta, GA, USA
[3]Cooperative Institute for Research in Environmental Sciences, University of Colorado, Boulder, CO, USA
[4]Department of Chemistry, University of Colorado, Boulder, CO, USA
[5]Meteorologisches Institut, Ludwig-Maximilians-Universität München, München, Germany
[6]Atmospheric Chemistry and Dynamics Laboratory, NASA GSFC, Greenbelt, MD, USA
[7]Office of Research and Development, U.S. EPA, Research Triangle Park, NC, USA
[8]NASA Langley Research Center, Hampton, VA, USA
[9]National Center for Atmospheric Research, Boulder, CO, USA
[10]National Institute of Environmental Research, Incheon, South Korea
[11]Aerodyne Research Inc., Billerica, MA, United States
[a]currently at: Division of Environment and Sustainability, Hong Kong University of Science and Technology, Hong Kong, China

**Correspondence:** Saewung Kim (saewung.kim@uci.edu)

**Abstract.** Nitryl chloride (ClNO$_2$) is a radical reservoir species that releases chlorine radicals upon photolysis. An integrated analysis of the impact of ClNO$_2$ on regional photochemistry in the Seoul Metropolitan Area (SMA) during the Korean-United States-Air Quality (KORUS-AQ) 2016 field campaign is presented. Comprehensive multiplatform observations were conducted aboard the NASA DC-8 and at two ground sites (Olympic Park, OP; Taehwa Research Forest, TRF), representing an urbanized area and a forested suburban region, respectively. Positive correlations between daytime Cl$_2$ and ClNO$_2$ were observed at both sites, the slope of which were dependent on O$_3$ levels. The possible mechanisms are explored through box model simulations constrained with observations. The overall diurnal variations of ClNO$_2$ at both sites appeared similar but the night time variations were systematically different. For about half of the observation days at the OP site the level of ClNO$_2$ increased at sunset but rapidly decreased at around midnight. On the other hand, high levels were observed throughout the night at the TRF site. Significant levels of ClNO$_2$ were observed at both sites for 4-5 hours after sunrise. Airborne observations, box model calculations, and back trajectory analysis consistently show that these high levels of ClNO$_2$ in the morning are likely from vertical or horizontal transport of air masses from the west. Box model results show that chlorine radical initiated chemistry can impact the regional photochemistry by elevating net chemical production rates of ozone by $\sim 25$ % in the morning.

## 1 Introduction

Nitryl chloride (ClNO$_2$) is a night time radical reservoir that generates chlorine radicals (Cl$\cdot$) upon sunrise (R1), with a lifetime ($\tau_{ClNO2}$) of around 30 minutes at midday in the northern hemisphere mid-latitude summer, under clear sky conditions (J$_{ClNO2}$ $\approx 5.47 \times 10^{-4}$ s$^{-1}$,(Madronich and Flocke, 1998)). It is produced through heterogeneous reaction of chloride (Cl$^-$) containing aerosols and dinitrogen pentoxide (N$_2$O$_5$(g)), which is generated from an equilibrium reaction with gas-phase nitrate radical (NO$_3^\cdot$) and nitrogen dioxide (NO$_2$) (R 2-4, (Finlayson-Pitts et al., 1989)). In acidic aerosols ($\approx$ pH 1.8), uptake of N$_2$O$_5$ (g) can also produce gas-phase chlorine (Cl$_2$, R5), resulting from enhanced ClNO$_2$ uptake coefficient of up to 3 - 4 orders of magnitude higher than neutral pH (Roberts et al., 2008). However, this reaction has yet to be proven in ambient conditions.

During the day, N$_2$O$_5$ exists at low levels due to its thermal instability (Malko and Troe, 1982) and the short lifetime of NO$_3$ ($\tau_{NO3} < 5$ s) from photolysis and reaction with NO (Wayne et al., 1991). Particulate Cl$^-$ and chlorine containing gas species can come from both natural sources such as sea salt and biomass burning (Blanchard, 1985; Woodcock, 1953), and anthropogenic sources such as steel making, incineration, bleaching processes, and coal-fired power plants (Hov, 1985; Reff et al., 2009; Tanaka et al., 2000; Lee et al., 2018; Fu et al., 2018). The efficiency of ClNO$_2$ production depends on heterogeneous loss of N$_2$O$_5$, which is a function of the N$_2$O$_5$ aerosol uptake coefficient ($\gamma_{N2O5}$), aerosol surface area, and N$_2$O$_5$ mean molecular speed, as well as the yield of ClNO$_2$ ($\phi_{ClNO2}$) (e.g., Thornton et al. 2003; Schweitzer et al. 1998; Behnke et al. 1997; Hu and Abbatt 1997; Bertram and Thornton 2009). Many recent studies have reported discrepancies between field derived and laboratory parameterized $\gamma_{N2O5}$ (e.g., Brown et al. 2009; Chang et al. 2016; Morgan et al. 2015; Phillips et al. 2016; McDuffie et al. 2018b; Tham et al. 2016; Wang et al. 2017a, b, c) and $\phi_{ClNO2}$ (e.g., McDuffie et al. 2018a; Riedel et al. 2013; Ryder et al. 2015; Tham et al. 2018; Thornton et al. 2010; Wagner et al. 2013; Wang et al. 2017b, c). In a nocturnal boundary layer, ClNO$_2$ can accumulate to significant levels due to its long lifetime ($\tau_{ClNO2} > 30$ h) with slow

loss mechanisms through heterogeneous uptake (Behnke et al., 1997; Frenzel et al., 1998; George et al., 1995). At sunrise, $ClNO_2$ rapidly photolyzes to generate chlorine radicals ($Cl^.$), which can react with most volatile organic compounds (VOCs). For alkanes, $Cl^.$ has up to 1 - 2 orders of magnitude larger rate constants than hydroxyl radicals (e.g., $k_{OH+n-C4H10} = 2.4 \times 10^{-12}$, $k_{Cl+n-C4H10} = 2.2 \times 10^{-10}$ at 298 K) (Atkinson, 1997; Atkinson and Arey, 2003). Therefore, $Cl^.$ can potentially influence the radical pool ($HO_x$-$RO_x$) and ozone ($O_3$) level, which can also affect the formation of secondary aerosols. This influence can be most prominent in the morning when concentrations of other oxidants are low (i.e., $NO_3^-$ and $^.OH$) (Finlayson-Pitts, 1993; Hov, 1985; Young et al., 2014).

$$ClNO_{2(g)} + hv \rightarrow Cl^._{(g)} + NO_{2(g)} \tag{R1}$$

$$NO_{2(g)} + O_{3(g)} \rightarrow NO^._{3(g)} + O_{2(g)} \tag{R2}$$

$$NO^._{3(g)} + NO_{2(g)} \rightleftharpoons N_2O_{5(g)} \tag{R3}$$

$$N_2O_{5(g)} \xrightarrow{\gamma(N_2O_5), Cl^-(aq)} (2-\phi) * HNO_{3(g)} + \phi * ClNO_{2(g)} \tag{R4}$$

$$ClNO_{2(g)} + Cl^-_{(aq)} + H^+_{(aq)} \rightarrow Cl_{2(g)} + HNO_{2(aq)} \tag{R5}$$

The first ambient measurements of $ClNO_2$ were carried out by Osthoff et al. (2008), from a ship sampling along the southeastern U.S. coast in 2006. In that study, $ClNO_2$ was observed up to $\sim$1 ppbv at night time, particularly during the time period influenced by urban pollution and ship plumes of the Houston ship channel. Since then, a growing number of measurements reported significant levels of $ClNO_2$, especially in polluted coastal regions with sources from natural and anthropogenic chloride and nitrogen oxides. Riedel et al. (2012) measured up to $\sim$2 ppbv of $ClNO_2$ off the coast of Santa Monica Bay, on board the research vessel Atlantis. Recent studies show that high levels of $ClNO_2$ are also present in mid-continental regions. Thornton et al. (2010) measured up to $\sim$ 400 pptv in Boulder, Colorado, which is $\sim$ 1,400 km away from the coastline. Mielke et al. (2011) reported up to $\sim$ 250 pptv in Calgary, Alberta, Canada, during spring, which is $\sim$ 800 km from the coastline. Back trajectory analysis results showed that the observations were most likely not influenced by marine air masses. More recently (in the past 5 years), increasing number of $ClNO_2$ observations have been conducted in Asia consistently showing significant levels of $ClNO_2$ present in the boundary layer (e.g., Tham et al. 2018, 2016; Wang et al. 2016, 2017c, 2014; Yun et al. 2018; Liu et al. 2017). $ClNO_2$ observations at semi-rural (Wangdu of Hebei province) and urban (Hong Kong, Jinan) regions in China have measured up to 2 ppbv and 776 pptv respectively (Tham et al., 2016; Wang et al., 2017b). At the mountain top (957 m above sea level) in Hong Kong, up to 4.7 ppbv of $ClNO_2$ was reported (Wang et al., 2016). The high levels of $ClNO_2$ in these studies were mostly correlated with continental pollution in vicinity (e.g., power plant plumes, biomass burning). A recent study by Yun et al. (2018) reported the highest-recorded mixing ratio of $ClNO_2$ (8.3 ppb), during a severe haze event in a semi-rural site downwind of the Pearl River Delta in the winter. Overall, observations have shown that $ClNO_2$ is ubiquitous in the tropospheric boundary layer.

However, measurements are still limited, as discrepancies remain between global chemical transport models and observations. Uncertainties in model simulated $ClNO_2$ can arise from limited emission inventories, low resolution of the grid,

uncertainties in $\gamma_{N2O5}$ and $\phi_{ClNO2}$ parameterization, complexity of the terrain, and meteorological conditions and these have been dealt in previous studies (e.g., Zhang et al. 2017; McDuffie et al. 2018b, a; Lowe et al. 2015; Sarwar et al. 2012, 2014; Sherwen et al. 2017). For instance, smoothing out local $ClNO_2$ peaks by diluting local $NO_x$ emissions, will result in limited $NO_3$ and $N_2O_5$ production. According to Sarwar et al. (2012, 2014), the Community Multiscale Air Quality (CMAQ) model with a finer grid (i.e. 12 km) simulated $ClNO_2$ that corresponded better to the observations, compared to the model runs with coarser grid size (i.e., 108 km), embedded with similar chemistry. Another modeling study by Sherwen et al. (2017) compared the $ClNO_2$ levels between the GEOS-Chem simulations and observations in inland areas (i.e., London, UK and a mountain top near Frankfurt, Germany) during the summer of 2015. Compared to observations, the simulations underestimated the $ClNO_2$ maxima levels by $\sim 7$ times in inland areas (Sherwen et al., 2017). Modeling studies have consistently suggested the significance of $Cl^-$ initiated reactions in regional and global $O_3$ production and in the lifetime of VOCs in the troposphere (Knipping and Dabdub, 2003; Tanaka et al., 2000, 2003; Sarwar et al., 2014; Sherwen et al., 2016; Simon et al., 2009). Sarwar et al. (2014) explored the production of $ClNO_2$ from sea salt and biomass burning and its impact in the Northern Hemisphere by including $ClNO_2$ formation chemistry in the CMAQ model. The results showed that, compared to the simulations without $ClNO_2$ formation, monthly 8 h wintertime maximum $O_3$ and $\cdot OH$ increased up to 15 % and 20 %, respectively. The impact was the largest in China and Western Europe. In the Hong Kong-Pearl River Delta (HK-PRD) region, Li et al. (2016) simulated up to $\sim 1$ ppbv of $ClNO_2$ originating from sea salt, biomass burning, and anthropogenic emissions (e.g., coal combustion) with the Weather Research and Forecasting coupled with Chemistry (WRF-CHEM) model. This resulted in $\sim 16$ % $O_3$ increase in the planetary boundary. Another modeling study of WRF-CHEM embedded with an updated chlorine chemistry, simulated 3-6 % of surface $O_3$ increase in the North China Plain and Yangtze River Delta during the summer (Zhang et al., 2017). A recent study by Wang et al. (2019) updated the standard version of the GEOS-Chem (Chen et al., 2017; Sherwen et al., 2016) to better track partitioning between aerosol chloride and gas-phase chlorine species. Comparison between their model simulations with and without $ClNO_2$ production showed enhanced $O_3$ up to 8 ppb during the winter season in Europe.

East Asian countries are of particular interest due to the rapid economic growth in the past decades with high anthropogenic emissions from densely populated megacities (e.g., Shanghai, Guangzhou, Beijing, Tokyo, Seoul). The majority of the world's megacities are situated in coastal regions (Neumann et al., 2015) with high $NO_x$ emissions and abundant sources of chloride from both anthropogenic and natural origin. These regional characteristics likely promote $ClNO_2$ production. Moreover, considering that nearly half the population in the world lives near the coast, defined as $< 100km$ from coastline (Hinrichsen, 1998), a careful evaluation of the impact of $ClNO_2$ on local tropospheric chemistry is crucial. In this study, we present $ClNO_2$ observation results from the Korean - United States Air Quality (KORUS-AQ) study conducted in the Seoul Metropolitan Area (SMA), South Korea during late Spring (May 2 to June 12, 2016). The field campaign was an international collaboration between the National Institute of Environmental Research (NIER) of South Korea and the National Aeronautics and Space Administration (NASA) of the United States with the aim to better understand the impact of a megacity on regional air quality. A comprehensive suite of measurements were deployed at two super sites (Olympic Park site, OP; Taewha Research Forest, TRF) and aboard the NASA DC-8 to make airborne observations over the South Korean peninsula and the Yellow Sea. We present observational and box model results to evaluate the impact of $ClNO_2$ towards regional air quality in SMA.

## 2 Methods

### 2.1 KORUS-AQ 2016 Field Campaign and Observation Sites

We present observations carried out at Olympic Park, (OP; lat:37° 30' 32.904" N, lon:127° 7' 20.136" E), Taehwa Research Forest (TRF; lat: 37° 19' 14.484" N, lon:127° 18' 32.58" E) and on the NASA DC-8. The two ground sites were within the

SMA region, which is the second largest metropolitan area in the world with a population of $\sim$ 24 million (Park et al., 2017). As shown in Figure 1(a), the OP site is located in the southern part of Seoul, surrounded by high rise residence buildings and close to major freeways. The TRF site is in the middle of a forested area, $\sim$ 26 km southeast of the OP site. Previous studies have shown that the TRF site is affected by both aged anthropogenic air masses from the city and fresh biogenic emissions from the forest (Kim et al., 2016, 2015). Both sites were $\sim$ 50 km to the east of the nearest coastline. Figure 1(b) shows the flight tracks

of the NASA DC-8, during the KORUS-AQ campaign. Spiral patterns were conducted near the TRF site to measure a vertical profile of the troposphere. Airborne observations were carried out during the daytime, between 8:00 and 17:00 local time. A summary of the analytical techniques of the measurements presented in this study are shown in Table 1. Meteorology during the observation period can be classified into dynamic (May 4th - 16th), stagnation (May 17th - 22nd), transport (May 25th - 31st), and blocking period. During the stagnant period, high pressure system was persistent in the Korean peninsula resulting

in local air masses to be more dominant within the SMA compared to the dynamic and transport (May 25nd - 31st) periods. Rex block patterns were observed during the blocking period (June 1st - 6th). During this period, a high pressure system was adjacent to a low pressure over the Korean peninsula resulting in more local influence with occasional stagnation.

### 2.2 Chemical Ionization Mass Spectrometry and Calibration

A THS Instruments LLC Chemical Ionization Mass Spectrometer (CIMS), using iodide ($I^-$) as the reagent ion was used for

measuring $Cl_2$ and $ClNO_2$ at the two ground supersites and on the NASA DC-8. The system was similar to what is described in Slusher et al. (2004) and Liao et al. (2011), and the inlet configuration during the campaign is shown in Figure S1. Ambient air was sampled through a stainless steel donut shaped inlet at TRF and a Polytetrafluoroethylene (PTFE) tube inlet at OP. The stainless steel donut inlet has been shown to effectively avoid wall loss of reactive halogens during previous campaigns (Liao et al., 2011). The lengths of the inlet lines of the three CIMS systems were 20 - 30 cm. The PTFE inlet line at the TRF site was

washed on a weekly basis and the ones at the OP and DC-8 were not washed routinely during the campaign due to difficulties on detaching the inlet. The potential bias of interactions of $Cl_2$ and $ClNO_2$ inside the inlet were not tested but the artifacts have been shown to be negligible in various field conditions (Riedel et al., 2012; Thornton et al., 2010; Liao et al., 2014). Therefore, the use of different types of inlets (e.g., the use of the donut), described above, at the two ground sites and on the DC-8 is not expected to be an issue for the quantitative comparisons in this study. The sampled air went through the first 3-way valves to be

delivered to an ambient or charcoal scrubber mode for background, alternating every 5 minutes. The second 3-way valve was for heated (150 $^o$C) and unheated cycles. $ClNO_2$ and $Cl_2$ were only quantified during the unheated cycles to avoid any potential artifacts as described in Liu et al. (2017). A total of 3000 standard liters per minute (slpm) was drawn in with a blower with an additional flow of 4 slpm drawn at the end of the inlet to reduce the residence time and 1 slpm was sampled into the CIMS. All

the inlet parts, after the blower, including the fittings and tubings, were made of PTFE. In the flow tube, the target compounds form clusters with $I^-$ (R 5-6, (Huey, 2007; Huey et al., 1995; McNeill et al., 2006)), which were generated by flowing 1 slpm $N_2$ through a methyliodide ($CH_3I$) permtube oven maintained at 50 $^oC$ . Polonium (NRD LLC, Static Master, Model: 2U500, Activity: 20 mCi) was used as the radioactive source for ionization. Clusters of $Cl_2$ isotopes were detected at the mass to charge ratio *(m/z)* of 197 and 199, and $ClNO_2$ was measured at 208 and 210. The natural abundance of $Cl_2$ and $ClNO_2$ isotopes are approximately 9:6:1 ($^{35}Cl^{35}Cl$ : $^{35}Cl^{37}Cl$ : $^{37}Cl^{37}Cl$) and 3:1 ($^{35}ClNO_2$ : $^{37}ClNO_2$) respectively. Mass 201 ($^{37}Cl^{37}Cl$) was not considered in the data processing due to artifacts.

$$ClNO_{2(g)} + I^- \rightarrow IClNO_{2\ (g)}^- \tag{R6}$$
$$Cl_{2(g)} + I^- \rightarrow ICl_{2\ (g)}^- \tag{R7}$$

Calibrations of $Cl_2$ and $ClNO_2$ were carried out during and after the campaign. $Cl_2$ in a cylinder (Airgas, 10 ppm in $N_2$) was diluted with zero air to be sampled in either ambient or scrubber (charcoal) mode (Figure S1). The $Cl_2$ in the cylinder was quantified through the method described by Liao et al. (2012) and was $8.84 \pm 0.43$ ppm. $ClNO_2$ was synthesized, based on Thaler et al. (2011). Briefly, $Cl_2$ gas in $N_2$ was passed through a pyrex reservoir (diameter = 1.3 cm, length = 5.5 cm) containing a bed of NaCl (MACRON) and $NaNO_2$ (Sigma Aldrich) with a molar ratio of 10 to 1. This slurry mixture contains $NO_2^-$ that reacts with the flowing $Cl_2$ to generate $ClNO_2$. The output flow was further diluted with 4 L min$^{-1}$ of zero air in order to sufficiently provide gas flow. The flow containing synthesized $ClNO_2$ was then analyzed at *m/z* of 208 and 210 with the CIMS. $NO_2$ and $NO-NO_y$ were simultaneously measured with a Cavity Ring Down Spectroscopy (CRDS, Los Gatos Research, detection limit: 10 pptv, precision: 50 pptv at $1\sigma$, model: 907-0009-0002) and chemiluminescence (CL, Thermo Scientific, detection limit: 50 pptv, model: 42 i) respectively. $ClNO_2$ is detected as $NO_y$ in the CL through conversion to NO on the heated (325 $^oC$) molybdenum catalytic converter (Williams et al., 1998). The efficiency of the conversion was assumed to be unity. Therefore, $ClNO_2$ could be determined by comparing the three instruments and subtracting the byproducts (HONO and $NO_2$) from the total $NO_y$. The averaged sensitivity of $Cl_2$ was $31.5 \pm 11.2$ Hz/ppt and $ClNO_2$ was $19.7 \pm 1.5$ Hz/ppt. The 2 sigma detection limits of $Cl_2$ and $ClNO_2$ were 2.9 and 1.5 ppt, respectively, over 30 min.

### 2.3 Modeling

We used Framework for 0-D Atmospheric Modeling (F0AM v3.1) for simulating three types of simulations: 1) daytime $Cl_2$ production (Figure 5), 2) in-situ $ClNO_2$ production in the morning (Figure 8), and 3) testing the impact of measured $ClNO_2$ on the regional tropospheric chemistry (Figure 10). F0AM is a MATLAB based open-source box model. Detailed descriptions of the model can be found in Wolfe et al. (2016). Each step of the model was constrained with the averaged meteorology parameters (e.g., pressure, temperature, relative humidity) and trace gases observed in the two ground sites during the campaign. The constrained trace gases include $ClNO_2$, $Cl_2$, $O_3$, NO, $NO_2$, CO, $CH_4$, and 20 non-methane hydrocarbons including 8 alkanes (i.e., ethane, propane, iso-butane, n-butane, iso-pentane, n-pentane, n-hexane, and n-heptane), that have relatively high reaction rate constants with $Cl\cdot$. A constant meteorology and trace gas observation set, collected at the corresponding time period, were

constrained throughout the 72-hour model simulation presented in Figure 5. Then, the $Cl_2$ concentrations at the end of the 72-hour simulation are compared to simultaneously observed mixing ratios of $ClNO_2$ in Figure 5. Simulations in Figure 8 were similarly constrained as those in Figure 5 but allow $ClNO_2$ concentrations to vary with time in order to assess $ClNO_2$ production predicted by the model. The model simulation presented in Figure 10 was constrained with a diurnal variation of the parameters. A full diurnal cycle of the model was for 24 hours consisting a total of 864 steps and each step was integrated for 100 seconds. Each step of the model was constrained with observations measured at that time of day. To assess the impact of $ClNO_2$ chemistry on net $O_3$ production, all species were constrained except for $NO_2$ and $O_3$, which were initialized with observed values and allowed to vary in time. Photolysis rate constants were derived through the hybrid method (Wolfe et al., 2016) in the F0AM box model. This method uses clear sky solar spectra from the tropospheric ultraviolet and visible radiation model (TUV v 5.2) and cross sections and quantum yields suggested by IUPAC. To capture the effects of pollution on photolysis rates, the ratio of the measured $J_{NO2}$ to the F0AM modeled $J_{NO2}$ was calculated. This ratio was then applied to other photolysis rate constants calculated in the model. Measured $J_{NO2}$ was taken from the DC-8 actinic flux measurements (Charged-coupled device Actinic Flux Spectroradiometer; CAFS) when flying near SMA at altitudes under 1 km. A diurnal cycle was applied to the DC-8 measurement to determine j-values at other times of day. Photolysis rate constants of $ClNO_2$, $Cl_2$, and $ClONO_2$ were not present in the F0AM model and therefore taken directly from the DC-8 measurements to be used in the model runs in this study. The Master Chemical Mechanism v3.3.1 (MCM) was taken from http://mcm.leeds.ac.uk/MCM and embedded in the box model. MCM v3.3.1 has a detailed gas photochemistry (i.e., 5832 species and 17224 reactions), including the oxidation of $CH_4$ and 142 non-methane primary emitted VOCs (Jenkin et al., 2015). Since MCM v3.3.1 only includes Cl· reactions with alkane species, additional chlorine chemistry was embedded in the model, similar to what Riedel et al. (2014) reported. This was done by including multiple Cl· precursors (e.g., $Cl_2$, $ClNO_2$, HCl, $ClONO_2$, HOCl) and Cl· reactions with non-alkane VOCs, such as alkene, alcohol, aromatics, alkynes, ketones, organic acids and nitrates. All the reactions embedded in the model can be found in the supplementary of Riedel et al. (2014) and Wolfe et al. (2016). Boundary layer height, emissions, and depositions were not considered in the model. More details on the setup of the box model are in the supplement material (S3). The FLEXible PARTi-cle dispersion model (FLEXPART v9.1, https://www.flexpart.eu) was used for the air mass source contribution (Figure 3) and backward trajectory analysis (Figure 9). The backward trajectories reported in our study were initialized 9:00 LST at TRF, following it 24 hours back in time. Only the center of the mass-weighted particles are shown in Figure 9 and clusters are included in the supporting information. These clusters represent fractional contributions of air masses (Figure S10). The trajectories were driven by the National Centers for Environmental Prediction (NCEP) Global Forecast System (GFS) with a 0.25 degree resolution. Influence of air mass originating from the ocean at TRF and OP was calculated every 6 hours following an air mass 5 days back in time. Meteorology was driven by WRF with a 5 km horizontal resolution. Since emissions of CO are very low in the ocean, and assumed to be inert in the model, it was used as a tracer for contribution of air originating from the ocean within a given air mass at each ground site.

## 3 Results and Discussions

### 3.1 ClNO$_2$ Observations

Figure 2 shows the temporal variation of trace gases measured during the campaign at (a) the OP site (May 17th - June 11th) and (b) the TRF site (May 5th - June 11th). The OP site, which was located near heavy traffic, showed high levels of NO$_x$ throughout the campaign. During most nights (except for May 24th - 26th, 30th - 31st, and June 6th - 7th), O$_3$ was completely titrated by NO. On the other hand, at the TRF site, which is a forested region downwind of the urban area, O$_3$ remained at $\sim$ 30 ppbv throughout the night. During the measurement period, measurable amounts of ClNO$_2$ were observed at both ground sites (Figure 3). The maximum observed ClNO$_2$ was $\sim$800 pptv (5 min averaged) and $\sim$2.5 ppbv (5 min averaged) at the OP and TRF sites, respectively. At both sites, ClNO$_2$ started accumulating at sunset and rapidly photolyzed upon sunrise, which was $\sim$5:30 local standard time (LST) during the campaign. Nighttime relationship between ClNO$_2$ and Cl$_2$ varied day by day and did not show a clear correlation. This implies that the sources or loss processes of Cl$_2$ and ClNO$_2$ were not consistent at night. This is similar to Riedel et al. (2012), where they reported a wide range of correlation between Cl$_2$ and ClNO$_2$ off the coast of LA.

Daytime (11:00 - 18:00, LST) ClNO$_2$ was up to $\sim$100 pptv at OP and $\sim$250 pptv at TRF (Figure 4). The level showed a positive correlation with Cl$_2$, especially in relatively high O$_3$ conditions ($>$ 50 ppbv). When O$_3$ was relatively low ($<$ 50 ppbv), Cl$_2$ production was suppressed, while ClNO$_2$ was not necessarily limited. Excluding the days with low O$_3$ (i.e., May 26th and 29th for OP and May 6th, 29th, and June 4th for TRF), the relationship between daytime ClNO$_2$ and Cl$_2$ showed positive correlation with R$^2$ of 0.49 and 0.80 for OP and TRF, respectively. This positive correlation is consistent with the results reported by Liu et al. (2017) in the North China Plain. In their study, up to $\sim$ 450 pptv of both Cl$_2$ and ClNO$_2$ was measured during the daytime (10:00 - 20:00, LST), with strong correlation of R$^2$ = 0.83. Cl$_2$ levels were also suppressed in low O$_3$ and OH conditions during low solar radiation periods. Therefore, the authors suggested that daytime Cl$_2$ levels could be positively related to photochemical activities. Considering the short lifetime of Cl$_2$ and ClNO$_2$ during the day (i.e., 11:00 - 18:00 LST in our study), the levels we observed are likely affected through local production. According to Liu et al. (2017), the air mass showed moderate correlation to SO$_2$ with possible influences from power plants. However, in this study, the ClNO$_2$ measured at both the OP and TRF sites was weakly correlated with SO$_2$ (R$^2$ = 0.02), which implies that the air masses that we sampled are not fresh emissions from coal combustion activities such as power plants.

The first possibility we explored is the direct generation of Cl$_2$ from reactions in acidic particles. ClNO$_2$ is very insoluble ($\gamma_{ClNO2} \approx 10^{-6}$ (Rossi, 2003)) in near-neutral pH. However, according to Roberts et al. (2008), $\gamma_{ClNO2}$ can increase up to 3 orders of magnitude in acidic surfaces ($\sim$ pH 1.8) leading to direct production of gas-phase Cl$_2$. Aerosol acidity was mostly below pH 2 during the campaign, based on thermodynamic calculations, constrained with airborne observations (Figure S3). Therefore, the efficiency of this reaction in ambient conditions requires further investigation. Another possibility is the autocatalytic production of Cl$_2$ from heterogeneous reactions of gas-phase ClONO$_2$ (i.e., ClONO$_{2(g)}$ + Cl$^-_{(aq)}$ + H$^+_{(aq)}$ $\rightarrow$ Cl$_{2(g)}$ + HNO$_3$, (Gebel and Finlayson-Pitts, 2001; Deiber et al., 2004)) and HOCl (i.e., HOCl$_{(g)}$ + Cl$^-_{(aq)}$ + H$^+_{(aq)}$ $\rightarrow$ Cl$_{2(g)}$ + H$_2$O, (Vogt et al., 1996)) on particles. These reactions are also favored as particle acidity increases. In order to further

investigate its possibility, daytime $Cl_2$ was simulated by constraining the box model with measurements of $ClNO_2$ and other trace gases corresponding to each data point in Figure 4. Based on the availability of parameters, we were able to simulate 1680 and 1229 runs for the OP and TRF, respectively. This corresponds to more than 96 % of the daytime data points shown in Figure 4. $\gamma_{ClONO2}$ and $\gamma_{HOCl}$ were set to 0.06 (Deiber et al., 2004; Hanson et al., 1994; Hanson and Ravishankara, 1994), which is an upper-limit of previous laboratory studies, and the yields were assumed to be unity. HCl generation from hydrogen abstraction of VOCs by $Cl^{.}$ were included in the mechanisms used in the model runs. The end points of the 72 hour simulation results are presented in Figure 5. As shown in the figure, the box model simulations were able to reproduce the positive correlation between $Cl_2$ and $ClNO_2$. Moreover, modeled $Cl_2$ was suppressed in low $O_3$ conditions, which corresponds to the observations. This can be explained by $Cl^{.}$ reacting with $O_3$, producing $ClO^{.}$, leading to gas-phase $ClONO_2$ and HOCl production. These can react on acidic aerosols to generate $Cl_2$. Sources of $Cl^{.}$ could be from photo-labile gas-phase chlorine compounds (e.g., $Cl_2$, $ClNO_2$, $ClONO_2$, HOCl) or oxidation of gas-phase HCl by OH. Although the reaction between HCl and OH is relatively slow (k = $7.86\times10^{-13}$ $cm^3$ $molecule^{-1}s^{-1}$ at 298K, (Atkinson et al., 2007)), it has been reported to be a significant source of $Cl^{.}$ in the daytime (Riedel et al., 2012). A sensitivity test was carried out by comparing modeled $Cl_2$ between runs with and without HCl production from oxidation of VOCs by $Cl^{.}$ (Figure S4 c,d). The results show that production of $Cl_2$ was suppressed by 40 - 70 % when HCl was not generated in the model. This significant contribution of gas-phase HCl as a $Cl^{.}$ source, should be an upper-limit as the deposition of HCl was not considered in the model. Nonetheless, our analysis leads us to conclude that the mechanisms we have explored could be the main contributors of the daytime $Cl_2$ production during KORUS-AQ.

### 3.2 Sources of $ClNO_2$

FLEXPART source contribution analysis shows that the level of $ClNO_2$ at the ground sites was highly correlated with the origin of the air mass (Figure 3). During the nights shaded in red in Figure 3 (OP: May 20th, 22nd, June 2nd, and 7th; TRF: May 11th, 19th-22nd, June 2nd, and June 6th-7th), there was limited production of $ClNO_2$ at the surface. These periods mostly corresponded to meteorological conditions of stagnation or blocking events, which both resulted in localized air masses to be more dominant with limited influence from the west coast. Stagnation events can be characterized by low wind speeds and increased atmospheric stability, possibly leading to enhanced levels of pollutants like $NO_x$. Previous studies have shown that stagnant conditions can result in enhanced levels of $N_2O_5$ driven by high ozone and $NO_2$. However, $ClNO_2$ production was limited during stagnation events in this study. This is likely due to limited availability of chloride as shown in submicron particle measurements of aerosol mass spectrometer (AMS) at the ground site for OP and airborne over TRF (Figure 3). Whether the chloride is from the ocean or anthropogenic emissions is uncertain since large point sources, such as power plants or petrochemical facilities, are also present along the west coast of the SMA. On the nights of May 20th and May 22nd, rapid changes in air quality were observed with fast shifts in $O_3$, $SO_2$, and CO. This corresponded with changes in $ClNO_2$ and $Cl_2$ (Figure S7). These events suggest the importance of boundary layer advection in controlling the $ClNO_2$ levels in the region.

Different diurnal variations of $ClNO_2$ were observed between OP and TRF (Figure 6). The measurements were averaged over selected days (OP: May 18th-20th, 22nd, 23rd, 29th, June 4th; TRF: May 5th, 8th, 9th, 12th, 17th, 18th, 30th, June 8th, 10th) that showed these two distinct profiles at each site. The description on these profiles are further explained in the following

sentences. At the TRF site (Figure 6b), far from direct NO emissions, significant levels of $ClNO_2$ were sustained throughout the night during most of the observation period with rapid photolysis upon sunrise. On the other hand, at OP (Figure 6 (a)), $ClNO_2$ started to increase upon sunset, followed by a rapid drop at around 22:00 LST. The trend was consistent with slower nitrate radical production rate ($d[NO_3]/dt = [NO_2][O_3]k$, where $k = 3.52 \times 10^{-17}$ at 298 K,(Atkinson et al., 2004)) as $O_3$ was titrated to zero by NO close to midnight. The wind direction, $SO_2$, and CO did not correlate. This suppressed $ClNO_2$ production in urbanized regions with high NO levels, have also been reported by Osthoff et al. (2018). However, significant levels of $N_2O_5$ and $ClNO_2$ could have been present in the upper part of the surface layer as shown in previous studies (Baasandorj et al., 2017; Young et al., 2012; Yun et al., 2018). According to Baasandorj et al. (2017), $O_3$ was completely titrated at the surface in Salt Lake Valley, Utah, while elevated mixing ratios of $N_2O_5$ were observed at 155 meters above ground level, at a site along the valley wall. On the other hand, airborne measurements at the LA basin (Young et al., 2012) showed a relatively uniform $ClNO_2$ profile throughout the nocturnal boundary layer as $O_3$ did not change significantly within the observed altitude range (< 600 m). During the 2015 Megacity Air Pollution Study (MAPS, Seoul, 2015), a Cavity Ringdown Spectrometer (CRDS) was installed on top of the Seoul tower in May - June that measured $N_2O_5$, $NO_x$, and $O_3$ (Brown et al., 2017). The elevation of the measurement site was 360 m above sea level (ASL), allowing for sampling further away from direct NO emissions. In their study, the average nighttime $O_3$ mixing ratio was around 50 ppbv and $N_2O_5$ was observed most nights, with mixing ratios reaching up to 5 ppbv. Therefore, it is very likely that $ClNO_2$ levels higher than the surface measurements could have been present at higher elevation during the observation period.

At both sites, $ClNO_2$ levels started to increase or sustained after the first 2-3 hours of rapid net loss upon sunrise. In the morning, $ClNO_2$ positively correlated to $Cl_2$ levels, but did not follow the nitrate production rate at the site (Figure S8). Box model simulations, initially constrained with observed $ClNO_2$ level, showed rapid photolysis upon sunrise (Figure S5, red dashed line). At TRF, this corresponded to the measurements until 7 - 8 am LST, when a second $ClNO_2$ peak was observed (Figure 6b). This $ClNO_2$ peak in the morning was observed about half the observation days during the campaign. With the net $ClNO_2$ production rate from the observation, and the loss rate from the simulated $ClNO_2$ from photolysis, a production rate of 400 pptv h$^{-1}$ would be required to reconcile the observation. In the case of $ClNO_2$ observed on May 5th at TRF (an insert of Figure S6b ), a maximum of 2.5 ppbv h$^{-1}$ of $ClNO_2$ production rate was required in the morning to reconcile the observations. At OP, 18 pptv h$^{-1}$ was required for the 7 averaged days. The $ClNO_2$ production rate required in the morning at TRF was much higher than the previous studies that have also reported high sustained levels of $ClNO_2$ in the morning (i.e., 20 - 200 pptv h$^{-1}$) (Faxon et al., 2015; Bannan et al., 2015; Tham et al., 2016). In these previous studies, three possibilities have been suggested that could explain the high sustained levels of $ClNO_2$ in the early morning : 1) in-situ generation of $ClNO_2$, 2) transport of $ClNO_2$ within the boundary layer, and 3) entrainment of $ClNO_2$ from residual layer. Each possibility is explored below.

In order to explore the possibility of in-situ formation, box model simulations of $ClNO_2$ production from heterogeneous reaction of $N_2O_5$ and chloride containing aerosols were conducted. $N_2O_5$ was calculated assuming a photo-stationary state of $NO_3^-$ (Brown et al., 2005). Aerosol surface area was taken from airborne observations over TRF. Based on the box model results in Figure 8, even with an assumption of 100 % yield, $ClNO_2$ from heterogeneous reaction was not able to reconcile the observed

level. Using the dry surface area for the first order loss of $N_2O_5$ on aerosols certainly could result in an underestimation of $ClNO_2$ production in the model. Kim et al. (2017, 2018) observed hygroscopic growth factor of less than 1.5 in the SMA region for particles below 150 nm during the KORUS campaign period. In other words, the discrepancy between observed and modeled $ClNO_2$ of more than 50-fold cannot be reconciled by this underestimation. The box model simulation on gas phase

production of $ClNO_2$ (i.e., $Cl\cdot_{(g)} + NO_{2(g)} + M \rightarrow ClONO_{(g)} + M$, $Cl\cdot_{(g)} + NO_{2(g)} + M \rightarrow ClNO_{2(g)} + M$) showed at most 2-10 pptv of $ClNO_2$ and $ClONO$ (Figure S6).

Therefore, horizontal or vertical transport from local sources would be the most likely explanation for the high $ClNO_2$ in the morning. Although $ClNO_2$ readily photolyzes during the day ($\tau_{ClNO2} \approx 30$ min at midday), the lifetime could be significantly long enough in the early morning to allow for transport of $ClNO_2$ to the ground sites. Based on the NCAR TUV v5.2 model,

the lifetime of $ClNO_2$, averaged between 5:30 and 8:30 LST was $\sim 2$ hours under clear sky conditions. Figure 9 shows back trajectory analysis initiated at 9 am local time at TRF. At high $ClNO_2$ days with the morning peaks, most of the air masses were from the west. During KORUS, the DC-8 did not fly to the west of the SMA in the early morning. However, there are large point sources, such as petrochemical facilities and industries, and vehicular emissions to the west and south west of the SMA region. Sullivan et al. (2019) reported that this resulted in enhanced levels of $O_3$ in receptor regions (i.e., Taehwa Research

Forest) downwind when westerlies were prevalent. Therefore, favorable conditions such as high chloride content in aerosols from both anthropogenic and natural sources and high levels of $NO_x$-$O_3$ could have led to significant levels of $ClNO_2$ to build up and transported to TRF before being completely photolyzed. During the campaign, influence of large biomass burning was negligible as reported in Tang et al. (2018, 2019).

At night time, the nocturnal boundary layer is decoupled from the residual layer (Stull, 1988), where the pollution from

the previous day resides. Being removed from direct NO emissions near the surface, $N_2O_5$ can effectively accumulate in the residual layer, with the major loss process being heterogeneous reaction on aerosols. Therefore, high levels of $NO_2$ and $O_3$ formed during the day can be trapped in the residual layer resulting in significant levels of $ClNO_2$ persisting throughout the night. Figure 7, shows (a) regional and (b) vertical distribution of airborne $ClNO_2$ throughout the campaign in the morning (8:00 - 8:30 LST) over the SMA region (lat: $37^o$ 12' 0" N - $37^o$ 38' 60" N, lon: $126^o$ 54' 0" E - $127^o$ 47' 60" E). During 3 flights

(i.e., May 25th, May 31st, and June 10th), $ClNO_2$ was observed in the residual layer with a max of $\sim 230$ pptv. However, the remaining flights observed an average of $17 \pm 56$ pptv of $ClNO_2$ (black circles). Even the three days (i.e., May 25th, 31st, and June 10th), that $ClNO_2$ was observed in the residual layer, the level (max 230 pptv) could not reconcile the observed levels at the TRF site, which was $342 \pm 330$ pptv when averaged over the corresponding 3 days at 8:00 - 8:30 LST. However, it is possible that the air mass that was measured by the DC-8 was not representative of the air mass aloft at the west side of the

ground observation sites. Backtrajectory analysis initialized at 9:00 local time showed that the TRF site was affected by both the residual layer and below (Figure S10). The enhancement of $O_3$ and $SO_2$ concurrent to elevation of $ClNO_2$ could be due to the transport from the residual layer where pollution from high point sources from the other day was trapped within. From the current dataset, it would be difficult to derive a clear conclusion on whether the cause of the significant $ClNO_2$ in the morning was dominantly of transport from horizontal, vertical, or both.

## 3.3 Impacts of $ClNO_2$ on $O_3$

Cl· produced from $ClNO_2$ photolysis can influence the local air quality through reactions with VOCs followed by enhanced production of $O_3$. The possible impact of Cl· initiated reactions on the local chemistry were investigated by running box model simulations constrained with measured $ClNO_2$. A 24 hour diurnal variation of $ClNO_2$ was averaged over the same selected days as in Figure 6, and these were constrained throughout the model simulations. The results illustrate that when the model was constrained with $ClNO_2$ and Cl· initiated chemistry, higher levels of $O_3$ were simulated (Figure 10) compared to the base runs without $ClNO_2$. The averaged net $O_3$ production rate was enhanced by up to 2 % and 25 % at OP and TRF in the morning and by 1 % and 2 % when averaged during the day. The OP had 7 times lower Cl· than the TRF site due to low $ClNO_2$ levels ($\sim$60 pptv) in the morning. Since the box model simulations in our study did not take into consideration boundary layer height dynamics, emission, and deposition, this net production rate is the result of just chemical production and loss. For ·OH, the net production rate at TRF increased by 2 % in the morning. The results particularly from TRF are comparable with the previous study in the mountaintop site in Hong Kong, China (Wang et al., 2016). The enhancement of $O_3$ (max - min) was higher than their moderate $ClNO_2$ case (11 %) but lower than the high $ClNO_2$ plume case (41 %).

## 4 Conclusions

Comprehensive measurements of $ClNO_2$, $Cl_2$, other trace gases, and aerosol concentrations and properties have been conducted on the NASA DC-8 and at two ground sites during the KORUS-AQ 2016 field campaign. The observed averaged diurnal variations are largely consistent with the previous observations and our understanding on the photochemistry of $ClNO_2$. The presence of $ClNO_2$ was substantially suppressed during strong stagnation events, which could have prevented the transport of chloride near the coast. During the night, $Cl_2$ and $ClNO_2$ levels were not correlated while moderate to strong positive relationships were observed at daytime. Through box model simulations, we presented a quantitative analysis of the daytime observations. The results showed that heterogeneous reactions of $ClONO_2$ and $HOCl$ in acidic aerosols may be responsible for the positive correlation between $Cl_2$ and $ClNO_2$, as well as its dependency on $O_3$. The second $ClNO_2$ peak in the morning, observed 4-5 hours after sunrise, required a significant source of $ClNO_2$ (up to 2.5 ppbv $h^{-1}$). Previous studies have attributed high sustained $ClNO_2$ in the morning to transport from the residual layer (Tham et al., 2016; Wang et al., 2016). In this study, box model runs of heterogeneous and gas-phase production of $ClNO_2$ could not reconcile the observed levels. Airborne observations near the ground sites in the early morning showed negligible $ClNO_2$ levels in the residual layer in most of the days. However, there is still a possibility of the contribution of vertical transport from the residual layer. Although the current data set is limited for us to pinpoint on the vertical locations (i.e., boundary layer v.s. residual layer), back trajectories illustrate that $ClNO_2$ rich air masses were mostly transported from the west, where there are significant sources of precursors. This shows that different meteorological or chemical conditions of the sites can lead to various causes of high $ClNO_2$ levels in the early morning. Finally, box model simulations constrained with observations suggest that Cl· initiated chemistry can lead up to $\sim$25 % increase of net chemical $O_3$ production rate in the morning.

*Data availability.* Dataset used in this study is open to public and can be downloaded at https://www.air.larc.nasa.gov/missions/korus-aq/

*Author contributions.* DJ, RS, DG, YL, DT, SK, and GH designed and executed field measurements for collecting $ClNO_2$ and $Cl_2$; BAN, JLJ, and PCJ provided the airborne AMS data; CK ran the FLEXPART analysis; TM and JS provided $O_3$ measurements; DRB provided
the WAS data; DS and AG provided the PTR-ToF-MS data; RL provided $NO_x$ and $O_3$ data; BA provided the data from LARGE; SRH and KU provided data from CAFS; HS provided AMS data at OP; SH provided HCHO measurements at TRF; YL, DK, and JA provided CO measurements at TRF. DJ and SK prepared the original manuscript, and all other authors contributed in editing the manuscript.

*Acknowledgements.* This study is supported by NIER and NASA (NNX15AT90G). CU HR-AMS measurements and pH and aerosol liquid water calculations (BAN, PCJ, and JLJ) were supported by NASA grant NNX15AT96G and 80NSSC18K0630. We thank the Wisthaler
research group (University of Oslo, University of Innsbruck) for providing airborne VOC data, John Crounse and Paul Wennberg(CALTECH) for the $HNO_3$ data, Hwajin Kim (Korea Institute of Science and Technology) for the discussions on aerosol composition, and Siyuan Wang (NCAR) for discussions on box model simulations. The authors appreciate logistical support from the research and supporting staff at Taehwa research forest operated by Seoul National University.

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

**Table 1.** Summary of the measurements carried out during the KORUS-AQ 2016 field campaign, used in this study.

| Compound | Method | | |
|---|---|---|---|
| | TRF | OP | DC-8 |
| NO | [a]CL | [b]FRM | [c]CL |
| $NO_2$ | [d]CRDS | [e]CAPS | [c]CL |
| $O_3$ | [f]DIAL | [g]SL-UV | [c]CL |
| $SO_2$ | [h]PF | [i]UV florescence | x |
| CO | [j]IR | [k]NDIR | x |
| $ClNO_2, Cl_2$ | [l]CIMS | [l]CIMS | [l]CIMS |
| VOCs | [m]PTR-ToF-MS | [o]QCL | [p]PTR-ToF-MS |
| | [n]TILDAS | | [q]WAS |
| chloride (< 1 $\mu$m) | x | | |
| nitrate (< 1 $\mu$m) | x | [r]ToF-AMS | [s]HR-ToF-AMS |
| sulfate (< 1 $\mu$m) | x | | |
| surface area (< 200 $nm$) | x | x | [t]SMPS |
| surface area (200 $nm$ - 5 $\mu$m) | x | x | [u]LAS |
| Jvalues | x | x | [v]CAFS |

[a]Chemiluminescence with a molybdenum converter (Thermo Scientific 42i - TL), [b]Chemiluminescence detector (Federal Reference Method, Teledyne T200U), [c]NCAR 4-channel chemiluminescence (Weinheimer et al., 1994), [d]Cavity Ring Down Spectroscopy (Los Gatos Research $NO_2$ analyzer), [e]Cavity Attenuated Phase Shift spectroscopy (Teledyne T500U CAPS analyzer), [f]NASA TROPospheric OZone DIfferential Absorption Lidar (Sullivan et al., 2014), [g]UV photometric method (2B 211), [h]Pulsed fluorescence method (Thermo Scientific 43i-HL), [i]UV florescence method (KENTEK), [j]Infrared CO analyzer (Thermo Scientific 48i-HL), [k]Non-Dispersive Infrared CO analyzer (KENTEK), [l]Chemical Ionization Mass Spectrometer (Slusher et al., 2004), [m]Proton-Transfer-Reaction Time-of-Flight Mass Spectrometer (IONICON), [n]Tunable Infrared Laser Direct Absorption Spectroscopy (Aerodyne), [o]Quantum Cascade Laser spectrometer (Aerodyne), [p]University of Oslo/Innsbruck Proton-Transfer-Reaction Time-Of-Flight Mass Spectrometer (Müller et al., 2014), [q]Whole Air Sampler (Colman et al., 2001), [r]Aerosol Mass Spectrometer (Aerodyne), [s]University of Colorado, Boulder,Aerosol Mass Spectrometer (Nault et al., 2018), [t]NASA, Scanning Mobility Particle Sizer, [u]NASA, Laser Aerosol Spectrometer, [v]NCAR, Charged-coupled device Actinic Flux Spectroradiometer (Shetter and Müller, 1999)

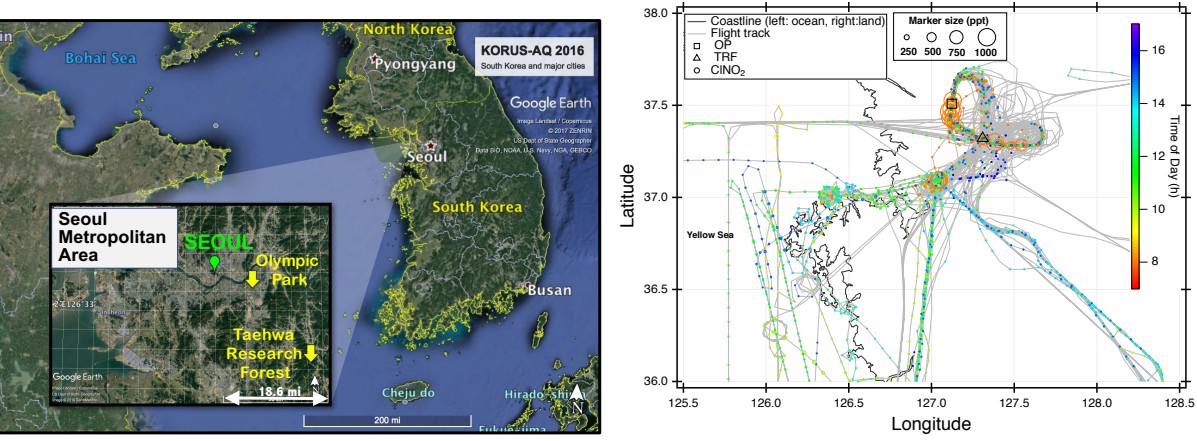

**Figure 1.** (a) Location of two ground sites (Taehwa Research Forest and Olympic Park) where the chemical ionization mass spectrometer (CIMS) was installed during the KORUS-AQ 2016 field campaign (b) Airborne measurements of ClNO$_2$ and DC-8 flight tracks during the whole campaign. The ClNO$_2$ data points are 60 sec averaged and color coded by time of day of the measurement. The marker size is proportional to the mixing ratio of ClNO$_2$.

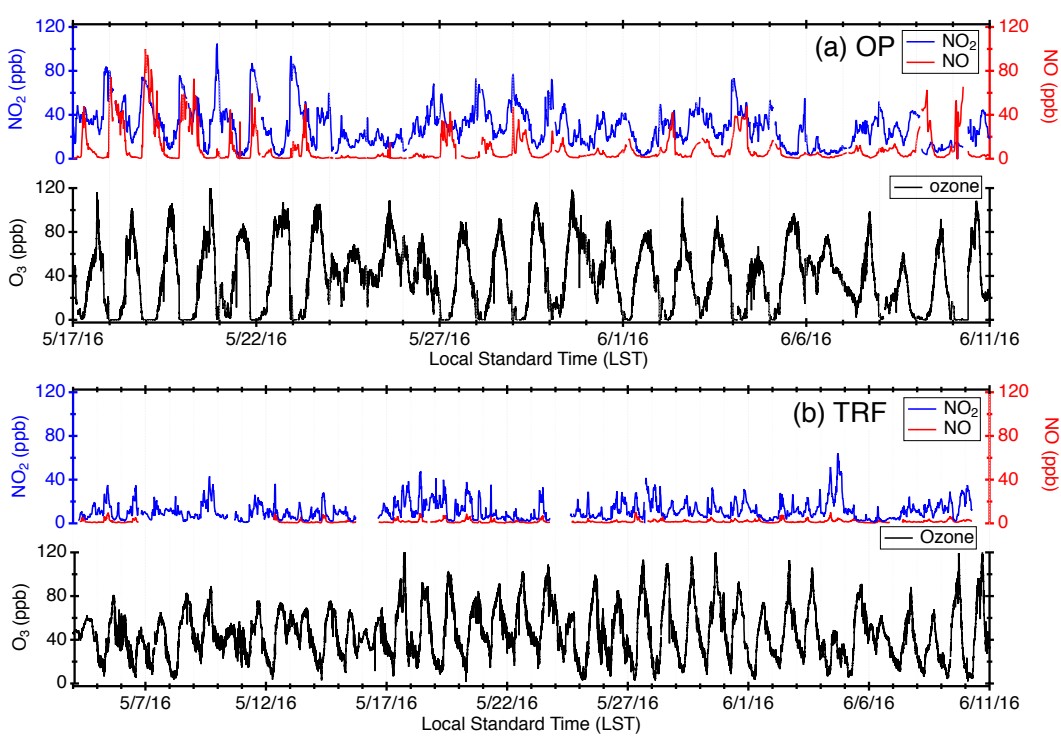

**Figure 2.** Temporal variation of trace gases measured at the (a) Olympic Park site (OP) and (b) Taehwa Research Forest (TRF). For both OP and TRF, the frequency of the averaged data is 10 min for $NO_x$ and 1 min for $O_3$.

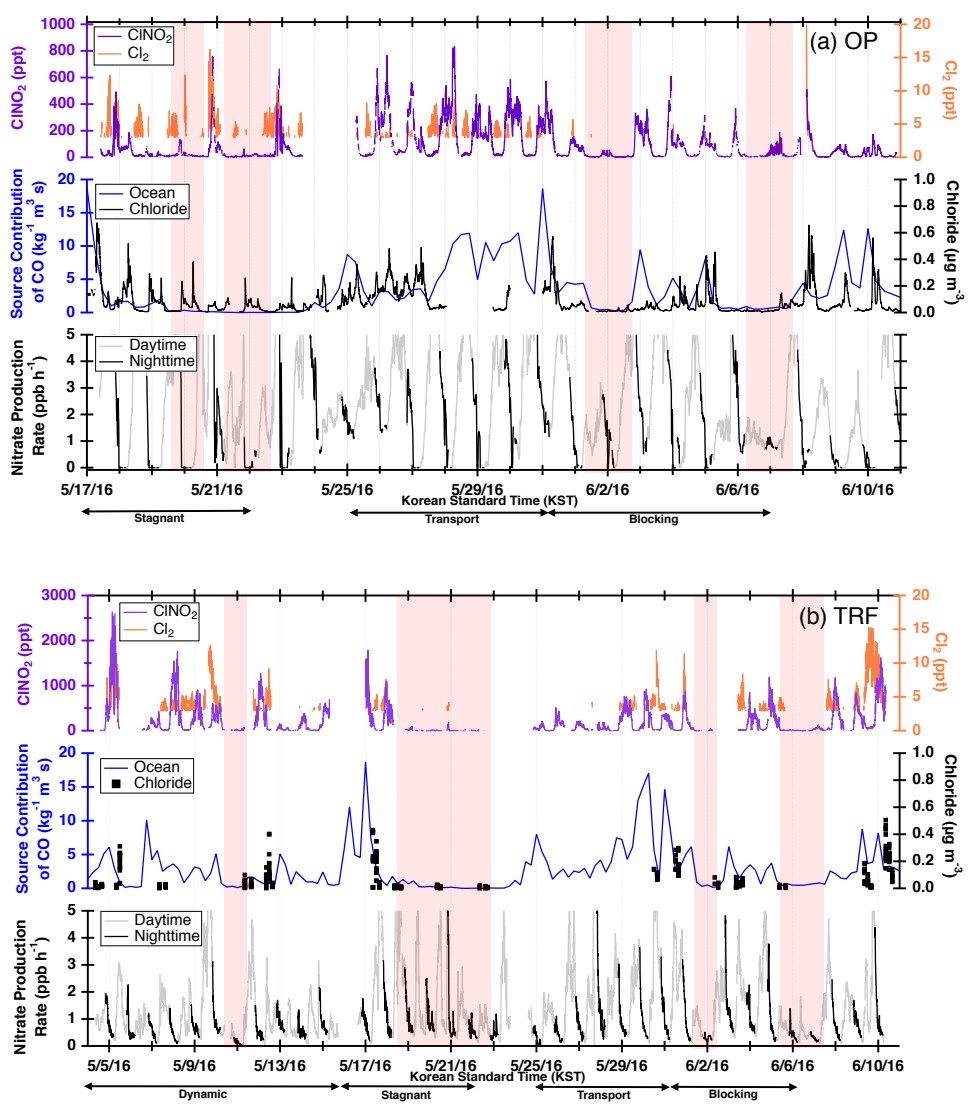

**Figure 3.** ClNO$_2$ and Cl$_2$ observation results at (a) OP and (b) TRF averaged over 5 min. FLEXPART back trajectory analysis were made for source contribution of CO-like substance originating from the ocean, assuming inert CO. Aerosol chloride mass concentration (ambient $\mu$g m$^{-3}$) was measured at the ground for the OP site and on the NASA DC-8 for TRF. For the airborne chloride, measurements below 1 km over the TRF site is shown. Red shades are the time frames with limited ClNO$_2$ production. The time frames for each meteorological condition that dominated during the observation period are classified in black arrows at the bottom of the Figures 3a and 3b

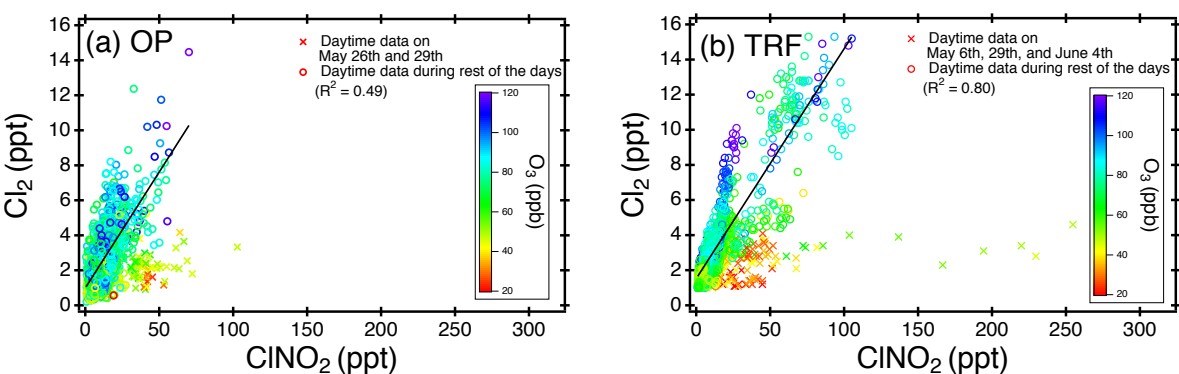

**Figure 4.** Scatter plot of daytime (11:00 - 18:00 local time) ClNO$_2$ and Cl$_2$ at (a) OP and (b)TRF, color coded with measured O$_3$. 5 min averaged data for the whole campaign were used for both sites. Data points of Cl$_2$ below detection limit (2.9 ppt, 2$\sigma$, over 30 min) are shown for the purpose of comparison to observed ClNO$_2$ levels.

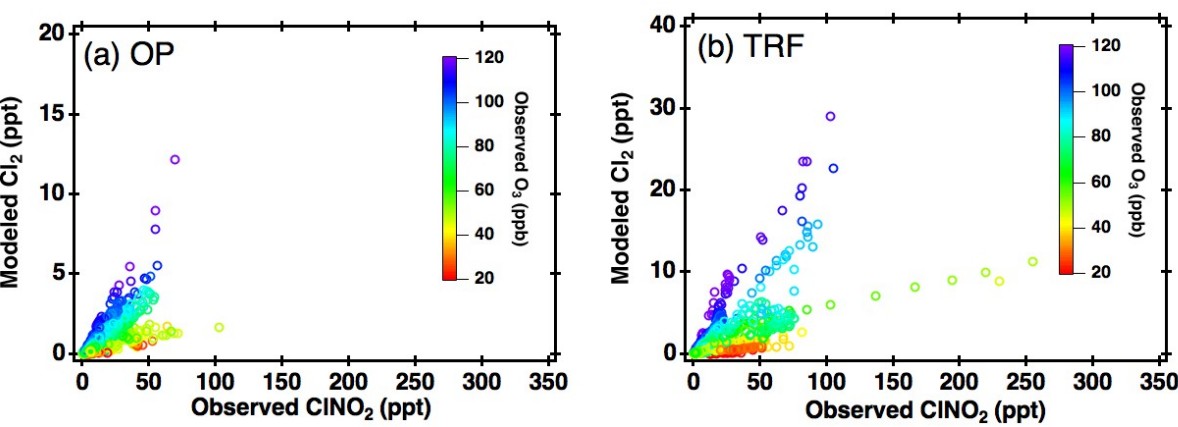

**Figure 5.** Correlation between box model simulated daytime (11:00 - 18:00 local time) Cl$_2$ and measured ClNO$_2$ at (a) OP and (b)TRF, color coded with measured O$_3$.

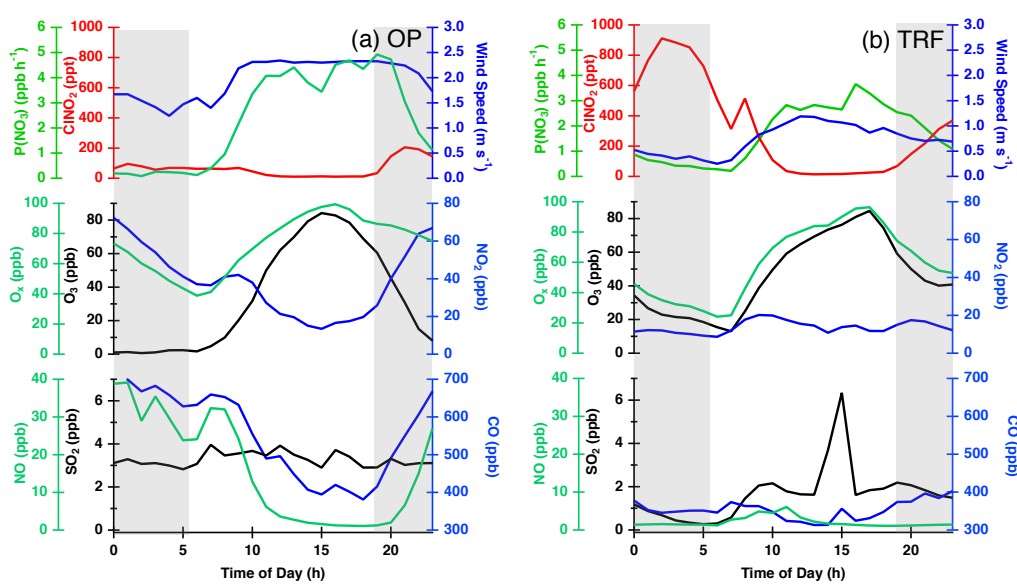

**Figure 6.** Diurnal variation of ClNO₂ and other trace gases measured during the campaign and averaged over selected days at (a) OP (7 days) and (b) TRF (9 days). Night time is shown as grey shades.

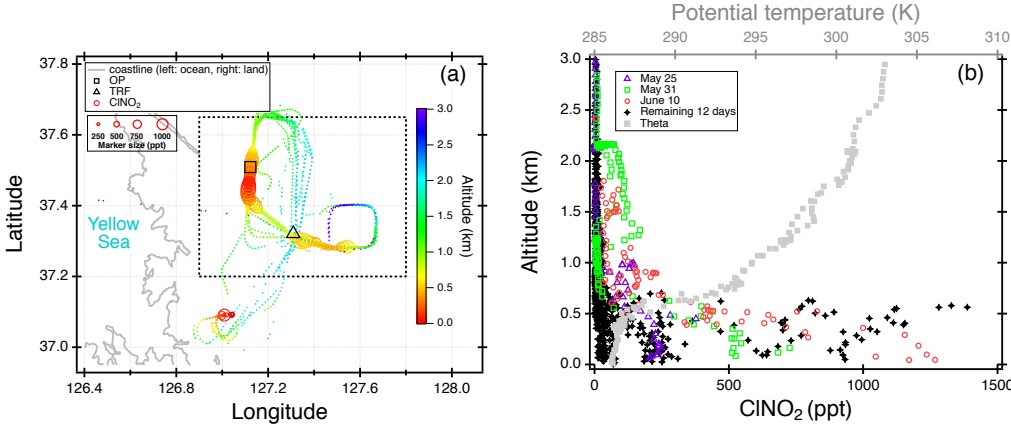

**Figure 7.** (a) Regional and (b) vertical distribution of airborne ClNO$_2$ measured over the Seoul Metropolitan Region (SMA) in the morning (8:00 - 8:30 local time).

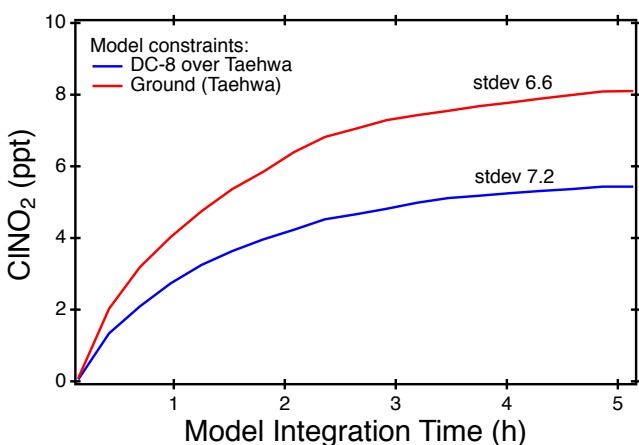

**Figure 8.** Steady state ClNO$_2$, simulated from a box model constrained with airborne measurements (blue) and ground site data from TRF (red), when there was a morning ClNO$_2$ peak. Averaged values of the model runs are shown here with standard deviations.

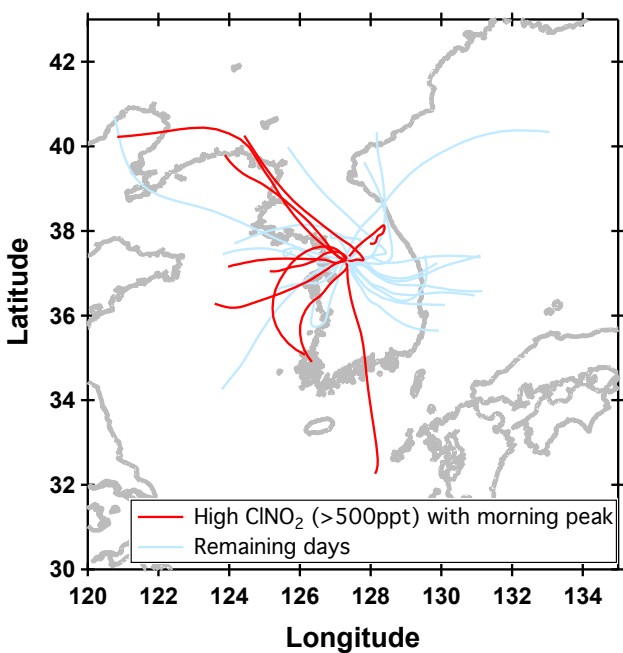

**Figure 9.** FLEXPART back trajectories from Taehwa Research Forest. Trajectories were initialized at 9 am local time and went 24 h backwards. Only the center trajectories with the highest percentage of airmasses are presented. Trajectories for days with high levels of $ClNO_2$ ($> 500$ pptv) at night are in red and the remaining days are shown in sky blue.

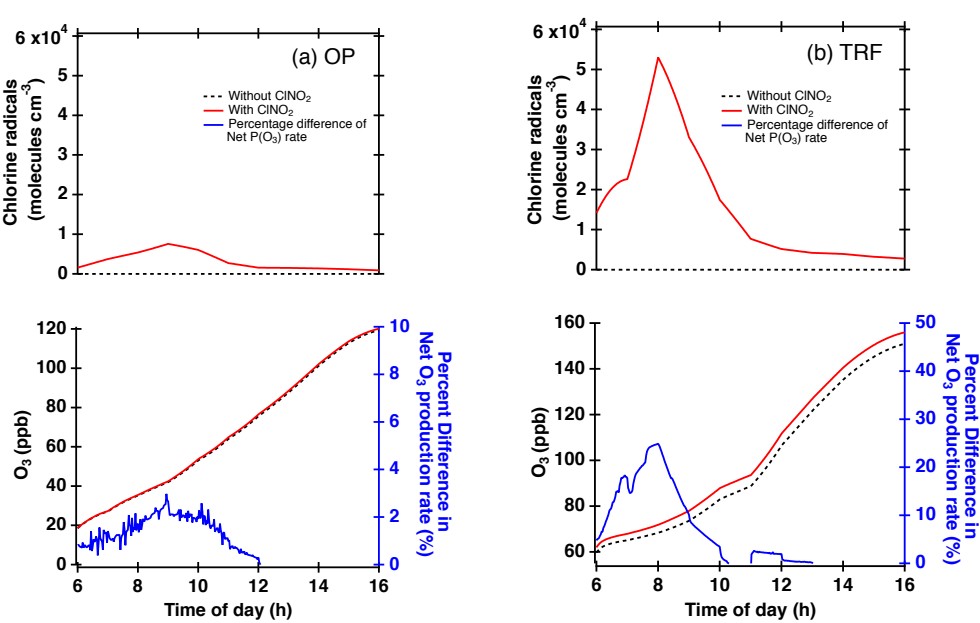

**Figure 10.** Box model simulations of chlorine radicals and $O_3$ at (a) OP and (b) TRF, constrained with $ClNO_2$ and other trace gases observed during the field campaign. Percent difference of net $O_3$ production rate (NetP($O_3$), blue line) was calculated from the difference of the NetP($O_3$) between simulations with and without $ClNO_2$ constrained in the model (i.e., 100*(wClNO$_2$ - woClNO$_2$)/woClNO$_2$).