# Peer review of "Integration of Airborne and Ground Observations of Nitryl Chloride in the Seoul Metropolitan Area and the Implications on Regional Oxidation Capacity During KORUS-AQ 2016"

_Atmospheric Chemistry and Physics, 2018_

## Referee Comment (RC1) · Anonymous Referee #1 · 20 Dec 2018

Jeong et al. reported ground-based and airborne observations of $ClNO_2$ in Korea during the Korean-United States-Air Quality (KORUS-AQ) 2016 field campaign. They analyzed the general characteristics, the sources and the effects of $ClNO_2$ on ozone during their measurement campaign. The study contributes to the growing body of $ClNO_2$ measurements around the world, and the content fits the scope of *Atmospheric chemistry and Physics*. The manuscript can be improved by adding more detailed description of measurements and more in-depth analysis/discussion on the elevated levels of $ClNO_2$ in the morning. In addition, there are a number of places which need to be modified for clarity.

Specific comments

(1) Introduction: the manuscript gave a detailed review of measurements of $ClNO_2$ conducted in North America and Europe, but didn't include some recent work in Asia (mostly in China). As the latter is more relevant to the present study due to proximity of the study regions, these studies should be reviewed (following the review of North America/Europe results). These studies are listed below:

- Wang, H., Lu, K., Guo, S., Wu, Z., Shang, D., Tan, Z., Wang, Y., Breton, M. L., Lou, S., and Tang, M.: Efficient $N_2O_5$ uptake and $NO_3$ oxidation in the outflow of urban Beijing, Atmospheric Chemistry and Physics, 18, 9705-9721, 2018.

- Zhou, W., Zhao, J., Ouyang, B., Mehra, A., Xu, W., Wang, Y., Bannan, T. J., Worrall, S. D., Priestley, M., and Bacak, A.: Production of $N_2O_5$ and $ClNO_2$ in summer in urban Beijing, China, Atmospheric Chemistry and Physics, 18, 11581-11597, 2018.

- Yun, H., Wang, W., Wang, T., Xia, M., Yu, C., Wang, Z., Poon, S. C. N., Yue, D., and Zhou, Y.: Nitrate formation from heterogeneous uptake of dinitrogen pentoxide during a severe winter haze in southern China, Atmospheric Chemistry and Physics, 2018, 23, 10.5194/acp-2018-698, 2018.

- Wang, Z., Wang, W., Tham, Y. J., Li, Q., Wang, H., Wen, L., Wang, X., and Wang, T.: Fast heterogeneous $N_2O_5$ uptake and $ClNO_2$ production in power plant and industrial plumes observed in the nocturnal residual layer over the North China Plain, Atmospheric Chemistry and Physics, 17, 12361-12378, 2017.

- X. Wang, H. Wang, L. Xue, T. Wang, L. Wang, R. Gu, W. Wang, Y. J. Tham, Z. Wang, L. Yang, J. Chen and W. Wang, Observations of $N_2O_5$ and $ClNO_2$ at a polluted urban surface site in North China: High $N_2O_5$ uptake coefficients and low $ClNO_2$ product yields, Atmospheric Environment, 156. 125-134. 2017.

(2) Line 14: The lifetime of $ClNO_2$ depends on the photolysis rate which varies among regions, seasons, weather conditions, etc. It's better to specify the condition in which the lifetime of $ClNO_2$ is ~ 30 min and provide a reference.

(3) Line 17: "resulting from an enhanced $ClNO_2$ uptake coefficient of up to 3 orders of magnitude" is not clear. Is the uptake in the condition of pH<2 "3 orders of magnitude" higher than that in the condition of pH=7? Please elaborate.

(4) Line 21: on source of chloride, Fu et al. (2018) presents a fine chloride emission inventory for China. This can be a good reference here.

- Fu, X., Wang, T., Wang, S., Zhang, L., Cai, S., Xing, J., and Hao, J.: Anthropogenic Emissions of Hydrogen Chloride and Fine Particulate Chloride in China, Environmental Science & Technology, 52, 1644-1654, 2018.

(5) Line 25: Is the heterogeneous uptake of $ClNO_2$ on acidic particle (Roberts et al., 2008) taken into account in the calculation of the lifetime (30h) during the nighttime?

(6) Line 45: These previous studies showed that $ClNO_2$ is ubiquitous at surface, within the boundary layer or in the lower troposphere around the world. No evidence has shown that the $ClNO_2$ is ubiquitous in the troposphere which could reach >10 Km above the sea level in mid-latitude region.

(7) Line 55-57: another modeling study Li et al. (2017) assessed the ozone impact of $ClNO_2$ in East Asia including Korea.
- Zhang, L., Li, Q., Wang, T., Ahmadov, R., Zhang, Q., Li, M., and Lv, M.: Combined impacts of nitrous acid and nitryl chloride on lower-tropospheric ozone: new module development in WRF-Chem and application to China, Atmospheric Chemistry and Physics, 17, 9733-9750, 2017.

(8) Line 58-65: the discussion on model resolution does not seem to be relevant to the present study.

(9) Line 64-65: I assume that the authors are referring to the simulations by Sherwen et al. (2017) that underestimated the $ClNO_2$ by 7 times. Please confirm.

(10) Line 73-78 discusses importance of chorine source in coastal cities and gives the reader an impression that it is only important near coast. But measurement data have shown it is present far inland as shown in Thronton et al. (2010). A recent compilation of PM2.5 data also shows high levels of chloride are present in inland regions of China (Yang et al., STOTEN, 2017). The relevant sentence should be modified.

- Yang, X., Wang, T., Xia, M., Gao, X., Li, Q., Zhang, N., Gao, Y., Lee, S., Wang, X., and Xue, L.: Abundance and origin of fine particulate chloride in continental China, Science of The Total Environment, 624, 1041-1051, 2018.

(11) Section 2.1. It would help to include a brief description of the meteorology during the campaign.

(12) Section 2.2 on CIMS and calibration: more detailed information is needed. What is the length of the sample line? Was it washed or replaced regularly in order to reduce the loss of $N_2O_5$? How frequent was the calibration? Was change of CIMS sensitivity to relative humidity taken into account in data reduction, and how? Could measurement of Cl2 with a Q-CIMS subject to interference? In line 115, 'the natural abundance of $Cl_2$ and $ClNO_2$ isotopes are approximately 9:6:1", what do the authors mean by this statement? No figure was shown and it is not clear how the isotopic ratios behave.

(13) Line 131: How was HONO measured?

(14) Section 2.3. Add description on the calculation of the impact of $ClNO_2$ on $O_3$ production rate and on the running of FLEXPART model.

(15) Line 136. Why is $NO_2$ not constrained in the box model?

(16) Line 138-139: elaborate how photolysis rates are determined by scaling on-board DC-8 measurements.

(17) Line 142-145. Please include the reactions and their rates in the paper, at least in the supplement.

(18) Line 145-148: B&T (2009) parameterization is likely to overestimate $N_2O_5$ uptake coefficient. Also, did you assume $ClNO_2$ yield to be unity?

(19) Section 3.1. It would be interesting to see a comparison of the observed values in this study with those reported elsewhere.

(20) Line 164-165 and Line 176-181: If I understand correctly, (1) when $O_3$ is low, $Cl_2$ level is low but $ClNO_2$ could be high or low, so $ClNO_2$ does not have correlation with $Cl_2$: (2) when $O_3$ is elevated, $ClNO_2$ has a good correlation with $Cl_2$, which could be due to $ClNO_2$ uptake on acidic aerosol to form $Cl_2$ (Roberts et al., 2008). Then what would be the cause of the (1) situation? Why there is no production of $Cl_2$ from $ClNO_2$ uptake when $O_3$ is low, considering that the uptake of $ClNO_2$ on aerosol does not require the presence of $O_3$? Is the pH not low enough? If the $Cl_2$ is solely produced from the $ClNO_2$ uptake, the correlation between $ClNO_2$ and $Cl_2$ at night should be good as well. Is it possible that the $Cl_2$ is mainly formed by gas phase reactions which are initiated by the photolysis of $ClNO_2$ and the reaction of $HCl+OH$, both of which requires the presence of light? I would recommend the authors to apply box model sensitivity studies to understand the characteristics of $ClNO_2$ (and $Cl_2$).

(21) Line 174-176: I don't see why the effect of organic coating on $N_2O_5$ uptake is relevant to the correlation of $ClNO_2$ and $Cl_2$.

(22) Line 184-185: An explanation is needed for the calculation of 'source contribution of CO' using FLEXPART.

(23) Line 184-185: Do the authors mean that the $ClNO_2$ is highly correlated to the oceanic sources? Please elaborate.

(24) Line 188-189: Apart from the back trajectory analysis, there are other methods to determine/estimate the source of chloride. Please refer to the previous studies on $ClNO_2$ measurements. For example, In Line 172-174, the correlation of $ClNO_2$ and $SO_2$ is extremely low, so the coal-burning activity is not responsible for the chloride measured during the campaign. Any evidence of biomass burning, chemical signature (e.g. $K^+$) or the fire detected by the satellite? Any evidence of sea-salt aerosol, e.g. how is the correlation of chloride and sodium? What about waste burning?

(25) Line 189-190: What does "nitrate production was limited due to $O_3$ titration" mean?
(26) Figure 5(b), Wind directions should be added to see if the morning peak of $ClNO_2$ and the peak of $SO_2$ at 15:00 was related to wind direction change.

(27) Line 196-197: an explanation is needed for the choice of the days.

(28) Line 201-202: where is the information on boundary layer height coming from?

(29) Line 231-234: More detailed discussion of the vertical profiles is needed. How do you define residual layer? In line 210-212, the authors suggested that the $ClNO_2$ in the residual layer could be higher than those at ground surface based on a previous tower measurement in the same region (5 ppb of $N_2O_5$ at 360m a.s.l.). Need to reconcile these statements. Was the wind direction different at different altitudes? I suggest the author compare individual vertical profile with ground measurements to better reveal their relationship.

(30) Line 234 and Figure 6(b): The maximum $ClNO_2$ on May 25 appears to be ~200 ppt, while that on May 31 is ~750 ppt and that on June 10 is ~1250 ppt. As to those on other days, dozens samples showed more than 500 ppt even close to 1500ppt in the residual layer (between the nocturnal boundary layer and the boundary layer at midday, in the present study between 200/300m and 1000/2000m). In Figure 6(a), similar results could also be found near the TRF site, over 1000 ppt $ClNO_2$ concentrations were recorded at the height of ~500 m (yellowish). I suggest that the authors revisit the figures and the text.

(31) Line 243-251: the evidence for contribution of horizontal transport to the morning peak is not convincing. According to the authors, in half of the days, they measured second peak of $ClNO_2$ at 7-8 am, and in these days, the air masses came from various directions mostly from northwest to southwest and various distance (approximately 200 to 400 Km in 24h) (Fig 8). If the horizontal transport (advection) within the boundary layer is the cause of the second peak of $ClNO_2$, that means in all these directions and distance, there is a bulk of air mass with higher $ClNO_2$ that would constantly arrive at the measurement site at 7-8 am, not before nor after. This is physically not possible.

(32) Line 260-262: Wang et al. 2016 showed much larger contribution of $ClNO_2$ to ozone increase compared to 2% at TRF site, the latter is similar to 3% at Wangdu in Tham et al. (2016).

(33) Line 270-272: the comment on the result of Tham et al. (2016) and Wang et al. (2016) is an incorrect interpretation of their findings. The three factors (downward transport, horizontal transport and local chemical production) may impact different locations differently. Tham et al. presented evidence for downwind transport in the early morning hours at a polluted rural site in the North China plain. Wang et al. (2016) measured the high $ClNO_2$ plumes at a 974 m mountain-top site and suggested presence of high $ClNO_2$ in the upper boundary layer in south China. They did not say that downward transport should apply to all locations. One has to analyze his/her own case.

(34) Figure 7. What was this figure used for?

---

## Referee Comment (RC2) · Anonymous Referee #2 · 30 Dec 2018

**Review of Jeong et al.: Integration of Airborne and Ground Observations of Nitryl Chloride in the Seoul Metropolitan Area and the Implications on Regional Oxidation Capacity During KORUS-AQ 2016**

Jeong and co-authors present an analysis of $ClNO_2$ observations from the 2016 NASA KORUS-AQ campaign. $ClNO_2$, additional trace gases, and measurement of aerosol composition were collected aboard the NASA DC-8 aircraft as well as at two ground-based sites, which were the main focus of this study. The first site was located in the Seoul metropolitan area, while the second was located ~26 km southeast in a forested area. The same observation method was used for all $ClNO_2$ measurements, allowing for direct comparisons between sites. Ground-based observations revealed that $ClNO_2$ was elevated at night at both sites, with higher concentrations at the TRF forested site, which was physically removed from NO emissions, which reduce $O_3$ concentrations and calculated nitrate radical production rates. At both sites, there were periods where $ClNO_2$ persisted at high levels throughout the morning after sunrise. Using additional box-model simulations, the authors found that morning entrainment of $ClNO_2$ from the residual layer, nor morning $ClNO_2$ production could reconcile differences between the simulations and observations. Based on these results and backward air mass trajectories, the authors conclude that horizontal transport is the likely cause of elevated morning $ClNO_2$ concentrations. A final set of 24-hour simulations with and without $ClNO_2$ production, constrained to observations, revealed that chlorine radical initiated chemistry can increase net $O_3$ production in the morning by between 2 and 25%.

The authors have provided a succinct and novel analysis that merits publication after the following comments are addressed. First, this manuscript requires more details in the methods section about the box model set-up and the types of simulations that were conducted. Specific comments are provided below. Without additional details, it is difficult to fully assess the model results presented here. Second, the authors conclude that horizontal transport is the most likely cause of elevated surface-level morning $ClNO_2$, without providing sufficient evidence. The authors present three possible causes based on previous studies and find no evidence for the first two, and therefore conclude that the third, transport, must be the main source. While this may be the actual cause, the authors need to provide additional evidence for this conclusion as discussed in specific comments below. Other comments largely include additional suggested references, requested clarification of the calculated net $O_3$ and $NO_3$ radical production rates and aerosol surface area, and other minor and editorial comments.

**Major Comments:**

Instrument Section:
Ln 130 – The authors note that $ClNO_2$ is thermally converted to NO at 325 C in a CL instrument. The thermal conversion efficiency of $ClNO_2$, however, is estimated to be between 300 – 500 C, depending on the inlet and heater set-up and flow rate of a particular instrument (e.g. Thaler et al., 2011; Wild et al., 2014; Wooldridge et al., 2010). As the CL instrument was used to calibrate the CIMS $ClNO_2$ measurement, the authors should discuss how they accounted for (or quantified) the thermal conversion efficiency of $ClNO_2$.

Model Description Section:
More details are required in this section about the model set-up and the types of simulations that were conducted.

Ln 135 –State which meteorological parameters were used as constraints. Also clarify the type of simulations that were run and how the model was constrained with observations. For example, were simulations run for 24, 48, 76 hours? Were simulations constrained every 10 minutes, 1 hours, 6 hours, etc.? Were different simulations run and constrained to observations from both of the ground sites or was a single simulation run with a combination of the two?

Ln 149 – Clarify, did the authors apply a hygroscopic growth factor to the measured aerosol surface area? What was the size range of the aerosol particles that contributed to the measured surface area? Did aircraft vertical profiles show that the aerosol surface area was relatively constant with altitude? As there are no measurements of aerosol surface area on the ground, this source of uncertainty in the model should be discussed.

In addition, it would be helpful to put the $N_2O_5$ uptake coefficient into context. The authors could cite previous studies that derived uptake coefficients in Asia (e.g. Brown et al., 2016; Tham et al., 2016; Wang, Z. et al., 2017; Wang, X. et al., 2017; Wang, H. et al., 2017).

As this manuscript is primarily about $ClNO_2$ production, the authors should also state how the $ClNO_2$ yield was calculated in the model. In the event that the Bertram and Thornton parameterization was used, it is also important to note that this has been shown to be an over-prediction of field-derived yields (McDuffie et al., 2018a; Riedel et al., 2013; Ryder et al., 2015; Tham et al., 2018; Thornton et al., 2010; Wagner et al., 2013; Wang, Z. et al., 2017; Wang, X. et al., 2017).

Ln 150 – Add more details about the FLEXPART simulations. For example, add more details such as those in the Figure 3 caption.

Results and Discussion Section:
Ln 161 – The statement that $ClNO_2$ rapidly photolyzes near sunrise contradicts the authors later statements on line 213 that $ClNO_2$ persists after sunrise.

Paragraph starting on line 243 –Is there further information in the flight data to support the hypothesis that boundary layer transport is the main source of elevated surface-level $ClNO_2$?

For example, were $NO_x$ and $O_3$ observations to the west of the observation sites elevated relative to the east? My concern is that this section reads as though the third proposed possibility must be correct since there was no evidence for the first two possibilities. As written, there is not enough evidence in this section to support the third possibility that transport is the main source of surface-level $ClNO_2$.

Section 3.3.
It is unclear how $ClNO_2$ was used to constrain the model simulations. For example, was the model only initialized with observed $ClNO_2$ mixing ratios, or was the model constrained to observations throughout the morning? As the authors spend time in the previous section discussing the elevated morning $ClNO_2$, it seems important that model simulations emulate that observed behavior.

Conclusions:
The authors mention that stagnation events were associated with low $ClNO_2$ production. There was no discussion in the text, however, about the meteorology associated with these low $ClNO_2$ events. Moreover, past studies have shown that certain types of stagnation events can actually enhance $N_2O_5$ chemistry (e.g. Baasandorj et al., 2017).

**Minor Comments:**

Throughout text – change 'ppb' to 'ppbv' and 'ppt' to 'pptv'

line (ln) 16 – The authors reference the laboratory work of Roberts et al. (2008) showing $Cl_2$ production from $N_2O_5$ uptake on acidic, chloride-containing aerosol (R5). There is no known field evidence of this reaction occurring on ambient aerosol, even at low pH. For example, the recent study by McDuffie, E. E. et al. (2018) found a negative correlation between particle acidity and $Cl_2$(g) during the WINTER aircraft campaign, which is the opposite expected trend from this reaction. While this one study cannot confirm or deny the presence of the net $N_2O_5$ → $Cl_2$ reaction, the authors should note that there are uncertainties regarding the occurrence of this particular reaction on ambient aerosol.

Ln 18 – In addition to the decreased lifetime of $NO_3$ during the day, the short lifetime of $N_2O_5$ is largely due to its thermal instability during the day.

Ln 19 – define 'reactive chlorine'

Ln 35 – since there is discussion of the efficiency of $ClNO_2$ production, it might be helpful to change R4 to the following reaction to show the dependence of $ClNO_2$ on the $ClNO_2$ yield and $N_2O_5$ uptake coefficient.

$$N_2O_5(g) \xrightarrow{\gamma(N_2O_5), \ Cl^-(p)} (2 - \varphi) * HNO_3 + (\varphi) * ClNO_2 \tag{R4}$$

Paragraph starting on 37: There are many more observations than the few U.S. studies that are referenced. It is not necessary to cite all of these past studies, however, there are a growing number of observations in Asia, which should be referenced/discussed here as this is the focus area of this manuscript. Including, but not limited to: (Liu et al., 2017; Tham et al., 2018; Tham et al., 2016; Tham et al., 2014; Wang, X. et al., 2017; Wang et al., 2016; Wang, Z. et al., 2017; Wang et al., 2018; Wang et al., 2014; Yun, Hui et al., 2018). Many of these studies are discussed in the paragraph starting on line 66, but should still be referenced here when first discussing the history of $ClNO_2$ measurements.

Ln 57 – Add a reference for the model tendency to underestimate $ClNO_2$. For instance, if the simulated $ClNO_2$ yield is too large, an under-estimation in emissions would not necessarily lead to an underestimation of $ClNO_2$. The authors should additionally add a statement in this section about current uncertainties in the $ClNO_2$ yield as this adds uncertainty to model-predicted $ClNO_2$ as well. This topic was recently reviewed in McDuffie et al. (2018a) and references therein.

Ln 66 – Add a citation to Yun, H. et al. (2018), who report the largest concentrations of $ClNO_2$ to-date (8.3 ppbv!).

Ln 85 – As this study also includes a box model analysis, the authors should include a reference to this part of the analysis here.

Ln 100 – Were THS CIMS used at both ground sites and on the NASA DC-8? Please clarify. Much of the information in this paragraph could be moved to the SI since the iodide adduct chemistry with $ClNO_2$ and $Cl_2$ is not novel and has been used for many of the past measurements of these species.

Ln 107 – It is unclear which CIMS instrument is being discussed here, or whether all three instruments have the same configuration. Please clarify.

Ln 161 – The authors use LST here and throughout the text and KST in Figures 2 and 3. Please change for consistency.

Ln 167 – Riedel et al. (2012) have also extensively discussed the correlation between $Cl_2$ and $ClNO_2$ in data offshore of LA, which should be cited/discussed here. It is helpful to discuss similarities and differences with additional urban areas outside of Asia.

Ln 172 – 175 – Please clarify here how the presence of power plants and aerosol organics would impact the observed $ClNO_2/Cl_2$ correlation.

Ln 180 – As previously mentioned, it should also be noted here that there is currently no field-evidence for this reaction.

Ln 200 – Please add details about the calculation of the nitrate radical production rate, such as the rate coefficient that was used. It is also unclear here how a 'slow nitrate radical production rate' is consistent with a rapid drop in $ClNO_2$ at 22:00 LST.

Ln 201 – At night the boundary layer becomes vertically stratified, which results in a surface layer and residual layer.

Ln 211 – There have been multiple studies that have discussed the change in $N_2O_5$ chemistry with altitude. Many of these have been in the context of nitrate aerosol production. It would be good to reference some of this past work (in addition to Brown et al. (2017)) when discussing the change in $ClNO_2$ production with altitude. For example: (Baasandorj et al., 2017; Tham et al., 2016; Young et al., 2012; Yun, Hui et al., 2018)

Ln 235 – Figure 9 does not show the agreement between the observed and simulated $ClNO_2$ mixing ratios as indicated. The authors might have meant to reference Figure 7.

Ln 257 – Please define net $O_3$ production rate and explain how this was calculated from model simulations.

Table 1 – Clarify that chloride, nitrate, and sulfate are concentrations from particles < 1um in diameter.

Figure 1 – Increase the text size and resolution of the images in panel a. The insert is difficult to read.

Figure 2. The $NO_x$ observations should be averaged to the same time interval (i.e., 10 minutes) to allow for direct comparison between the sites.

Figure 3 – This timeseries makes $Cl_2$ appear as if it has a constant background of ~2 pptv at each site. Could the authors comment on this background and discuss whether it is real or an instrument artifact?

Figure 6 – Expand panel a to increase the visibility of the $ClNO_2$ data.

Figure 7 – Clarify that sunrise is at time zero. Clarify what the standard deviations are referring to. The authors should also mention why $ClNO_2$ at t=0 is 0 ppbv when there is actually $ClNO_2$ present at sunrise.

SUPPLEMENT
Section 1 –
As the AMS typically reports aerosol pH, the authors should compare their pH calculations to those from the AMS.

Figure S1 – Clarify whether this inlet configuration is for the ground-based CIMS or aircraft instrument. If for the ground-based CIMS, was it the same for both instruments?

Figure S6 – It is unclear how the May 5[th] profile of $ClNO_2$ relates to the diurnal average profile. The font size of the insert also needs to be increased.

**Editorial Comments:**
Throughout the text – check the consistency of 'aerosol' vs. 'aerosols' when using the plural form

line (ln) 5 – Change "in both sites" to "at both sites"

ln 6 – Either change "variation" to "variations" or "were" to "was"

ln 11 – Either change to "the net ozone production rate" or to "net ozone production rates by"

ln 15 – Change to ", which generated from an equilibrium reaction with…"

ln 16 – Change "In acidic aerosols" to "On acidic aerosol"

ln 20 – Change to "coal-fired power plants"

ln 22 – Change to "$N_2O_5$ aerosol uptake coefficient ($\gamma(N_2O_5)$), aerosol surface area, and $N_2O_5$ mean molecular speed, as well as the yield…"

ln 23 – The references provided are only a subset of the relevant literature. Change to (e.g., Thornton et al. …)

ln 29 – Change "level of ozone" to "ozone ($O_3$) level"

ln 37 – change to "The first ambient measurements of $ClNO_2$ were carried out by Osthoff et al. (2008), from a ship sampling along the southeastern U.S. coast in 2006"

ln 47 – change to "models"

ln 53 – Change 7 ppbv and 20% so that both $O_3$ and OH changes are either reported in percent or ppbv.

ln 71 – Change "correlated to" to "correlated with"

ln 78 – Change from "observation results during" to "observations from"

ln 82 – Change "were conducted" to "were collected"

ln 83 – Change to "and included airborne observations from the NASA DC-8…"

ln 84 – Change "ground" to "ground-based"

ln 95 – Change to "research flights when $ClNO_2$ was measured'.

Ln 105 – Change to "in various field conditions"

Ln 106 – Change to "at the two ground sites"

Ln 110 – Define slpm

Ln 111 – PTFE is used on line 103, prior to the definition here.

Ln 157 – Change to "During most nights…"

Ln 161 – Change to "At both sites,…"

Ln 164 – Change to "positive correlation with $Cl_2$…"

Ln 173 – Change to "the $ClNO_2$ measured at both the OP and TRF sites was weakly correlated…"

Ln 175 – Add "e.g.," before the McDuffie et al. (2018b) and Thornton et al. (2003) references as there have been many more studies that have looked at the organic influence on $N_2O_5$ uptake.

Ln 184 – Change to "at the ground sites was highly correlated with the origin of the air mass…"

Ln 185 – Change to "During the nights shaded in red in Figure 3 …"

Ln 186 – Change to "there was limited production of $ClNO_2$ at the surface." Also, change to "These periods corresponded to low contributions from air masses originating over the ocean and with limited particle chloride concentrations measured by the…"

Ln 190 – Change to "when nitrate production was limited at the surface due to $O_3$ reaction with NO, $ClNO_2$…"

Ln 198 – Change to "significant levels of $ClNO_2$ were sustained throughout the night during most of the…"

Ln 204 – Change to "near the surface"

Ln 207 – Change to "a Cavity Ringdown Spectrometer (CRDS) was installed on top of the Seoul tower in May -June that measured $N_2O_5$, $NO_x$, and $O_3$"

Ln 209 – Change to "the average nighttime $O_3$ mixing ratio was around 50 ppbv and $N_2O_5$ was observed most nights, with mixing ratios reaching up to 5 ppbv."

Ln 226 – Change to "At night…"

Ln 233 – Change to "However, the remaining flights observed an average of only 17 ± 56 pptv of $ClNO_2$ (black circles)."

Ln 255 – Change to "methods section."

Ln 270 – Place the reference to Tham and Wang in parentheses.

Figure 3 - Remove the extra ')' after $m^{-3}$. Change to "$ClNO_2$ and $Cl_2$ observations and results…"

Figure 5 – Change to "Diurnal variation of $ClNO_2$ and other parameters and trace gases calculated and measured during the campaign…"

Figure 9 – provide a label for the x-axis.

SUPPLEMENT
Ln 5 – Change to "model was run in the reverse mode"

Ln 16 – Change to "model was run similarly here, and it took…"

Ln 21 – Change to "pH and liquid water concentrations for sub-micron aerosol."

Figure S5. Change to "production rate of the nitrate radical."

Figure S6 – Change to "The insert in (b) is the…"

Figure S7 – Change the Figure 5 reference to Figure 6.

**References**

Baasandorj, M., Hoch, S. W., Bares, R., Lin, J. C., Brown, S. S., Millet, D. B., et al. (2017). Coupling between chemical and meteorological processes under persistent cold-air pool conditions: Evolution of wintertime $PM_{2.5}$ pollution events and $N_2O_5$ observations in Utah's Salt Lake Valley. *Environmental Science & Technology, 51*(11), 5941-5950. https://doi.org/10.1021/acs.est.6b06603

Brown, S. S., An, H.-J., Lee, M., Park, J.-H., Lee, S.-D., Fibiger, D., et al. (2017). Cavity enhanced spectroscopy for measurement of nitrogen oxides in the anthropocene: Results from the Seoul Tower during MAPS 2015. *Faraday Discussions*. https://doi.org/10.1039/C7FD00001D

Brown, S. S., Dubé, W. P., Tham, Y. J., Zha, Q., Xue, L., Poon, S., et al. (2016). Nighttime chemistry at a high altitude site above Hong Kong. *Journal of Geophysical Research: Atmospheres, 121*(5), 2457-2475. https://doi.org/10.1002/2015JD024566

Liu, X., Qu, H., Huey, L. G., Wang, Y., Sjostedt, S., Zeng, L., et al. (2017). High levels of daytime molecular chlorine and nitryl chloride at a rural site on the North China Plain. *Environmental Science & Technology, 51*(17), 9588-9595. https://doi.org/10.1021/acs.est.7b03039

McDuffie, Fibiger, D. L., Dube, W. P., Lopez-Hilfiker, F., Lee, B. H., Jaegle, L., et al. (2018a). $ClNO_2$ yields from aircraft measurements during the 2015 WINTER campaign and critical evaluation of the current parameterization. *Journal of Geophysical Research: Atmospheres - Submitted*.

McDuffie, Fibiger, D. L., Dubé, W. P., Lopez-Hilfiker, F., Lee, B. H., Thornton, J. A., et al. (2018b). Heterogeneous $N_2O_5$ uptake duirng winter: Aircraft measurements during the 2015 WINTER campaign and critical evaluation of current parameterizations. *Journal of Geophysical Research: Atmospheres, 123*(8), 4345-4372. https://doi.org/10.1002/2018JD028336

McDuffie, E. E., Fibiger, D. L., Dubé, W. P., Lopez Hilfiker, F., Lee, B. H., Jaeglé, L., et al. (2018). $ClNO_2$ yields from aircraft measurements during the 2015 WINTER campaign and critical evaluation of the current parameterization. *Journal of Geophysical Research: Atmospheres, 0*(0). https://doi.org/10.1029/2018JD029358

Riedel, T. P., Bertram, T. H., Crisp, T. A., Williams, E. J., Lerner, B. M., Vlasenko, A., et al. (2012). Nitryl chloride and molecular chlorine in the coastal marine boundary layer. *Environmental Science & Technology, 46*(19), 10463-10470. https://doi.org/10.1021/es204632r

Riedel, T. P., Wagner, N. L., Dubé, W. P., Middlebrook, A. M., Young, C. J., Öztürk, F., et al. (2013). Chlorine activation within urban or power plant plumes: Vertically resolved $ClNO_2$ and $Cl_2$ measurements from a tall tower in a polluted continental setting. *Journal of Geophysical Research: Atmospheres, 118*(15), 8702-8715. https://doi.org/10.1002/jgrd.50637

Roberts, J. M., Osthoff, H. D., Brown, S. S., & Ravishankara, A. R. (2008). $N_2O_5$ oxidizes chloride to $Cl_2$ in acidic atmospheric aerosol. *Science, 321*(5892), 1059. https://doi.org/10.1126/science.1158777

Ryder, O. S., Campbell, N. R., Shaloski, M., Al-Mashat, H., Nathanson, G. M., & Bertram, T. H. (2015). Role of organics in regulating $ClNO_2$ production at the air–sea interface. *The Journal of Physical Chemistry A, 119*(31), 8519-8526. https://doi.org/10.1021/jp5129673

Thaler, R. D., Mielke, L. H., & Osthoff, H. D. (2011). Quantification of Nitryl Chloride at Part Per Trillion Mixing Ratios by Thermal Dissociation Cavity Ring-Down Spectroscopy. *Analytical Chemistry, 83*(7), 2761-2766. https://doi.org/10.1021/ac200055z

Tham, Y. J., Yan, C., Xue, L., Zha, Q., Wang, X., & Wang, T. (2014). Presence of high nitryl chloride in Asian coastal environment and its impact on atmospheric photochemistry. *Chinese Science Bulletin, 59*(4), 356-359. https://doi.org/10.1007/s11434-013-0063-y

Tham, Y. J., Wang, Z., Li, Q., Wang, W., Wang, X., Lu, K., et al. (2018). Heterogeneous $N_2O_5$ uptake coefficient and production yield of $ClNO_2$ in polluted northern China: Roles of aerosol water content and chemical composition. *Atmospheric Chemistry and Physics Discussions, 2018*, 1-27. https://doi.org/10.5194/acp-2018-313

Tham, Y. J., Wang, Z., Li, Q., Yun, H., Wang, W., Wang, X., et al. (2016). Significant concentrations of nitryl chloride sustained in the morning: investigations of the causes and impacts on ozone production in a polluted region of northern China. *Atmospheric Chemistry and Physics, 16*(23), 14959-14977. https://doi.org/10.5194/acp-16-14959-2016

Thornton, J. A., Braban, C. F., & Abbatt, J. P. D. (2003). $N_2O_5$ hydrolysis on sub-micron organic aerosols: the effect of relative humidity, particle phase, and particle size. *Physical Chemistry Chemical Physics, 5*(20), 4593-4603. https://doi.org/10.1039/B307498F

Thornton, J. A., Kercher, J. P., Riedel, T. P., Wagner, N. L., Cozic, J., Holloway, J. S., et al. (2010). A large atomic chlorine source inferred from mid-continental reactive nitrogen chemistry. *Nature, 464*(7286), 271-4. https://doi.org/10.1038/nature08905

Wagner, N. L., Riedel, T. P., Young, C. J., Bahreini, R., Brock, C. A., Dubé, W. P., et al. (2013). $N_2O_5$ uptake coefficients and nocturnal $NO_2$ removal rates determined from ambient wintertime measurements. *Journal of Geophysical Research: Atmospheres, 118*(16), 9331-9350. https://doi.org/10.1002/jgrd.50653

Wang, H., Lu, K., Guo, S., Wu, Z., Shang, D., Tan, Z., et al. (2018). Efficient $N_2O_5$ Uptake and $NO_3$ Oxidation in the Outflow of Urban Beijing. *Atmospheric Chemistry and Physics Discussions, 2018*, 1-27. https://doi.org/10.5194/acp-2018-88

Wang, H., Lu, K., Chen, X., Zhu, Q., Chen, Q., Guo, S., et al. (2017). High $N_2O_5$ concentrations observed in urban Beijing: Implications of a large nitrate formation pathway. *Environmental Science & Technology Letters, 4*(10), 416-420. https://doi.org/10.1021/acs.estlett.7b00341

Wang, T., Tham, Y. J., Xue, L., Li, Q., Zha, Q., Wang, Z., et al. (2016). Observations of nitryl chloride and modeling its source and effect on ozone in the planetary boundary layer of southern China. *Journal of Geophysical Research: Atmospheres, 121*(5), 2476-2489. https://doi.org/10.1002/2015JD024556

Wang, X., Wang, T., Yan, C., Tham, Y. J., Xue, L., Xu, Z., & Zha, Q. (2014). Large daytime signals of $N_2O_5$ and $NO_3$ inferred at 62 amu in a TD-CIMS: chemical interference or a real atmospheric phenomenon? *Atmospheric Measurement Techniques, 7*(1), 1. https://doi.org/10.5194/amt-7-1-2014

Wang, X., Wang, H., Xue, L., Wang, T., Wang, L., Gu, R., et al. (2017). Observations of $N_2O_5$ and $ClNO_2$ at a polluted urban surface site in North China: High $N_2O_5$ uptake coefficients and low $ClNO_2$ product yields. *Atmospheric Environment, 156*, 125-134. https://doi.org/10.1016/j.atmosenv.2017.02.035

Wang, Z., Wang, W., Tham, Y. J., Li, Q., Wang, H., Wen, L., et al. (2017). Fast heterogeneous $N_2O_5$ uptake and $ClNO_2$ production in power plant and industrial plumes observed in the nocturnal residual layer over the North China Plain. *Atmospheric Chemistry and Physics, 17*(20), 12361-12378. https://doi.org/10.5194/acp-17-12361-2017

Wild, R. J., Edwards, P. M., Dube, W. P., Baumann, K., Edgerton, E. S., Quinn, P. K., et al. (2014). A measurement of total reactive nitrogen, $NO_y$, together with $NO_2$, NO, and $O_3$ via cavity ring-down spectroscopy. *Environmental Science & Technology, 48*(16), 9609-15. https://doi.org/10.1021/es501896w

Wooldridge, P. J., Perring, A. E., Bertram, T. H., Flocke, F. M., Roberts, J. M., Singh, H. B., et al. (2010). Total Peroxy Nitrates (ΣPNs) in the atmosphere: the Thermal Dissociation-Laser Induced Fluorescence (TD-LIF) technique and comparisons to speciated PAN measurements. *Atmos. Meas. Tech., 3*(3), 593-607. https://doi.org/10.5194/amt-3-593-2010

Young, C. J., Washenfelder, R. A., Roberts, J. M., Mielke, L. H., Osthoff, H. D., Tsai, C., et al. (2012). Vertically resolved measurements of nighttime radical reservoirs in Los Angeles and their contribution to the urban radical budget. *Environmental Science & Technology, 46*(20), 10965-73. https://doi.org/10.1021/es302206a

Yun, H., Wang, T., Wang, W., Tham, Y. J., Li, Q., Wang, Z., & Poon, S. C. N. (2018). Nighttime $NO_x$ loss and $ClNO_2$ formation in the residual layer of a polluted region: Insights from field measurements and an iterative box model. *Science of The Total Environment, 622-623*, 727-734. https://doi.org/https://doi.org/10.1016/j.scitotenv.2017.11.352

Yun, H., Wang, W., Wang, T., Xia, M., Yu, C., Wang, Z., et al. (2018). Nitrate formation from heterogeneous uptake of dinitrogen pentoxide during a severe winter haze in southern China. *Atmos. Chem. Phys., 18*(23), 17515-17527. https://doi.org/10.5194/acp-18-17515-2018

---

## Author Comment (AC1) · 17 Apr 2019

**Response to reviewer comments on "Integration of Airborne and Ground Observations of Nitryl Chloride in the Seoul Metropolitan Area and the Implications on Regional Oxidation Capacity During KORUS-AQ 2016"**

We would like to thank the reviewers for their careful consideration of the paper. The comments have been addressed and additional information has been added in the paper for clarification. The comments from the reviewers are in *black*, our responses are in *blue*, and parts that have been added to the manuscript are in ***bold blue***.

**Reviewer #1**

Jeong and co-authors present an analysis of ClNO2 observations from the 2016 NASA KORUS-AQ campaign. ClNO2, additional trace gases, and measurement of aerosol composition were collected aboard the NASA DC-8 aircraft as well as at two ground-based sites, which were the main focus of this study. The first site was located in the Seoul metropolitan area, while the second was located ~26 km southeast in a forested area. The same observation method was used for all ClNO2 measurements, allowing for direct comparisons between sites. Ground-based observations revealed that ClNO2 was elevated at night at both sites, with higher concentrations at the TRF forested site, which was physically removed from NO emissions, which reduce O3 concentrations and calculated nitrate radical production rates. At both sites, there were periods where ClNO2 persisted at high levels throughout the morning after sunrise. Using additional box-model simulations, the authors found that morning entrainment of ClNO2 from the residual layer, nor morning ClNO2 production could reconcile differences between the simulations and observations. Based on these results and backward air mass trajectories, the authors conclude that horizontal transport is the likely cause of elevated morning ClNO2 concentrations. A final set of 24-hour simulations with and without ClNO2 production, constrained to observations, revealed that chlorine radical initiated chemistry can increase net O3 production in the morning by between 2 and 25%.

The authors have provided a succinct and novel analysis that merits publication after the following comments are addressed. First, this manuscript requires more details in the methods section about the box model set-up and the types of simulations that were conducted. Specific comments are provided below. Without additional details, it is difficult to fully assess the model results presented here. Second, the authors conclude that horizontal transport is the most likely cause of elevated surface-level morning ClNO2, without providing sufficient evidence. The authors present three possible causes based on previous studies and find no evidence for the first two, and therefore conclude that the third, transport, must be the main source. While this may be the actual cause, the authors need to provide additional evidence for this conclusion as discussed in specific comments below. Other comments largely include additional suggested references, requested clarification of the calculated net O3 and NO3 radical production rates and aerosol surface area, and other minor and editorial comments.

**Major Comments:**

Instrument Section:

Ln 130 – The authors note that ClNO2 is thermally converted to NO at 325 C in a CL instrument. The thermal conversion efficiency of ClNO2, however, is estimated to be between 300 – 500 C, depending on the inlet and heater set-up and flow rate of a particular instrument (e.g. Thaler et al., 2011; Wild et al., 2014; Wooldridge et al., 2010). As the CL instrument was used to calibrate the CIMS ClNO2 measurement, the authors should discuss how they accounted for (or quantified) the thermal conversion efficiency of ClNO2.

The following has been added for clarity:

(Ln 147-149) **"$ClNO_2$ is detected as $NO_y$ in the CL through conversion to NO on the heated (325 ºC) molybdenum catalytic converter (Williams et al., 1998). The efficiency of the conversion was assumed to be unity."**

Model Description Section:

More details are required in this section about the model set-up and the types of simulations that were conducted.

Ln 135 –State which meteorological parameters were used as constraints. Also clarify the type of simulations that were run and how the model was constrained with observations. For example, were simulations run for 24, 48, 76 hours? Were simulations constrained every 10 minutes, 1 hours, 6 hours, etc.? Were different simulations run and constrained to observations from both of the ground sites or was a single simulation run with a combination of the two?

The 2.3 Modeling section was revised as suggested:

(Ln 155-156) **"Each step of the model was constrained with the averaged meteorology parameters (e.g., pressure, temperature, relative humidity) and trace gases observed…"**

Additional information of the model setup has been included in the supplementary:

(Supplementary Ln 31-34) **"Daytime steady state $Cl_2$ simulations (Figure 5) were constrained with meteorology and trace gas observations corresponding to each point of the data shown in Figure 4. The constrained parameters were kept constant throughout the 72 hours of integration time and the end points are shown in Figure 5. Steady state simulations of $ClNO_2$ production in the morning (Figure 8) were ran similarly by holding constant of all the measured parameters throughout the runs, except for $ClNO_2$."**

(Supplementary Ln 49-55) **"Impact of measured $ClNO_2$ on $O_3$ production (Figure 10) was explored by constraining the box model with diurnal variation of observations throughout each step. Constraining the model with the diurnal variation of measured $ClNO_2$, allowed the box model to capture its trend throughout the course of the day.**

**Since our purpose of the simulations were to explore the possible impact of ClNO$_2$ on O$_3$ production, NO$_2$ and O$_3$ were only constrained initially at the first step with observations and then calculated based on the chemistry embedded in the model. More specifically, the initial concentration of each following step was taken from the value in the previous step. The integration time in the model was 100 sec and the model was run for 1 day."**

Ln 149 – Clarify, did the authors apply a hygroscopic growth factor to the measured aerosol surface area? What was the size range of the aerosol particles that contributed to the measured surface area? Did aircraft vertical profiles show that the aerosol surface area was relatively constant with altitude? As there are no measurements of aerosol surface area on the ground, this source of uncertainty in the model should be discussed. In addition, it would be helpful to put the N2O5 uptake coefficient into context. The authors could cite previous studies that derived uptake coefficients in Asia (e.g. Brown et al., 2016; Tham et al., 2016; Wang, Z. et al., 2017; Wang, X. et al., 2017; Wang, H. et al., 2017). As this manuscript is primarily about ClNO2 production, the authors should also state how the ClNO2 yield was calculated in the model. In the event that the Bertram and Thornton parameterization was used, it is also important to note that this has been shown to be an over- prediction of field-derived yields (McDuffie et al., 2018a; Riedel et al., 2013; Ryder et al., 2015; Tham et al., 2018; Thornton et al., 2010; Wagner et al., 2013; Wang, Z. et al., 2017; Wang, X. et al., 2017).

The following has been added in the supplementary for clarity:

**(Supplementary Ln 34 - 48) "Heterogeneous reactions of gas-phase N$_2$O$_5$ (i.e., $N_2O_{5(g)}$ + $Cl^-_{(aq)}$ → $ClNO_{2(g)}$), ClONO$_2$ (i.e., $ClONO_{2(g)}$ + $Cl^-_{(aq)}$ + $H^+_{(aq)}$ → $Cl_{2(g)}$ + HNO$_3$), and HOCl (i.e., $HOCl_{(g)}$ + $Cl^-_{(aq)}$ + $H^+_{(aq)}$ → $Cl_{2(g)}$ + H$_2$O) were included in the model. For these heterogeneous reactions, a simple first-order reaction was assumed by accounting for γ, Φ, molecular speed of the gases, and surface area of aerosols. Hygroscopic growth factor was not considered in the model. $γ_{N2O5}$ was calculated from the Bertram and Thornton (2009) study using measured inorganic aerosol composition, temperature, and relative humidity and water content derived from the thermodynamic model Extended Aerosol Inorganics Model (E-AIMS, (Clegg et al., 1998; Friese and Ebel, 2010)). The average and median $γ_{N2O5}$ values during the whole campaign were both 0.017. This is in the lower range of what has been derived from previous field observations in Asia that ranges from a campaign average of 0.004 to 0.072 (Yun et al., 2018; Brown et al., 2016; Tham et al., 2016; Wang et al., 2017b, d, a, c). γ values of ClONO$_2$ and HOCl were set to 0.06 (Deiber et al., 2004; Hanson et al., 1994; Hanson and Ravishankara, 1994). The yields (Φ) of the three heterogeneous reactions were assumed to be 1, therefore the steady state simulations would be an upper-limit of Cl$_2$ or ClNO$_2$ production. Aerosol surface area was taken from airborne measurements of particle size distributions. An averaged value was used from data retrieved below 1 km over the SMA. The airborne data did not show a significant**

**vertical dependence within the daytime boundary layer. Based on this, an average of 78 ± 41 $\mu m^2 cm^{-3}$ were estimated for particle sizes between 10 nm and 5 $\mu m$."**

Ln 150 – Add more details about the FLEXPART simulations. For example, add more details such as those in the Figure 3 caption.

The following has been added for clarity:

(Ln 173 - 180) **"The FLEXible PARTi-cle dispersion model (FLEXPART v9.1, https://www.flexpart.eu) was used for the air mass source contribution (Figure 3) and backward trajectory analysis (Figure 9). The backward trajectories reported in our study were initialized 9:00 LST at TRF, following it 24 hours back in time. The trajectories were driven by the National Centers for Environmental Prediction (NCEP) Global Forecast System (GFS) with a 0.25 degree resolution. Influence of air mass originating from the ocean at TRF and OP was calculated every 6 hours following an air mass 5 days back in time. Meteorology was driven by WRF with a 5 km horizontal resolution. Since emissions of CO are very low in the ocean, and assumed to be inert in the model, it was used as a tracer for contribution of air originating from the ocean within a given air mass at each ground site."**

Results and Discussion Section:
Ln 161 – The statement that ClNO2 rapidly photolyzes near sunrise contradicts the authors later statements on line 213 that ClNO2 persists after sunrise.

Line 265 has been re-written as below:

**"At both sites, ClNO$_2$ levels started to increase or sustained after the first 2-3 hours of rapid net loss upon sunrise.. "**

Paragraph starting on line 243 –Is there further information in the flight data to support the hypothesis that boundary layer transport is the main source of elevated surface-level ClNO2? For example, were NOx and O3 observations to the west of the observation sites elevated relative to the east? My concern is that this section reads as though the third proposed possibility must be correct since there was no evidence for the first two possibilities. As written, there is not enough evidence in this section to support the third possibility that transport is the main source of surface-level ClNO2.

Following has been included in the discussion
(Ln 299-305) **"During KORUS, the DC-8 did not fly to the west of the SMA in the early morning. However, there are large point sources, such as petrochemical facilities and industries, and vehicular emissions to the west and south west of the SMA region. Sullivan et al. (2019) reported that this resulted in enhanced levels of O$_3$ in**

**receptor regions (i.e., Taehwa Research Forest) downwind when westerlies were prevalent. Therefore, favorable conditions such as high chloride content in aerosols from both anthropogenic and natural sources and high levels of $NO_x$-$O_3$ could have lead to significant levels of $ClNO_2$ to build up and transported to TRF before being completely photolyzed. During the campaign, influence of large biomass burning was negligible as reported in Tang et al. (2018, 2019)."**

Section 3.3.
It is unclear how ClNO2 was used to constrain the model simulations. For example, was the model only initialized with observed ClNO2 mixing ratios, or was the model constrained to observations throughout the morning? As the authors spend time in the previous section discussing the elevated morning ClNO2, it seems important that model simulations emulate that observed behavior.

A more detailed description of the box model runs have been added in the methods and supplementary. The following has been added for clarity:

(Supplementary Ln 49-51) **"Impact of measured $ClNO_2$ on $O_3$ production (Figure 10) was explored by constraining the box model with diurnal variation of observations throughout each step. Constraining the model with the diurnal variation of measured $ClNO_2$, allowed the box model to capture its trend throughout the course of the day"**

Conclusions:
The authors mention that stagnation events were associated with low ClNO2 production. There was no discussion in the text, however, about the meteorology associated with these low ClNO2 events. Moreover, past studies have shown that certain types of stagnation events can actually enhance N2O5 chemistry (e.g. Baasandorj et al., 2017).

Description on the meteorology during the campaign have been included in the Methods section for clarity:

(Ln 110-115) **"Meteorology during the observation period can be classified into dynamic (May 4th - 16th), stagnation (May 17th - 22nd), transport (May 25th - 31st), and blocking period as shown in Figure 3. During the stagnant period, high pressure system was persistent in the Korean peninsula resulting in local air masses to be more dominant within the SMA compared to the dynamic and transport (May 25nd - 31st) periods. Rex block patterns were observed during the blocking period (June 1st - 6th), which also resulted in more local influence."**

The following have been included in the discussion for further discussion on stagnation events and $N_2O_5$:

(Ln 236 -241) **"Stagnation events can be characterized by low wind speeds and increased atmospheric stability, possibly leading to enhanced levels of pollutants like $NO_x$. Previous studies have shown that these stagnant conditions can result in enhanced levels of $N_2O_5$ (Baasandorj et al., 2017) driven by high ozone and $NO_2$.**

> **However, ClNO$_2$ production was limited during stagnation events in this study. This is likely due to limited availability of chloride as shown in submicron particle measurements of aerosol mass spectrometer (AMS) at the ground site for OP and airborne over TRF (Figure 3)."**

**Minor Comments:**

Throughout text – change 'ppb' to 'ppbv' and 'ppt' to 'pptv'

Changes have been made throughout the text.

line (ln) 16 – The authors reference the laboratory work of Roberts et al. (2008) showing Cl2 production from N2O5 uptake on acidic, chloride-containing aerosol (R5). There is no known field evidence of this reaction occurring on ambient aerosol, even at low pH. For example, the recent study by McDuffie, E. E. et al. (2018) found a negative correlation between particle acidity and Cl2(g) during the WINTER aircraft campaign, which is the opposite expected trend from this reaction. While this one study cannot confirm or deny the presence of the net N2O5 → Cl2 reaction, the authors should note that there are uncertainties regarding the occurrence of this particular reaction on ambient aerosol.

The following has been added:
> (Ln 21) **"However, this reaction has yet to be proven in ambient conditions."**

Ln 18 – In addition to the decreased lifetime of NO3 during the day, the short lifetime of N2O5 is largely due to its thermal instability during the day.

The sentence has been revised as below:
> (Ln22-23) **"During the day, N$_2$O$_5$ exists at low levels due to its thermal instability (Malko and Troe, 1982) and the short lifetime of NO$_3$ (NO$_3$ < 5 s) from photolysis and reaction with NO (Wayne et al., 1991)."**

Ln 19 – define 'reactive chlorine'

The sentence has been revised as below:
> (Ln 23-24) **"Particulate Cl$^-$ and chlorine containing gas species can come from both natural sources such as sea salt…"**

Ln 35 – since there is discussion of the efficiency of ClNO2 production, it might be helpful to change R4 to the following reaction to show the dependence of ClNO2 on the ClNO2 yield and N2O5 uptake coefficient.

R4 has been replaced as below:

$$N_2O_{5(g)} \xrightarrow{\gamma(N_2O_5), Cl^-(aq)} (2-\phi) * HNO_{3(g)} + \phi * ClNO_{2(g)} \tag{R4}$$

Paragraph starting on 37: There are many more observations than the few U.S. studies that are referenced. It is not necessary to cite all of these past studies, however, there are a growing number of observations in Asia, which should be referenced/discussed here as this is the focus area of this manuscript. Including, but not limited to: (Liu et al., 2017; Tham et al., 2018; Tham et al., 2016; Tham et al., 2014; Wang, X. et al., 2017; Wang et al., 2016; Wang, Z. et al., 2017; Wang et al., 2018; Wang et al., 2014; Yun, Hui et al., 2018). Many of these studies are discussed in the paragraph starting on line 66, but should still be referenced here when first discussing the history of ClNO2 measurements.

The sentences have been rearranged and further revised to include the studies in Asia.

(Ln 55-64) **"More recently (in the past 5 years), increasing number of ClNO$_2$ observations have been conducted in Asia consistently showing significant levels of ClNO$_2$ present in the boundary layer (e.g., Tham et al. 2018, 2016; Wang et al. 2016, 2017c, 2014; Yun et al. 2018; Liu et al. 2017). ClNO$_2$ observations at semi-rural (Wangdu of Hebei province) and urban (Hong Kong, Jinan) regions in China have measured up to 2 ppbv and 776 pptv respectively. At the mountain top (957 m above sea level) in Hong Kong, up to 4.7 ppbv of ClNO$_2$ was reported. The high levels of ClNO$_2$ in these studies were mostly correlated with continental pollution in vicinity (e.g., power plant plumes, biomass burning). A recent study by Yun et al. (2018) reported the highest ClNO$_2$ (8.3 ppb), during a severe have event in a semi-rural site downwind of the Pearl River Delta in the winter. Overall, observations have shown that ClNO$_2$ is ubiquitous in the tropospheric boundary layer."**

Ln 57 – Add a reference for the model tendency to underestimate ClNO2. For instance, if the simulated ClNO2 yield is too large, an underestimation in emissions would not necessarily lead to an underestimation of ClNO2. The authors should additionally add a statement in this section about current uncertainties in the ClNO2 yield as this adds uncertainty to model-predicted ClNO2 as well. This topic was recently reviewed in McDuffie et al. (2018a) and references therein.

The sentences have been re-written as below :

Ln 66-68 **"Uncertainties in model simulated ClNO$_2$ can arise from limited emission inventories, low resolution of the grid, uncertainties in $\gamma_{N2O5}$ and $\Phi_{ClNO2}$ parameterization, complexity of the terrain, and meteorological conditions and these have been dealt in previous studies (e.g., Zhang et al. 2017; McDuffie et al. 2018b, a; Lowe et al. 2015; Sarwar et al. 2012, 2014; Sherwen et al. 2017)."**

Ln 66 – Add a citation to Yun, H. et al. (2018), who report the largest concentrations of ClNO2 to-date (8.3 ppbv!)

(Ln 61-63) "**A recent study by Yun et al. (2018) reported the highest CINO$_2$ (8.3 ppb), during a severe have event in a semi-rural site downwind of the Pearl River Delta in the winter.**"

Ln 85 – As this study also includes a box model analysis, the authors should include a reference to this part of the analysis here.

(Ln 97) **"We present observational and box model results to evaluated the impact of CINO2 towards regional air quality in SMA."**

Ln 100 – Were THS CIMS used at both ground sites and on the NASA DC-8? Please clarify. Much of the information in this paragraph could be moved to the SI since the iodide adduct chemistry with CINO2 and Cl2 is not novel and has been used for many of the past measurements of these species.

We agree with the reviewer that the iodide adduct chemistry is not novel and has been used in many previous studies. However, the information included in our method section is just a very brief summary of it, which we believe will help the readers for better understanding.

The same THS quadrupole CIMS were used at both ground sites and also on the DC-8. The related sentences have been revised as below for clarity:

(Ln 117-118) **"A THS Instruments LLC Chemical Ionization Mass Spectrometer (CIMS), using iodide (I$^-$) as the reagent ion was used for measuring Cl$_2$ and CINO$_2$ at the two ground supersites and on the NASA DC-8."**

Ln 107 – It is unclear which CIMS instrument is being discussed here, or whether all three instruments have the same configuration. Please clarify.

We used the same CIMS system on the two ground sites and airborne. However, the inlet configuration slightly varied as described in the manuscript. For clarification, the sentence has been revised as below:

(Ln 123-125) **"Therefore, the use of different types of inlets (e.g., the use of the donut), described above, at the two ground sites and on the DC-8 is not expected to be an issue for the quantitative comparisons in this study."**

Ln 161 – The authors use LST here and throughout the text and KST in Figures 2 and 3. Please change for consistency.

Ln 167 – Riedel et al. (2012) have also extensively discussed the correlation between Cl2 and ClNO2 in data offshore of LA, which should be cited/discussed here. It is helpful to discuss similarities and differences with additional urban areas outside of Asia.

Comparison with Riedel et al. (2012) has been included as below:

(Ln 190-192) **"Nighttime relationship between $ClNO_2$ and $Cl_2$ varied day by day and did not show a clear correlation. This implies that the sources of $Cl_2$ and $ClNO_2$ was not consistent at night. This is similar to Riedel et al. (2012), where they reported a wide range of correlation between $Cl_2$ and $ClNO_2$ off the coast of LA "**

 Ln 172 – 175 – Please clarify here how the presence of power plants and aerosol organics would impact the observed ClNO2/Cl2 correlation.

Ln 174-176 have been removed and the following sentence has been re-written as below:

(Ln 203-205) **"However, in this study, the $ClNO_2$ measured at both the OP and TRF sites was weakly correlated with $SO_2$ ($R^2 = 0.02$), which implies that the air masses that we sampled are not fresh emissions from power plants."**

Ln 180 – As previously mentioned, it should also be noted here that there is currently no field-evidence for this reaction.

The following has been added:

(Ln 210) "**Therefore, the efficiency of this reaction in ambient conditions requires further investigation.**"

Ln 200 – Please add details about the calculation of the nitrate radical production rate, such as the rate coefficient that was used. It is also unclear here how a 'slow nitrate radical production rate' is consistent with a rapid drop in ClNO2 at 22:00 LST.

The suggestions have been added as below:

(Ln 251-253) **"The trend was consistent with slower nitrate radical production rate ($d[NO_3]/dt = [NO_2]*[O_3]*k$, where $k= 3.52 \times 10^{-17}$ at 298 K, Atkinson et al., (2004)), as $O_3$ was titrated to zero by NO close to midnight."**

Ln 201 – At night the boundary layer becomes vertically stratified, which results in a surface layer and residual layer.

The sentence has been removed.

Ln 211 – There have been multiple studies that have discussed the change in N2O5 chemistry with altitude. Many of these have been in the context of nitrate aerosol production. It would be good to reference some of this past work (in addition to Brown et al. (2017)) when discussing the change in ClNO2 production with altitude. For example: (Baasandorj et al., 2017; Tham et al., 2016; Young et al., 2012; Yun, Hui et al., 2018)

The references have been added and discussed as below:

(Ln 254-259) **"However, significant levels of $N_2O_5$ and $ClNO_2$ could have been present in the upper part of the surface layer as shown in previous studies (Baasandorj et al., 2017; Young et al., 2012; Yun et al., 2018). According to Baasandorj et al. (2017), significant levels of $N_2O_5$ were observed aloft, within the boundary layer, while $O_3$ was completely titrated near the surface. On the other hand, airborne measurements at the LA basin (Young et al., 2012) showed a relatively uniform $ClNO_2$ profile throughout the boundary layer as $O_3$ did not change significantly within the measured altitude."**

Ln 235 – Figure 9 does not show the agreement between the observed and simulated ClNO2 mixing ratios as indicated. The authors might have meant to reference Figure 7.

The reference to figure 9 has been corrected to 8 in the text.

Ln 257 – Please define net O3 production rate and explain how this was calculated from model simulations.

The following has been included in the supplementary for clarity:

(supplementary Ln 55-57) **Net $O_3$ production rate was calculated in the box model as below, where f is the stoichiometric coefficient of $O_3$ and k is the rate constant corresponding to each reaction i. More details can be found in the supplements of Wolfe et al. (2016) :**

$$d[O_3]/dt = O_3\ production\ rate - O_3\ loss\ rate = \sum_{i=1}^{\#\ of\ reactions} f_i \times (product\ of\ reactants)_i \times k_i$$

Table 1 – Clarify that chloride, nitrate, and sulfate are concentrations from particles < 1um in diameter.
Changed
Figure 1 – Increase the text size and resolution of the images in panel a. The insert is difficult to read.
Changed
Figure 2. The NOx observations should be averaged to the same time interval (i.e., 10 minutes) to allow for direct comparison between the sites.

Changed

Figure 3 – This timeseries makes Cl2 appear as if it has a constant background of ~2 pptv at each site. Could the authors comment on this background and discuss whether it is real or an instrument artifact?

Figure 3 has been corrected to remove data points below detection limit. For $Cl_2$, the detection limit was around 2.9 ppt (2 sigma, over 30 min).

 Figure 6 – Expand panel a to increase the visibility of the ClNO2 data.
Changed

Figure 7 – Clarify that sunrise is at time zero. Clarify what the standard deviations are referring to. The authors should also mention why ClNO2 at t=0 is 0 ppbv when there is actually ClNO2 present at sunrise.

Figure 8 caption has been revised as below:
> **"Steady state $ClNO_2$, simulated from a box model constrained with airborne measurements (blue) and ground site data from TRF (red), when there was morning $ClNO_2$ peak in the morning. Averaged values of the model runs are shown here with standard deviations."**

More details of the box model simulations were added as below:
> **(Supplementary Ln 31-34) "Daytime steady state $Cl_2$ simulations (Figure 5) were constrained with meteorology and trace gas observations corresponding to each point of the data shown in Figure 4. The constrained parameters were kept constant throughout the 72 hours of integration time and the end points are shown in Figure 5. Steady state simulations of $ClNO_2$ production in the morning (Figure 8) were ran similarly by holding constant of all the measured parameters throughout the runs, except for $ClNO_2$."**

SUPPLEMENT
Section 1 –
As the AMS typically reports aerosol pH, the authors should compare their pH calculations to those from the AMS.
The AMS doesn't typically report aerosol pH. The AMS does provide ammonium balance but previous studies (Hennigan et al., ACP, 2015; Guo et al., JGR, 2016; etc.) have shown that this solely can't be used to determine aerosol pH. This is due to aerosol pH having a non-linear relationship with temperature and liquid water/RH (e.g., Guo et al., JGR, 2016; Guo et al., ACP, 2017; Song et al., ACP, 2018).  Similarly, subtracting anions from cations to get $H^+$ concentration wouldn't be plausible since $H^+$ depends on water and temperature. Therefore, the

only way to determine pH would be through running a thermodynamic model as have been done in our study.

Figure S1 – Clarify whether this inlet configuration is for the ground-based CIMS or aircraft instrument. If for the ground-based CIMS, was it the same for both instruments?

> (Supplementary Ln 5-8) **"During the KORUS-AQ 2016 field campaign a chemical ionization mass spectrometer (CIMS) was deployed to measure $Cl_2$ and $ClNO_2$. These systems were deployed at the Taehwa Research Forest (TRF), Olympic Park (OP), and on-board the NASA DC-8. The configuration of the inlet at the two ground sites is shown in Figure S1. The CIMS on the DC-8 had a similar configuration but without the heating inlet."**

Figure S6 – It is unclear how the May 5th profile of ClNO2 relates to the diurnal average profile. The font size of the insert also needs to be increased.

As mentioned in the Ln 270-271, in the morning of May 5th at TRF, maximum levels of $ClNO_2$ net production was required to reconcile the measurements. This insert was included as the 'max' case along with the 'average' case. The font size of the insert has been adjusted.

**Editorial Comments:**
Throughout the text – check the consistency of 'aerosol' vs. 'aerosols' when using the plural form
 changed
line (ln) 5 – Change "in both sites" to "at both sites"
 changed
ln 6 – Either change "variation" to "variations" or "were" to "was"
 changed
ln 11 – Either change to "the net ozone production rate" or to "net ozone production rates by"
changed
ln 15 – Change to ", which generated from an equilibrium reaction with…"
changed
ln 16 – Change "In acidic aerosols" to "On acidic aerosol"
changed
ln 20 – Change to "coal-fired power plants"
 changed
ln 22 – Change to "N2O5 aerosol uptake coefficient (g (N2O5)), aerosol surface area, and N2O5 mean molecular speed, as well as the yield…"
changed
ln 23 – The references provided are only a subset of the relevant literature. Change to (e.g., Thornton et al. …)
changed

ln 29 – Change "level of ozone" to "ozone (O3) level"

changed

ln 37 – change to "The first ambient measurements of ClNO2 were carried out by Osthoff et al. (2008), from a ship sampling along the southeastern U.S. coast in 2006"

changed

ln 47 – change to "models"

changed

ln 53 – Change 7 ppbv and 20% so that both O3 and OH changes are either reported in percent or ppbv.

Ln 78-79 has been changed as below:

**"The results showed that, compared to the simulations without ClNO$_2$ formation, monthly 8 h wintertime maximum O$_3$ and ☐OH increased up to 15 % and 20 %, respectively."**

ln 71 – Change "correlated to" to "correlated with"

changed

ln 78 – Change from "observation results during" to "observations from"

changed

ln 82 – Change "were conducted" to "were collected"

changed

ln 83 – Change to "and included airborne observations from the NASA DC-8…"

changed

ln 84 – Change "ground" to "ground-based"

changed

ln 95 – Change to "research flights when ClNO2 was measured'.

changed

Ln 105 – Change to "in various field conditions"

changed

Ln 106 – Change to "at the two ground sites"

changed

Ln 110 – Define slpm

revised

Ln 111 – PTFE is used on line 103, prior to the definition here.

Revised to define earlier

Ln 157 – Change to "During most nights…"

changed

Ln 161 – Change to "At both sites,…"

changed

Ln 164 – Change to "positive correlation with Cl2…"

changed

Ln 173 – Change to "the ClNO2 measured at both the OP and TRF sites was weakly correlated…"

changed

Ln 175 – Add "e.g.," before the McDuffie et al. (2018b) and Thornton et al. (2003) references as there have been many more studies that have looked at the organic influence on N2O5 uptake.

changed

Ln 184 – Change to "at the ground sites was highly correlated with the origin of the air mass…"

changed

Ln 185 – Change to "During the nights shaded in red in Figure 3 …"

changed

Ln 186 – Change to "there was limited production of ClNO2 at the surface." Also, change to "These periods corresponded to low contributions from air masses originating over the ocean and with limited particle chloride concentrations measured by the…"

changed

Ln 190 – Change to "when nitrate production was limited at the surface due to O3 reaction with NO, ClNO2…"

changed

Ln 198 – Change to "significant levels of ClNO2 were sustained throughout the night during most of the…"

changed

Ln 204 – Change to "near the surface"

changed

Ln 207 – Change to "a Cavity Ringdown Spectrometer (CRDS) was installed on top of the Seoul tower in May -June that measured N2O5, NOx, and O3"

changed

Ln 209 – Change to "the average nighttime O3 mixing ratio was around 50 ppbv and N2O5 was observed most nights, with mixing ratios reaching up to 5 ppbv."

changed

Ln 226 – Change to "At night…"

changed

Ln 233 – Change to "However, the remaining flights observed an average of only 17 ± 56 pptv of ClNO2 (black circles)."

changed

Ln 255 – Change to "methods section."

changed

Ln 270 – Place the reference to Tham and Wang in parentheses.

changed

Figure 3 - Remove the extra ')' after m-3. Change to "ClNO2 and Cl2 observations and results…"

 changed

Figure 5 – Change to "Diurnal variation of ClNO2 and other parameters and trace gases calculated and measured during the campaign…"

changed

Figure 9 – provide a label for the x-axis.

changed

SUPPLEMENT

Ln 5 – Change to "model was run in the reverse mode"

changed

Ln 16 – Change to "model was run similarly here, and it took…"

changed

Ln 21 – Change to "pH and liquid water concentrations for sub-micron aerosol."

changed

Figure S5. Change to "production rate of the nitrate radical."

changed

Figure S6 – Change to "The insert in (b) is the…"

changed

Figure S7 – Change the Figure 5 reference to Figure 6.

changed

**Reviewer #2**

Jeong et al. reported ground-based and airborne observations of ClNO2 in Korea during the Korean- United States-Air Quality (KORUS-AQ) 2016 field campaign. They analyzed the general characteristics, the sources and the effects of ClNO2 on ozone during their measurement campaign. The study contributes to the growing body of ClNO2 measurements around the world, and the content fits the scope of *Atmospheric chemistry and Physics.* The manuscript can be improved by adding more detailed description of measurements and more in-depth analysis/discussion on the elevated levels of ClNO2 in the morning. In addition, there are a number of places which need to be modified for clarity.

Specific comments

(1) Introduction: the manuscript gave a detailed review of measurements of ClNO2 conducted in North America and Europe, but didn't include some recent work in Asia (mostly in China). As the latter is more relevant to the present study due to proximity of the study regions, these studies should be reviewed (following the review of North America/Europe results). These studies are listed below:

· Wang, H., Lu, K., Guo, S., Wu, Z., Shang, D., Tan, Z., Wang, Y., Breton, M. L., Lou, S., and Tang, M.: Efficient N2O5 uptake and NO3 oxidation in the outflow of urban Beijing, Atmospheric Chemistry and Physics, 18, 9705-9721, 2018.
· Zhou, W., Zhao, J., Ouyang, B., Mehra, A., Xu, W., Wang, Y., Bannan, T. J., Worrall, S. D., Priestley, M., and Bacak, A.: Production of N2O5 and ClNO2 in summer in urban Beijing, China, Atmospheric Chemistry and Physics, 18, 11581-11597, 2018.

· Yun, H., Wang, W., Wang, T., Xia, M., Yu, C., Wang, Z., Poon, S. C. N., Yue, D., and Zhou, Y.: Nitrate formation from heterogeneous uptake of dinitrogen pentoxide during a severe winter haze in southern China, Atmospheric Chemistry and Physics, 2018, 23, 10.5194/acp-2018-698, 2018.

· Wang, Z., Wang, W., Tham, Y. J., Li, Q., Wang, H., Wen, L., Wang, X., and Wang, T.: Fast heterogeneous N2O5 uptake and ClNO2 production in power plant and industrial plumes observed in the nocturnal residual layer over the North China Plain, Atmospheric Chemistry and Physics, 17, 12361-12378, 2017.

· X. Wang, H. Wang, L. Xue, T. Wang, L. Wang, R. Gu, W. Wang, Y. J. Tham, Z. Wang, L. Yang, J. Chen and W. Wang, Observations of N2O5 and ClNO2 at a polluted urban surface site in North China: High N2O5 uptake coefficients and low ClNO2 product yields, Atmospheric Environment*, 156. 125-134. 2017.

More studies in Asia have been added as suggested:

(Ln 55-64) **"More recently (in the past 5 years), increasing number of ClNO$_2$ observations have been conducted in Asia consistently showing significant levels of ClNO$_2$ present in the boundary layer (e.g., Tham et al. 2018, 2016; Wang et al. 2016, 2017c, 2014; Yun et al. 2018; Liu et al. 2017). ClNO$_2$ observations at semi-rural (Wangdu of Hebei province) and urban (Hong Kong, Jinan) regions in China have measured up to 2 ppbv and 776 pptv respectively. At the mountain top (957 m above sea level) in Hong Kong, up to 4.7 ppbv of ClNO$_2$ was reported. The high levels of ClNO$_2$ in these studies were mostly correlated with continental pollution in vicinity (e.g., power plant plumes, biomass burning). A recent study by Yun et al. (2018) reported the highest ClNO$_2$ (8.3 ppb), during a severe have event in a semi-rural site downwind of the Pearl River Delta in the winter. Overall, observations have shown that ClNO$_2$ is ubiquitous in the tropospheric boundary layer."**

(2) Line 14: The lifetime of ClNO2 depends on the photolysis rate which varies among regions, seasons, weather conditions, etc. It's better to specify the condition in which the lifetime of ClNO2 is ~ 30 min and provide a reference.

The suggested information has been added as below:

(Ln 15-17) **"Nitryl chloride (ClNO$_2$) is a night time radical reservoir that generates chlorine radicals (Cl□) upon sunrise (R1), with a lifetime ($\tau_{ClNO2}$) of ~ 30 minutes at midday in the northern hemisphere mid-latitude summer, under clear sky conditions ($J_{ClNO2}$=5.47×10$^{-04}$ s$^{-1}$, (Madronich and Flocke, 1998))."**

(3) Line 17: "resulting from an enhanced ClNO2 uptake coefficient of up to 3 orders of magnitude" is not clear. Is the uptake in the condition of pH<2 "3 orders of magnitude" higher than that in the condition of pH=7? Please elaborate.

**(Ln 19-21) "In acidic aerosols (~pH 1.8), uptake of $N_2O_5$(g) can also produce gas-phase chlorine ($Cl_2$, R5), resulting from enhanced $ClNO_2$ uptake coefficient of up to 3 - 4 orders of magnitude higher than neutral pH (Roberts et al., 2008)."**

(4) Line 21: on source of chloride, Fu et al. (2018) presents a fine chloride emission inventory for China. This can be a good reference here.
· Fu, X., Wang, T., Wang, S., Zhang, L., Cai, S., Xing, J., and Hao, J.: Anthropogenic Emissions of Hydrogen Chloride and Fine Particulate Chloride in China, Environmental Science & Technology, 52, 1644-1654, 2018.

The suggested reference has been added in Ln 26.

(5) Line 25: Is the heterogeneous uptake of ClNO2 on acidic particle (Roberts et al., 2008) taken into account in the calculation of the lifetime (30h) during the nighttime?

No, it was not taken into account since this specific reaction on enhanced uptake coefficient of $ClNO_2$ on acidic particles has been reported in a laboratory study by Roberts et al. (2008) and uncertainties do remain in terms of its efficiency in ambient aerosols.

(6) Line 45: These previous studies showed that ClNO2 is ubiquitous at surface, within the boundary layer or in the lower troposphere around the world. No evidence has shown that the ClNO2 is ubiquitous in the troposphere which could reach >10 Km above the sea level in mid-latitude region.

Re-written as below:

(Ln 63) "**Overall, observations have shown that $ClNO_2$ is ubiquitous in the tropospheric boundary layer.**"

(7) Line 55-57: another modeling study Li et al. (2017) assessed the ozone impact of ClNO2 in East Asia including Korea.
· Zhang, L., Li, Q., Wang, T., Ahmadov, R., Zhang, Q., Li, M., and Lv, M.: Combined impacts of nitrous acid and nitryl chloride on lower-tropospheric ozone: new module development in
WRF-Chem and application to China, Atmospheric Chemistry and Physics, 17, 9733-9750,2017.

The reference has been added as below:

(Ln 84-85) **"Another modeling study of WRF-CHEM embedded with an updated chlorine chemistry, simulated 3-6 % of surface $O_3$ increase in the North China Plain and Yangtze River Delta during the summer (Zhang et al., 2017)."**

(8) Line 58-65: the discussion on model resolution does not seem to be relevant to the present study.

The discussion on model resolution was included to present the discrepancies remaining between observation and model results. These discrepancies lead to the importance of more observations being required in different chemical regimes.

(9) Line 64-65: I assume that the authors are referring to the simulations by Sherwen et al. (2017) that underestimated the ClNO2 by 7 times. Please confirm.

The reference has been added in the text as below:
> (Ln 73-74) **"Compared to observations, the simulations underestimated the ClNO$_2$ maxima levels by ~ 7 times in inland areas (Sherwen et al., 2017)."**

(10) Line 73-78 discusses importance of chorine source in coastal cities and gives the reader an impression that it is only important near coast. But measurement data have shown it is present far inland as shown in Thronton et al. (2010). A recent compilation of PM2.5 data also shows high levels of chloride are present in inland regions of China (Yang et al., STOTEN, 2017). The relevant sentence should be modified.

· Yang, X., Wang, T., Xia, M., Gao, X., Li, Q., Zhang, N., Gao, Y., Lee, S., Wang, X., and Xue, L.: Abundance and origin of fine particulate chloride in continental China, Science of The Total Environment, 624, 1041-1051,2018.

As mentioned in Ln 52 ("Recent studies show that high levels of ClNO$_2$ are also present in mid-continental regions.") and in the following sentences, we give examples of field observations affected by continental pollution.

(11) Section 2.1. It would help to include a brief description of the meteorology during the campaign.

The following has been added as suggested:
> (Ln 110-115) **"Meteorology during the observation period can be classified into dynamic (May 4th - 16th), stagnation (May 17th - 22nd), transport (May 25th - 31st), and blocking period as shown in Figure 3. During the stagnant period, high pressure system was persistent in the Korean peninsula resulting in local air masses to be more dominant within the SMA compared to the dynamic and transport (May 25nd - 31st) periods. Rex block patterns were observed during the blocking period (June 1st - 6th), which also resulted in more local influence."**

(12) Section 2.2 on CIMS and calibration: more detailed information is needed. What is the length of the sample line? Was it washed or replaced regularly in order to reduce the loss of N2O5? How frequent was the calibration? Was change of CIMS sensitivity to relative humidity taken into account in data reduction, and how? Could measurement of Cl2 with a Q-CIMS subject to interference? In line 115, 'the natural abundance of Cl2 and ClNO2 isotopes are approximately 9:6:1", what do the authors mean by this statement? No figure was shown and it is not clear how the isotopic ratios behave.

The sampling lines were cleaned regularly and $ClNO_2$ calibration was carried out before and after the campaign and with additional calibrations during the campaign for $Cl_2$. $N_2O_5$ data are not presented in this study. The sensitivity of the CIMS was taken into account by normalizing the signals to the water cluster (amu 147 $I\cdot H_2^{18}O$). In the manuscript, it says "The natural abundance of $Cl_2$ and $ClNO_2$ isotopes are approximately 9:6:1 and 3:1 respectively." which means natural $Cl_2$ has an isotopic ratio of 9:6:1 and $ClNO_2$ has 3:1. The $I\cdot^{37}Cl^{37}Cl$ (amu 201) was subject to interference with elevated signals that didn't match the isotopic ratio, therefore wasn't considered in our $Cl_2$ data.

(13) Line 131: How was HONO measured?

HONO is measured as $NO_y$ with the chemiluminescence instrument (Thermo scientific 42i). This is done by conversion of HONO to NO on the oxidized molybdenum surface. The generation of $ClNO_2$ was monitored for a couple of days to ensure minimum production of HONO (Thaler et al., 2011). The data that showed a stable signal of $ClNO_2$ after 2-3 days of the experiment were considered in the calibration.

(14) Section 2.3. Add description on the calculation of the impact of ClNO2 on O3 production rate and on the running of FLEXPART model.

The following has been added in the supplementary:

(supplementary Ln 55-57) **Net $O_3$ production rate was calculated in the box model as below, where f is the stoichiometric coefficient of $O_3$ and k is the rate constant corresponding to each reaction i. More details can be found in the supplements of Wolfe et al. (2016) :**

*d[O$_3$]/dt = O$_3$ production rate - O$_3$ loss rate =* $\sum_{i=1}^{\text{\# of reactions}} fi \times (product\ of\ reactants)i \times ki$

The following has been added:

(Ln 173 - 180) **"The FLEXible PARTi-cle dispersion model (FLEXPART v9.1, https://www.flexpart.eu) was used for the air mass source contribution (Figure 3) and backward trajectory analysis (Figure 9). The backward trajectories reported in our**

**study were initialized 9:00 LST at TRF, following it 24 hours back in time. The trajectories were driven by the National Centers for Environmental Prediction (NCEP) Global Forecast System (GFS) with a 0.25 degree resolution. Influence of air mass originating from the ocean at TRF and OP was calculated every 6 hours following an air mass 5 days back in time. Meteorology was driven by WRF with a 5 km horizontal resolution. Since emissions of CO are very low in the ocean, and assumed to be inert in the model, it was used as a tracer for contribution of air originating from the ocean within a given air mass at each ground site."**

(15) Line 136. Why is NO2 not constrained in the box model?

(Supplementary Ln 49-54) **"Impact of measured $ClNO_2$ on $O_3$ production (Figure 10) was explored by constraining the box model with diurnal variation of observations throughout each step. Constraining the model with the diurnal variation of measured $ClNO_2$, allowed the box model to capture its trend throughout the course of the day. Since our purpose of the simulations were to explore the possible impact of $ClNO_2$ on $O_3$ production, $NO_2$ and $O_3$ were only constrained initially at the first step with observations and then calculated based on the chemistry embedded in the model. More specifically, the initial concentration of each following step was taken from the value in the previous step.."**

(16) Line 138-139: elaborate how photolysis rates are determined by scaling on-board DC-8 measurements.

(Ln 158 - 166) **"Photolysis rate constants were derived through the hybrid method (Wolfe et al., 2016) in the F0AM box model. This method uses clear sky solar spectra from the tropospheric ultraviolet and visible radiation model (TUV v 5.2) and cross sections and quantum yields suggested by IUPAC. To capture the effects of pollution on photolysis rates, the ratio of the measured $J_{NO2}$ to the F0AM modeled $J_{NO2}$ was calculated. This ratio was then applied to other photolysis rate constants calculated in the model. Measured $J_{NO2}$ was taken from the DC-8 actinic flux measurements (Charged-coupled device Actinic Flux Spectroradiometer; CAFS) when flying near SMA at altitudes under 1 km. A diurnal cycle was applied to the DC-8 measurement to determine j-values at other times of day. Photolysis rate constants of $ClNO_2$, $Cl_2$, and $ClONO_2$ were not present in the F0AM model and therefore taken directly from the DC-8 measurements."**

(17) Line 142-145. Please include the reactions and their rates in the paper, at least in the supplement.

The following sentences have been revised to add related references and sources of the reactions embedded in the model:

(Ln 166-173) **"The Master Chemical Mechanism v3.3.1 (MCM) was taken from http://mcm.leeds.ac.uk/MCM and embedded in the box model. MCM v3.3.1 has a detailed gas photochemistry (i.e., 5832 species and 17224 reactions), including the oxidation of $CH_4$ and 142 non-methane primary emitted VOCs (Jenkin et al., 2015). Since MCM v3.3.1 only includes Cl· reactions with alkane species, additional chlorine chemistry was embedded in the model, similar to what Riedel et al. (2014) reported. This was done by including multiple Cl· precursors (e.g., $Cl_2$, $ClNO_2$, HCl, $ClONO_2$, HOCl) and Cl· reactions with non-alkane VOCs, such as alkene, alcohol, aromatics, alkynes, ketones, organic acids and nitrates. All the reactions embedded in the model can be found in the supplementary of Riedel et al. (2014) and Wolfe et al. (2016)."**

(18) Line 145-148: B&T (2009) parameterization is likely to overestimate N2O5 uptake coefficient. Also, did you assume ClNO2 yield to be unity?

The purpose of the box model simulation of $ClNO_2$ was to explore whether the runs can reproduce the levels that we observed in the morning. Based on the simulations, the model highly underestimated the measured levels by more than 60 times. Therefore, an overestimation of the $N_2O_5$ uptake coefficient or the yield of $ClNO_2$ would not change our conclusion.

The following has been added in the supplementary:

(Ln 44-45) **"The yields ($\Phi$) of the three heterogeneous reactions were assumed to be 1, therefore the steady state simulations would be an upper-limit of $Cl_2$ or $ClNO_2$ production."**

(19) Section 3.1. It would be interesting to see a comparison of the observed values in this study with those reported elsewhere.

$ClNO_2$ observations from previous studies have been dealt in the introduction.

(20) Line 164-165 and Line 176-181: If I understand correctly, (1) when O3 is low, Cl2 level is low but ClNO2 could be high or low, so ClNO2 does not have correlation with Cl2: (2) when O3 is elevated, ClNO2 has a good correlation with Cl2, which could be due to ClNO2 uptake on acidic aerosol to form Cl2 (Roberts et al., 2008). Then what would be the cause of the (1) situation? Why there is no production of Cl2 from ClNO2 uptake when O3 is low, considering that the uptake of ClNO2 on aerosol does not require the presence of O3? Is the pH not low enough? If the Cl2 is solely produced from the ClNO2 uptake, the correlation between ClNO2 and Cl2 at night should be good as well. Is it possible that the Cl2 is mainly formed by gas phase reactions which are initiated by the photolysis of ClNO2 and the reaction of HCl+OH, both of which requires the presence of light? I would recommend the authors to apply box model sensitivity studies to understand the characteristics of ClNO2 (and Cl2).

Box model analysis on the daytime $Cl_2$ production has been added in the study as below:

(Ln 210 - 230) **"Another possibility is the autocatalytic production of $Cl_2$ from heterogeneous reactions of gas-phase $ClONO_2$ (i.e., $ClONO_{2(g)}$ + $Cl^-_{(aq)}$ + $H^+_{(aq)} \rightarrow Cl_{2(g)}$ + $HNO_3$, (Gebel and Finlayson-Pitts, 2001; Deiber et al., 2004)) and HOCl (i.e., $HOCl_{(g)}$ + $Cl^-_{(aq)}$ + $H^+_{(aq)}$ $\rightarrow$ $Cl_{2(g)}$ + $H_2O$, (Vogt et al., 1996)) on particles. These reactions are also favored as particle acidity increases. In order to further investigate its possibility, daytime $Cl_2$ was simulated by constraining the box model with measurements of $ClNO_2$ and other trace gases corresponding to each data point in Figure 4. Based on the availability of parameters, we were able to simulate 1680 and 1229 runs for the OP and TRF, respectively. This corresponds to more than 96 % of the daytime data points shown in Figure 4. $ClONO_2$ and HOCl were set to 0.06 (Deiber et al., 2004; Hanson et al., 1994; Hanson and Ravishankara, 1994), which is an upper-limit of previous laboratory studies, and the yields were assumed to be unity. HCl generation from hydrogen abstraction of VOCs by Cl□ were included in the mechanisms used in the model runs. The end points of the 72 hour simulation results are presented in Figure 5. As shown in the Figure, the box model simulations were able to reproduce the positive correlation between $Cl_2$ and $ClNO_2$. Moreover, modeled $Cl_2$ was suppressed in low $O_3$ conditions, which corresponds to the observations. This can be explained by Cl□ reacting with $O_3$, producing ClO□, leading to gas-phase $ClONO_2$ and HOCl. These can react on acidic aerosols to generate $Cl_2$. Sources of Cl□ could be from photo-labile gas-phase chlorine compounds (e.g., $Cl_2$, $ClNO_2$, $ClONO_2$, HOCl) or oxidation of gas-phase HCl by OH. Although the reaction between HCl and OH is relative slow (k = 7.86 ✕ $10^{-13}$ $cm^3$ $molecule^{-1}s^{-1}$ at 298K, (Atkinson et al., 2007)), it has been reported to be a significant source of Cl in the daytime (Riedel et al., 2012). A sensitivity test was carried out by comparing modeled $Cl_2$ between runs with and without HCl production from oxidation of VOCs by Cl□ (Figure S4 c,d). The results show that production of $Cl_2$ was suppressed by 40 - 70 % when HCl was not generated in the model. This significant contribution of gas-phase HCl as a Cl source, should be an upper-limit as the deposition of HCl was not considered in the model. Nonetheless, our analysis leads us to conclude that the mechanisms we have explored could be the main contributions of the daytime $Cl_2$ production during KORUS-AQ."**

[Figure]

**Figure 5.** Correlation between box model simulated daytime (11:00 - 18:00 local time) $Cl_2$ and measured $ClNO_2$ at (a) OP and (b)TRF, color coded with measured $O_3$.

[Figure]

**Figure S4.** Correlation between measured $Cl_2$ and modeled $Cl_2$ at (a) OP and (b) TRF. Sensitivity tests of HCl were carried out (c and d) by switching off HCl production from chlorine radicals reacting with VOCs.

(21) Line 174-176: I don't see why the effect of organic coating on N2O5 uptake is relevant to the correlation of ClNO2 and Cl2.

Ln 174-176 have been removed

(22) Line 184-185: An explanation is needed for the calculation of 'source contribution of CO' using FLEXPART.

The following has been added for clarity:

(Ln 173 - 180) **"The FLEXible PARTi-cle dispersion model (FLEXPART v9.1, https://www.flexpart.eu) was used for the air mass source contribution (Figure 3) and backward trajectory analysis (Figure 9)..... Influence of air mass originating from the ocean at TRF and OP was calculated every 6 hours following an air mass 5 days back in time. Meteorology was driven by WRF with a 5 km horizontal resolution. Since emissions of CO are very low in the ocean, and assumed to be inert in the model, it was used as a tracer for contribution of air originating from the ocean within a given air mass at each ground site."**

(23) Line 184-185: Do the authors mean that the ClNO2 is highly correlated to the oceanic sources?Please elaborate.

(Ln 241 - 242) **"Whether the chloride is from the ocean or anthropogenic emissions is uncertain since large point sources, such as power plants or petrochemical facilities, are also present along the west coast of the SMA"**

(24) Line 188-189: Apart from the back trajectory analysis, there are other methods to determine/estimate the source of chloride. Please refer to the previous studies on ClNO2 measurements. For example, In Line 172-174, the correlation of ClNO2 and SO2 is extremely low, so the coal-burning activity is not responsible for the chloride measured during the campaign. Any evidence of biomass burning, chemical signature (e.g. K+) or the fire detected by the satellite? Any evidence of sea-salt aerosol, e.g. how is the correlation of chloride and sodium? What about waste burning?

At the Taehwa Research Forest, there were no aerosol composition measurements carried out during the campaign. FLEXPART back trajectory analysis initialized at 9:00 LST shows an overall westerlies when there was a second morning peak of $ClNO_2$. The following has been added in the discussion:

(Ln 299-305) **"During KORUS, the DC-8 did not fly to the west of the SMA in the early morning. However, there are large point sources, such as petrochemical facilities and industries, and vehicular emissions to the west and south west of the SMA region. Sullivan et al. (2019) reported that this resulted in enhanced levels of $O_3$ in receptor regions (i.e., Taehwa Research Forest) downwind when westerlies were prevalent. Therefore, favorable conditions such as high chloride content in aerosols from both anthropogenic and natural sources and high levels of $NO_x$-$O_3$ could have**

**lead to significant levels of ClNO$_2$ to build up and transported to TRF before being completely photolyzed. During the campaign, influence of large biomass burning was negligible as reported in Tang et al. (2018, 2019)."**

(25) Line 189-190: What does "nitrate production was limited due to O3 titration" mean?

The sentence has been removed.

(26) Figure 5(b), Wind directions should be added to see if the morning peak of ClNO2 and the peak of SO2 at 15:00 was related to wind direction change.

We did not observe any changes in the wind direction when the morning peak of ClNO$_2$ was happening.

(27) Line 196-197: an explanation is needed for the choice of the days.

The following sentence has been added and the explanation on the profiles are mentioned in the following sentences.

**(Ln 246) "...The description on these profiles are further explained in the following sentences."**

(28) Line 201-202: where is the information on boundary layer height coming from?

The "200 - 300 m" boundary layer height in Line 201-202 was referring to a typical nocturnal boundary layer height during the night, which is not specific for our study. In our study, we used vertical profiles of potential temperature to get the boundary layer heights (Figure 6). Since the DC-8 only flew during the daytime (mostly between 8 am to 4 pm local time), we do not have nighttime measurements of vertical potential temperature profiles during KORUS. Line 201-202 has been removed.

(29) Line 231-234: More detailed discussion of the vertical profiles is needed. How do you define residual layer? In line 210-212, the authors suggested that the ClNO2 in the residual layer could be higher than those at ground surface based on a previous tower measurement in the same region (5 ppb of N2O5 at 360m a.s.l.). Need to reconcile these statements. Was the wind direction different at different altitudes? I suggest the author compare individual vertical profile with ground measurements to better reveal their relationship.

In our study, the boundary layer height was determined based on the vertical profile of the airborne potential temperature measurements. This was averaged during the same timeframe and region as the ClNO$_2$ morning time data shown in Figure 6 (i.e., 8:00 - 8:30 local time). In

terms of our statement in ln 210-212, we were suggesting that it is possible that enhanced $ClNO_2$ could have been present in the upper surface layer not the residual layer.

(30) Line 234 and Figure 6(b): The maximum ClNO2 on May 25 appears to be ~200 ppt, while that on May 31 is ~750 ppt and that on June 10 is ~1250 ppt. As to those on other days, dozens samples showed more than 500 ppt even close to 1500ppt in the residual layer (between the nocturnal boundary layer and the boundary layer at midday, in the present study between 200/300m and 1000/2000m). In Figure 6(a), similar results could also be found near the TRF site, over 1000 ppt ClNO2 concentrations were recorded at the height of ~500 m (yellowish). I suggest that the authors revisit the figures and the text.

As in comment #28 and #29, we estimated the nocturnal boundary layer based on the vertical profile of the potential temperature measured on-board the DC-8 in the morning. Based on this measurement, at 8:00-8:30 am local time, the boundary layer height is estimated to be 500 - 600 m. Between this height and above, we did not observe significant $ClNO_2$ levels that could reconcile the ground observations except for the days with colored green square (May 31st), purple triangle (May 25th), and green square (May 31st).

(31) Line 243-251: the evidence for contribution of horizontal transport to the morning peak is not convincing. According to the authors, in half of the days, they measured second peak of ClNO2 at 7-8 am, and in these days, the air masses came from various directions mostly from northwest to southwest and various distance (approximately 200 to 400 Km in 24h) (Fig 8). If the horizontal transport (advection) within the boundary layer is the cause of the second peak of ClNO2, that means in all these directions and distance, there is a bulk of air mass with higher ClNO2 that would constantly arrive at the measurement site at 7-8 am, not before nor after. This is physically not possible.

Following has been added for further discussion:

> (Ln 299-305) **"During KORUS, the DC-8 did not fly to the west of the SMA in the early morning. However, there are large point sources, such as petrochemical facilities and industries, and vehicular emissions to the west and south west of the SMA region. Sullivan et al. (2019) reported that this resulted in enhanced levels of $O_3$ in receptor regions (i.e., Taehwa Research Forest) downwind when westerlies were prevalent. Therefore, favorable conditions such as high chloride content in aerosols from both anthropogenic and natural sources and high levels of $NO_x$-$O_3$ could have lead to significant levels of $ClNO_2$ to build up and transported to TRF before being completely photolyzed. During the campaign, influence of large biomass burning was negligible as reported in Tang et al. (2018, 2019)."**

(32) Line 260-262: Wang et al. 2016 showed much larger contribution of ClNO2 to ozone increase compared to 2% at TRF site, the latter is similar to 3% at Wangdu in Tham et al. (2016).

Line 260 is referring to OH not $O_3$

(33) Line 270-272: the comment on the result of Tham et al. (2016) and Wang et al. (2016) is an incorrect interpretation of their findings. The three factors (downward transport, horizontal transport and local chemical production) may impact different locations differently. Tham et al. presented evidence for downwind transport in the early morning hours at a polluted rural site in the North China plain. Wang et al. (2016) measured the high ClNO2 plumes at a 974 m mountain- top site and suggested presence of high ClNO2 in the upper boundary layer in south China. They did not say that downward transport should apply to all locations. One has to analyze his/her own case.

We agree on the comment. The following has been added as suggested:

(Ln 327-332) **"Previous studies have attributed high sustained $ClNO_2$ in the morning to transport from the residual layer (Tham et al., 2016; Wang et al., 2016). In this study, box model runs of heterogeneous and gas-phase production of $ClNO_2$ could not reconcile the observed levels. Moreover, airborne observations in the early morning showed negligible $ClNO_2$ levels in the residual layer in most of the days. Therefore, local transport of $ClNO_2$ from highly polluted airmasses from the west is the most plausible explanation. This shows that different meteorological or chemical conditions of the sites can lead to various causes of high $ClNO_2$ levels in the early morning."**

(34) Figure 7. What was this figure used for?

The reference to figure 9 has been corrected to 7 in the text.

**References** (*only the references in our responses have been added)

Atkinson, R., Baulch, D. L., Cox, R. A., Crowley, J. N., Hampson, R. F., Hynes, R. G., Jenkin, M. E., Rossi, M. J., and Troe, J.: Evaluated kinetic and photochemical data for atmospheric chemistry: Volume I - gas phase reactions of $O_x$, $HO_x$, $NO_x$, and $SO_x$, species, Atmos. Chem. Phy., 4, 1461–1738, https://doi.org/10.5194/acp-4-1461-2004, 2004.

Atkinson, R., Baulch, D. L., Cox, R. A., Crowley, J. N., Hampson, R. F., Hynes, R. G., Jenkin, M. E., Rossi, M. J., and Troe, J.: Evaluated kinetic and photochemical data for atmospheric chemistry: Volume III - Gas phase reactions of inorganic halogens, Atmos. Chem. Phy., 7, 981–1191, https://doi.org/10.5194/acp-7-981-2007, 2007.

Baasandorj, M., Hoch, S. W., Bares, R., Lin, J. C., Brown, S. S., Millet, D. B., Martin, R., Kelly, K., Zarzana, K. J., Whiteman, C. D., Dube, W. P., Tonnesen, G., Jaramillo, I. C., and Sohl, J.: Coupling between Chemical and Meteorological Processes under Persistent Cold-Air Pool Conditions: Evolution of Wintertime PM2:5 Pollution Events and $N_2O_5$ Observations in Utah's Salt Lake Valley, Environ. Sci. Technol., 51, 5941–5950, https://doi.org/10.1021/acs.est.6b06603, 2017.

Bertram, T. H. and Thornton, J. A.: Toward a general parameterization of $N_2O_5$ reactivity on aqueous particles: the competing effects of particle liquid water, nitrate and chloride, Atmos. Chem. Phy., 9, 8351–8363, https://doi.org/10.5194/acp-9-8351-2009, 2009.

Brown, S. S., Dubé, W. P., Tham, Y. J., Zha, Q., Xue, L., Poon, S., Wang, Z., Blake, D. R., Tsui, W., Parrish, D. D., and Wang, T.: Nighttime chemistry at a high altitude site above Hong Kong, 
[revised manuscript text omitted]

---

## Referee Report (RR1)

**R1 - Review of "Integration of Airborne and Ground Observations of Nitryl Chloride in the Seoul Metropolitan Area and the Implications on Regional Oxidation Capacity During KORUS-AQ"**

The authors have provided a thoughtful response to the original review. While the manuscript has been greatly improved, there are a few remaining issues that have not been fully addressed in the updates. The primary concerns are (1) the description of each type of box model simulation and (2) the role of vertical vs. horizontal mixing as a primary source of surface-level $ClNO_2$. In the comments below, the original review comments are in gray, the author response is in blue, and the new comments are in red. A few minor comments are at the end in black.

**Comments on Author Response**

Comment (1) - The authors mention that stagnation events were associated with low ClNO2 production. There was no discussion in the text, however, about the meteorology associated with these low ClNO2 events. Moreover, past studies have shown that certain types of stagnation events can actually enhance N2O5 chemistry (e.g. Baasandorj et al., 2017).

Description on the meteorology during the campaign have been included in the Methods section for clarity:
(Ln 110-115) **"Meteorology during the observation period can be classified into dynamic (May 4th - 16th), stagnation (May 17th - 22nd), transport (May 25th - 31st), and blocking period as shown in Figure 3. During the stagnant period, high pressure system was persistent in the Korean peninsula resulting in local air masses to be more dominant within the SMA compared to the dynamic and transport (May 25nd - 31st) periods. Rex block patterns were observed during the blocking period (June 1st - 6th), which also resulted in more local influence."**

The following have been included in the discussion for further discussion on stagnation events and $N_2O_5$:
(Ln 236 -241) **"Stagnation events can be characterized by low wind speeds and increased atmospheric stability, possibly leading to enhanced levels of pollutants like $NO_x$. Previous studies have shown that these stagnant conditions can result in enhanced levels of $N_2 O_5$ (Baasandorj et al., 2017) driven by high ozone and $NO_2$. However, $ClNO_2$ production was limited during stagnation events in this study. This is likely due to limited availability of chloride as shown in submicron particle measurements of aerosol mass spectrometer (AMS) at the ground site for OP and airborne over TRF (Figure 3)."**

Lines 112 – 114 – The authors have improved the manuscript with a description of the meteorological events observed during the study. However, neither Figure 2 nor 3 appear to explicitly show the meteorological conditions as described in this section. The only indicator seems to be the source contribution from the ocean. Please clarify how the different meteorological conditions are represented in these time series. In addition, please define or explain a 'Rex block pattern'.

Line 232 – Same comment here. It is not clear how Figure 3 represents the different meteorological conditions experienced at the sites. Please update the figure or add clarification in the text.

Line 238 – Please specify that the stagnation events reported in Baasandorj et al., 2017 are during the winter season.

Comment (2) - More details are required in this section about the model set-up and the types of simulations that were conducted.

Ln 135 –State which meteorological parameters were used as constraints. Also clarify the type of simulations that were run and how the model was constrained with observations. For example, were simulations run for 24, 48, 76 hours? Were simulations constrained every 10 minutes, 1 hours, 6 hours, etc.? Were different simulations run and constrained to observations from both of the ground sites or was a single simulation run with a combination of the two?

The 2.3 Modeling section was revised as suggested:
(Ln 155-156) **"Each step of the model was constrained with the averaged meteorology parameters (e.g., pressure, temperature, relative humidity) and trace gases observed…"**

Additional information of the model setup has been included in the supplementary:
(Supplementary Ln 31-34) **"Daytime steady state $Cl_2$ simulations (Figure 5) were constrained with meteorology and trace gas observations corresponding to each point of the data shown in Figure 4. The constrained parameters were kept constant throughout the 72 hours of integration time and the end points are shown in Figure 5. Steady state simulations of $ClNO_2$ production in the morning (Figure 8) were ran similarly by holding constant of all the measured parameters throughout the runs, except for $ClNO_2$. "**

(Supplementary Ln 49-55) **"Impact of measured $ClNO_2$ on $O_3$ production (Figure 10) was explored by constraining the box model with diurnal variation of observations throughout each step. Constraining the model with the diurnal variation of measured $ClNO_2$, allowed the box model to capture its trend throughout the course of the day. Since our purpose of the simulations were to explore the possible impact of $ClNO_2$ on $O_3$ production, $NO_2$ and $O_3$ were only constrained initially at the first step with observations and then calculated based on the chemistry embedded in the model. More specifically, the initial concentration of each following step was taken from the value in the previous step. The integration time in the model was 100 sec and the model was run for 1 day."**

Section 2.3 – While the authors have provided many details in their update, it remains unclear exactly how the simulations were setup, how many different types of simulations were run (and what the similarities and differences were), as well as important missing details about the model treatment of boundary layer dynamics and deposition.

First, the authors need to state explicitly in the Section 2.3 of the main text what the duration of each model simulation was (i.e., 72 hours?) and the frequency at which the model was constrained with observations (i.e. every 100 minutes?). Currently, the supplement indicates a discrepancy where it states that the model duration is 1 day. However, it becomes apparent later in the main text that (at least) two different types of simulations were conducted (both 72 hour and 1 day simulations). The setup details (e.g., duration, initialization time, constraint frequency, any chemical/physical differences, number of simulations, etc. ) are needed for each type of simulation in order to avoid confusion later. For instance, on Line 219, it is unclear what the 'end points' of 72-hour simulations are. Are these final mixing ratios at the end of different simulations? What time of day are the simulations set to end? How many individual simulations were run and what were the differences between them? At the moment, Section 2.3 reads as if there was only one type of simulation used.

Second, the authors need to include additional information about how boundary layer dynamics are treated in both the 72 and 24-hour simulations. For instance, how does the model treat the separation of the surface and residual layer at night and the entrainment of $O_3$ in the morning? Does the model include deposition? These details are missing from both the main text and the supplement and are important for the accurate simulation of $O_3$ production.

Third, on line 34 in the supplement, it is unclear what the authors mean by 'all the measured parameters'. Does this mean that all concentrations are held constant? All rate constants? All photolysis frequencies? Please clarify.

Comment (3) - Ln 211 – There have been multiple studies that have discussed the change in N2O5 chemistry with altitude. Many of these have been in the context of nitrate aerosol production. It would be good to reference some of this past work (in addition to Brown et al. (2017)) when discussing the change in ClNO2 production with altitude. For example: (Baasandorj et al., 2017; Tham et al., 2016; Young et al., 2012; Yun, Hui et al., 2018)

The references have been added and discussed as below:
(Ln 254-259) **"However, significant levels of N$_2$O$_5$ and ClNO$_2$ could have been present in the upper part of the surface layer as shown in previous studies (Baasandorj et al., 2017; Young et al., 2012; Yun et al., 2018). According to Baasandorj et al. (2017), significant levels of N$_2$O$_5$ were observed aloft, within the boundary layer, while O$_3$ was completely titrated near the surface. On the other hand, airborne measurements at the LA basin (Young et al., 2012) showed a relatively uniform ClNO$_2$ profile throughout the boundary layer as O$_3$ did not change significantly within the measured altitude."**

Paragraph on line 246 –In this section, instead of referring to the boundary layer at night, please clarify that the boundary layer splits into a nocturnal surface layer and a residual layer. This distinction will help clarify that the authors are discussing $N_2O_5$ and $ClNO_2$ profiles at night in this section. In addition, since this is the first discussion of elevated $N_2O_5$/$ClNO_2$ aloft, it seems to make more sense to move lines 278-281 to line 255.

Line 256 – Suggest changing this sentence to, "According to Baasandorj et al. (2017), $O_3$ was completely titrated at the surface in Salt Lake Valley, Utah, while elevated mixing ratios of $N_2O_5$ were observed at 155m AGL, at a site along the valley wall." These changes are meant to clarify that Baasandorj study reported ground measurements only, whereas the current text indicates that aircraft or tower measurements were collected.

Line 258 – Specify the altitude range of data reported by Young et al. (2012).

Comment (4) - Paragraph starting on line 243 –Is there further information in the flight data to support the hypothesis that boundary layer transport is the main source of elevated surface-level ClNO2? For example, were NOx and O3 observations to the west of the observation sites elevated relative to the east? My concern is that this section reads as though the third proposed possibility must be correct since there was no evidence for the first two possibilities. As written, there is not enough evidence in this section to support the third possibility that transport is the main source of surface-level ClNO2.

Following has been included in the discussion
(Ln 299-305) **"During KORUS, the DC-8 did not fly to the west of the SMA in the early morning. However, there are large point sources, such as petrochemical facilities and industries, and vehicular emissions to the west and south west of the SMA region. Sullivan et al. (2019) reported that this resulted in enhanced levels of $O_3$ in receptor regions (i.e., Taehwa Research Forest) downwind when westerlies were prevalent. Therefore, favorable conditions such as high chloride content in aerosols from both anthropogenic and natural sources and high levels of $NO_x$- $O_3$ could have lead to significant levels of $ClNO_2$ to build up and transported to TRF before being completely photolyzed. During the campaign, influence of large biomass burning was negligible as reported in Tang et al. (2018, 2019)."**

Lines 277 & 294 – Please clarify, when the authors state, 'transport', do they mean *only* horizontal transport or *both* vertical and horizontal transport? It seems that the authors are arguing that high $ClNO_2$ levels are being transported horizontally to the ground sites from other (non-measured) sources to the west of the sites. It also seems possible, however, that $ClNO_2$ is being produced aloft at night in the west and these concentrations are being transported both horizontally and vertically in the morning as the air parcel moves. Please clarify and discuss any evidence for the role of horizontal vs. vertical transport from the western region.

In light of this possibility, it seems that the role of vertical mixing cannot be completely ruled out, as is currently suggested in the discussion section. I would suggest that the paragraph starting on line 278 clarify that while aircraft measurements collected to the south/east of the sites provided limited evidence of the role of vertical mixing contributions to elevated surface-level $ClNO_2$, there may be enhanced vertical contributions from air parcels arriving from the west.

Comment (5) - Figure 3 – This timeseries makes Cl2 appear as if it has a constant background of ~2 pptv at each site. Could the authors comment on this background and discuss whether it is real or an instrument artifact?

Figure 3 has been corrected to remove data points below detection limit. For $Cl_2$, the detection limit was around 2.9 ppt (2 sigma, over 30 min).

Figure 3 – With the author updates, it is now clear why the $Cl_2$ data appear to have an offset in Figure 3. The below LOD data points, however, appear to be included in later figures, such as Figure 4. Please clarify why below LOD data were removed in Figure 3 and not others.

Comment (6) - Clarify, did the authors apply a hygroscopic growth factor to the measured aerosol surface area? What was the size range of the aerosol particles that contributed to the measured surface area? Did aircraft vertical profiles show that the aerosol surface area was relatively constant with altitude? As there are no measurements of aerosol surface area on the ground, this source of uncertainty in the model should be discussed. In addition, it would be helpful to put the N2O5 uptake coefficient into context. The authors could cite previous studies that derived uptake coefficients in Asia (e.g. Brown et al., 2016; Tham et al., 2016; Wang, Z. et al., 2017; Wang, X. et al., 2017; Wang, H. et al., 2017). As this manuscript is primarily about ClNO2 production, the authors should also state how the ClNO2 yield was calculated in the model. In the event that the Bertram and Thornton parameterization was used, it is also important to note that this has been shown to be an over- prediction of field-derived yields (McDuffie et al., 2018a; Riedel et al., 2013; Ryder et al., 2015; Tham et al., 2018; Thornton et al., 2010; Wagner et al., 2013; Wang, Z. et al., 2017; Wang, X. et al., 2017).

The following has been added in the supplementary for clarity:

(Supplementary Ln 34 - 48) "**Heterogeneous reactions of gas-phase $N_2O_5$ (i.e., $N_2O_{5(g)} + Cl^-$ $(aq) \rightarrow ClNO_{2(g)}$ ) , $ClONO_2$ (i.e., $ClONO_{2(g)} + Cl^-(aq) + H^+(aq) \rightarrow Cl_{2(g)} + HNO_3$) , and HOCl (i.e., $HOCl_{(g)} + Cl^-(aq) + H^+(aq) \rightarrow Cl_{2(g)} + H_2O$ ) were included in the model. For these heterogeneous reactions, a simple first-order reaction was assumed by accounting for $\gamma$ , $\Phi$, molecular speed of the gases, and surface area of aerosols. Hygroscopic growth factor was not considered in the model. $\gamma_{N2O5}$ was calculated from the Bertram and Thornton (2009) study using measured inorganic aerosol composition, temperature, and relative humidity and water content derived from the thermodynamic model Extended Aerosol Inorganics Model (E-AIMS, (Clegg et al., 1998; Friese and Ebel, 2010)). The average and median $\gamma_{N2O5}$ values during the whole campaign were both 0.017. This is in the lower range of what has been derived from previous field observations in Asia that ranges from a campaign average of 0.004 to 0.072 (Yun et al., 2018; Brown et al., 2016; Tham et al., 2016; Wang et al., 2017b, d, a, c). $\gamma$ values of $ClONO_2$ and HOCl were set to 0.06 (Deiber et al., 2004; Hanson et al., 1994; Hanson and Ravishankara, 1994). The yields ($\Phi$) of the three heterogeneous reactions were assumed to be 1, therefore the steady state simulations would**

**be an upper-limit of $Cl_2$ or $ClNO_2$ production. Aerosol surface area was taken from airborne measurements of particle size distributions. An averaged value was used from data retrieved below 1 km over the SMA. The airborne data did not show a significant vertical dependence within the daytime boundary layer. Based on this, an average of $78 \pm 4$ 1 $\mu m^2$ $cm^{-3}$ were estimated for particle sizes between 10 nm and 5 $\mu m$."**

Supp. Line 38 – Please clarify why dry surface area was used to calculate gamma($N_2O_5$). This assumption will artificially increase the $N_2O_5$ rate constant as gamma($N_2O_5$) is typically calculated using the wet aerosol surface area.

Comment (7) - Ln 257 – Please define net O3 production rate and explain how this was calculated from model simulations.

The following has been included in the supplementary for clarity:
(supplementary Ln 55-57) **Net $O_3$ production rate was calculated in the box model as below, where f is the stoichiometric coefficient of $O_3$ and k is the rate constant corresponding to each reaction i. More details can be found in the supplements of Wolfe et al. (2016) :**
*# of reactions*
*$d[O_3]/dt = O_3$ production rate - $O_3$ loss rate = $\sum f i \times$ (product of reactants)$i \times ki$*

Equation 1 – Please check the formatting of this equation in the final manuscript.

**Additional Minor Comments:**

Line 6 – Change to, '…sites, the slope of which were dependent on $O_3$ levels."

Line 29 – The authors have appropriately cited the discrepancies between field and laboratory results. It remains unclear, however, where the 'recommended' value comes from. Could the authors please specify. I only ask because both field and lab measurements have reported a wide range of values from $< 1 \times 10^{-4}$ to $> 0.04$.

Line 62 – Change to, '…reported the highest-recorded mixing ratio of $ClNO_2$ (8.3 ppbv)…'

Paragraph staring on line 65 – The authors might also want to include the recent modeling paper by X. Wang (2019) ACP, who simulated $ClNO_2$ production in GEOS-Chem and found large contributions of this chemistry to daytime radical production.

Line 112 – Figure 3 is referenced before Figure 2.

Line 122 – Change, 'However' to 'In addition'

Line 129 – What is the 'blower' that the authors refer to? This isn't indicated in Figure S1.

Line 149 – There are $NO_y$ species in addition to HONO and $NO_2$ that would need to be subtracted to quantify the ambient $ClNO_2$. These include, but are not limited to $N_2O_5$, $NO_3$, NO, and $HNO_3$. Please clarify whether these species were included in the subtraction or note that they were not and state the resulting bias.

Line 165 – Just to clarify, $ClNO_2$, $Cl_2$, and $ClONO_2$ photolysis frequencies were not included in TUV, but the authors incorporated the measured values into the F0AM model, correct? If so, please adjust the text accordingly.

Line 188 – Please report observations averaged over the same time period (i.e. 10 minutes) so that mixing ratios at different sites can be compared directly. Check for consistency throughout the manuscript when mixing ratio from the aircraft and ground sites are reported and compared to each other.

Line 191 – Different nighttime correlations between $Cl_2$ and $ClNO_2$ may not only indicate different sources, but potentially different loss processes as well.

Line 216 – Change, '1229 runs…' 'to '1229 points…'. Unless the authors did run 1229 simulations that were each 72-hours in duration.

Line 310, 335 and abstract – The authors should specify that the percent increases only account for the chemical production of $O_3$. This is an important point because net $O_3$ production

observed from the ground will also be influenced by the morning entrainment of $O_3$ from aloft (which doesn't seem to be included in the model here).

Line 324 – Change 'on the' to 'of the'

Figure 4 – It is unclear why modeled $Cl_2$ is compared to observed and not modeled $ClNO2$. This comparison could help provide additional validation of the model.

Supplement:
Section S3. – When the authors state that 'average' values were used, does that mean that a single value was used throughout the entire model simulation? Please clarify. Also, in this section, please change Figure references 'Figure 4' and 'Figure 5' to 'Figure S4' and 'Figure S5'.

---

## Referee Report (RR2)

**Review of Jeong et al.,**

Original comments are in *gray italics*, author responses are in *blue italics*, additional comments are in black.

Comment #1

*Supp. Line 38 – Please clarify why dry surface area was used to calculate gamma(N2O5). This assumption will artificially increase the N2O5 rate constant as gamma(N2O5) is typically calculated using the wet aerosol surface area.*

*In our study, dry surface area from observations were not used in deriving the gamma($N_2O_5$) . As described in the paper, it was derived based on the Bertram and Thornton (2009) study where they used an empirical pre-factor in the equation. Other parameters used in deriving gamma are described in the Supporting Information section 3. The estimation by Bertram and Thornton (2009) can be an overestimation as mentioned in the first round of reviewer comments. However, the purpose of the box model simulations was to make sense whether the observed $ClNO_2$ levels were in the range that could be explained by in-situ chemistry. The box model results significantly underestimated the observations (~50 times), therefore leading to a conclusion that the second $ClNO_2$ peak in the morning is likely not from in-situ production.*

I misspoke in my original question, I meant to ask why the authors had used dry aerosol SA in the calculation of the $1^{st}$ order loss rate coefficient of $N_2O_5$, not gamma($N_2O_5$). The authors state on line 39 in the supplement that they do not consider aerosol hygroscopic growth when they calculate the $1^{st}$ order loss rate coefficient for $N_2O_5$ (k = 0.25*c*SA*gamma). Using dry SA will result in a smaller loss rate constant and could lead to an under-prediction in $ClNO_2$ production (as is observed by the model). Please clarify why you did not use hygroscopic growth in the calculation of kN2O5 as is stated on line 39 in the supplemental and what implications this may have on your results.

Comment #2

*Section 2.3 – While the authors have provided many details in their update, it remains unclear exactly how the simulations were setup, how many different types of simulations were run (and what the similarities and differences were), as well as important missing details about the model treatment of boundary layer dynamics and deposition.*

*First, the authors need to state explicitly in the Section 2.3 of the main text what the duration of each model simulation was (i.e., 72 hours?) and the frequency at which the model was constrained with observations (i.e. every 100 minutes?). Currently, the supplement indicates a discrepancy where it states that the model duration is 1 day. However, it becomes apparent later in the main text that (at least) two different types of simulations were conducted (both 72 hour and 1 day simulations). The setup details (e.g., duration, initialization time, constraint frequency, any chemical/physical differences, number of simulations, etc. ) are needed for each type of simulation in order to avoid confusion later. For instance, on Line 219, it is unclear what the 'end points' of 72-hour simulations are. Are these final mixing ratios at the end of different simulations? What time of day are the simulations set to end? How many individual simulations were run and what were the differences between them? At the moment, Section 2.3 reads as if there was only one type of simulation used.*

*Second, the authors need to include additional information about how boundary layer dynamics are treated in both the 72 and 24-hour simulations. For instance, how does the model treat the separation of the surface and residual layer at night and the entrainment of O3 in the morning? Does the model include deposition? These details are missing from both the main text and the supplement and are important for the accurate simulation of O3 production.*

*Third, on line 34 in the supplement, it is unclear what the authors mean by 'all the measured parameters'. Does this mean that all concentrations are held constant? All rate constants? All photolysis frequencies? Please clarify.*

*As stated in the first sentence of the 2.3 modeling section, there were 3 types of simulations carried out each represented in Figure 5, 8, and 10.*

*We revised it as below for clarification:*

**(Ln 158-160) "We used Framework for 0-D Atmospheric Modeling (F0AM v3.1) for simulating three types of simulations: 1) daytime $Cl_2$ production (Figure 5), 2) in-situ $ClNO_2$ production in the morning (Figure 8), and 3) testing the impact of measured $ClNO_2$ on the regional tropospheric chemistry (Figure 10)."**

Thank you, it is now clear that the authors conducted 3 different types of simulations. I will reiterate, however, that it is important to move some of the simulation setup details from the supplement to the main text here. The reader should be able to understand the basic differences between these three simulations without having to refer to the supplement, which is not currently the case.

For instance, the authors added: **(Line 165-167) "For simulations presented in Figures 5 and 8, a constant meteorology and trace gas observation set, collected at the corresponding time point, were constrained throughout the model run. For Figure 10, the model was constrained with a diurnal variation of the parameters."** and

**(Ln 182-183) "More details on the setup of the box model and the main differences between the three types of simulations are in the supplement material (S3)."**

Please add a few more basic simulation details to lines 165-167. For example, change to something like… "For simulations presented in Figures 5, a constant meteorology and trace gas observation set, collected at the corresponding time point, were constrained throughout the 72-hour model simulation. The Cl2 concentrations at the end of the 72-hour simulation are compared to simultaneously observed mixing ratios of $ClNO_2$ in Figure 5. Simulations in Figure 8 were similarly constrained as those in Figure 5 but allow $ClNO_2$ concentrations to vary with time in order to assess $ClNO_2$ production predicted by the model. For Figure 10, the model was constrained with a diurnal variation of the parameters for the [72-hour?] model simulation. To assess the impact of $ClNO_2$ chemistry on net $O_3$ production, all species were constrained except for $NO_2$ and $O_3$, which were initialized with observed values and allowed to vary in time"

These are just suggestions, but information like this would give the reader enough to be able to understand how and why the different simulations were conducted. The reader can then go to the supplement if they want more information.

*The duration of each of the model simulations are in the supplementary. The supplementary does not indicate that the model duration was 1 day.*

I'm still confused about the duration of the 3$^{rd}$ set of simulations (shown in Figure 10). The authors state on line 55 in the supplemental: "The integration time in the model was 100 sec and the model was run for 1 day." Please clarify.

**Additional Minor Comments:**

Line 71 – The authors use CMAQ here but don't define it until later on line 80.

Line 89 – Remove '…due to prolonged nighttime'.

Line 255 – remove the word 'these'. This makes it sound as though Bassandorj (2018) was studying the same stagnation events in Korea, which is not the case.

Line 256 – It looks like there is an extra part of a sentence included here. Remove the text between 'NO$_2$.' on line 255 and 'However' on line 257.

Line 364 – The authors clearly show in Section 3.1 that removing HCl production reduces Cl$_2$ production. However, in order to conclude here that ClONO2 and HOCl uptake are responsible for the positive Cl2/ClNO$_2$ correlations and dependency of this correlation on O3, the authors should show how changes in uptake, not HCl, impact the correlation. While it may be the case that ClONO2 and HOCl uptake are responsible for this the result, I suggest either a sensitivity test with the ClONO2 and HOCl uptake coefficients set to 0 or changing this concluding sentence to state that HCl production from VOCs may be responsible for the observed trends (which is the test that was actually conducted).

Line 372 – Change 'it is clear' to 'back-trajectories suggest that air masses were mostly transported from the west …".

---

## Editor Decision (ED1)

**Review of Jeong et al.,**

Original comments are in *gray italics*, author responses are in *blue italics*, additional comments are in black.

**Comment #1**

Supp. Line 38 - Please clarify why dry surface area was used to calculate gamma(N2O5). This assumption will artificially increase the N2O5 rate constant as gamma(N2O5) is typically calculated using the wet aerosol surface area.

In our study, dry surface area from observations were not used in deriving the gamma( $N_2O_5$ ). As described in the paper, it was derived based on the Bertram and Thornton (2009) study where they used an empirical pre-factor in the equation. Other parameters used in deriving gamma are described in the Supporting Information section 3. The estimation by Bertram and Thornton (2009) can be an overestimation as mentioned in the first round of reviewer comments. However, the purpose of the box model simulations was to make sense whether the observed  $CINO_2$  levels were in the range that could be explained by in-situ chemistry. The box model results significantly underestimated the observations (~50 times), therefore leading to a conclusion that the second  $CINO_2$  peak in the morning is likely not from insitu production.

I misspoke in my original question, I meant to ask why the authors had used dry aerosol SA in the calculation of the 1st order loss rate coefficient of  $N_2O_5$ , not gamma( $N_2O_5$ ). The authors state on line 39 in the supplement that they do not consider aerosol hygroscopic growth when they calculate the 1st order loss rate coefficient for  $N_2O_5$  (k = 0.25\*c\*SA\*gamma). Using dry SA will result in a smaller loss rate constant and could lead to an under-prediction in CINO2 production (as is observed by the model). Please clarify why you did not use hygroscopic growth in the calculation of kN2O5 as is stated on line 39 in the supplemental and what implications this may have on your results.

**Comment #2**

Section 2.3 – While the authors have provided many details in their update, it remains unclear exactly how the simulations were setup, how many different types of simulations were run (and what the similarities and differences were), as well as important missing details about the model treatment of boundary layer dynamics and deposition.

First, the authors need to state explicitly in the Section 2.3 of the main text what the duration of each model simulation was (i.e., 72 hours?) and the frequency at which the model was constrained with observations (i.e. every 100 minutes?). Currently, the supplement indicates a discrepancy where it states that the model duration is 1 day. However, it becomes apparent later in the main text that (at least) two different types of simulations were conducted (both 72 hour and 1 day simulations). The setup details (e.g., duration, initialization time, constraint frequency, any chemical/physical differences, number of simulations, etc. ) are needed for each type of simulation in order to avoid confusion later. For instance, on Line 219, it is unclear what the 'end points' of 72-hour simulations set to end? How many individual simulations were run and what were the differences between them? At the moment, Section 2.3 reads as if there was only one type of simulation used.

Second, the authors need to include additional information about how boundary layer dynamics are treated in both the 72 and 24-hour simulations. For instance, how does the model treat the separation of the surface and residual layer at night and the entrainment of O3 in the morning? Does the model include deposition? These details are missing from both the main text and the supplement and are important for the accurate simulation of O3 production.

Third, on line 34 in the supplement, it is unclear what the authors mean by 'all the measured parameters'. Does this mean that all concentrations are held constant? All rate constants? All photolysis frequencies? Please clarify.

*As stated in the first sentence of the 2.3 modeling section, there were 3 types of simulations carried out each represented in Figure 5, 8, and 10.*

We revised it as below for clarification:

(Ln 158-160) "We used Framework for 0-D Atmospheric Modeling (FOAM v3.1) for simulating three types of simulations: 1) daytime Cl2 production (Figure 5), 2) in-situ ClNO2 production in the morning (Figure 8), and 3) testing the impact of measured ClNO2 on the regional tropospheric chemistry (Figure 10)."

Thank you, it is now clear that the authors conducted 3 different types of simulations. I will reiterate, however, that it is important to move some of the simulation setup details from the supplement to the main text here. The reader should be able to understand the basic differences between these three simulations without having to refer to the supplement, which is not currently the case.

For instance, the authors added: (Line 165-167) "For simulations presented in Figures 5 and 8, a constant meteorology and trace gas observation set, collected at the corresponding time point, were constrained throughout the model run. For Figure 10, the model was constrained with a diurnal variation of the parameters." and

**(Ln 182-183) "More details on the setup of the box model and the main differences between the three types of simulations are in the supplement material (S3)."**

Please add a few more basic simulation details to lines 165-167. For example, change to something like... "For simulations presented in Figures 5, a constant meteorology and trace gas observation set, collected at the corresponding time point, were constrained throughout the 72-hour model simulation. The Cl2 concentrations at the end of the 72-hour simulation are compared to simultaneously observed mixing ratios of ClNO2 in Figure 5. Simulations in Figure 8 were similarly constrained as those in Figure 5 but allow ClNO2 concentrations to vary with time in order to assess ClNO2 production predicted by the model. For Figure 10, the model was constrained with a diurnal variation of the parameters for the [72hour?] model simulation. To assess the impact of ClNO2 chemistry on net O3 production, all species were constrained except for NO2 and O3, which were initialized with observed values and allowed to vary in time"

These are just suggestions, but information like this would give the reader enough to be able to understand how and why the different simulations were conducted. The reader can then go to the supplement if they want more information.

**The duration of each of the model simulations are in the supplementary. The supplementary does not indicate that the model duration was 1 day.**

I'm still confused about the duration of the 3rd set of simulations (shown in Figure 10). The authors state on line 55 in the supplemental: "The integration time in the model was 100 sec and the model was run for 1 day." Please clarify.

**Additional Minor Comments:**

Line 71 – The authors use CMAQ here but don't define it until later on line 80.

Line 89 – Remove '...due to prolonged nighttime'.

Line 255 – remove the word 'these'. This makes it sound as though Bassandorj (2018) was studying the same stagnation events in Korea, which is not the case.

Line 256 - It looks like there is an extra part of a sentence included here. Remove the text between 'NO2' on line 255 and 'However' on line 257.

Line 364 - The authors clearly show in Section 3.1 that removing HCl production reduces  $Cl_2$  production. However, in order to conclude here that ClONO2 and HOCl uptake are responsible for the positive  $Cl2/ClNO_2$  correlations and dependency of this correlation on O3, the authors should show how changes in uptake, not HCl, impact the correlation. While it may be the case that ClONO2 and HOCl uptake are responsible for this the result, I suggest either a sensitivity test with the ClONO2 and HOCl uptake coefficients set to 0 or changing this concluding sentence to state that HCl production from VOCs may be responsible for the observed trends (which is the test that was actually conducted).

Line 372 – Change 'it is clear' to 'back-trajectories suggest that air masses were mostly transported from the west ...".

---

## Author Response (AR2)

**Responses to review of "Integration of Airborne and Ground Observations of Nitryl Chloride in the Seoul Metropolitan Area and the Implications on Regional Oxidation Capacity During KORUS-AQ"**

We would like to thank the thoughtful comments from the reviewers. We have revised the paper after carefully considering each comment. Reviewer comments are in _black_, authors' responses are in _blue_, parts that have been added to the manuscript are in **_bold blue_**.

**Anonymous Referee #1**

The authors have addressed most of my comments. However, there are two important points requiring author's further consideration and follow up.

First is on CIMS. More information on the sampling line needs to be provided. What is the length? How often was it cleaned? Although ClNO2 has small loss in the sampling line, N2O5 can be converted to ClNO2 (i.e., give positive bias) if the sample line is not washed frequently. Was this potential bias tested during the field study? If not, need to mention this interference to ClNO2. Also please provide information on CIMS's sensitivity and detection limit or give a reference on them.

The information requested is added to the manuscript as below:

> **(Ln 124-127) "The lengths of the inlet lines of the three CIMS systems were 20 - 30 cm. The PTFE inlet line at the TRF site was washed on a weekly basis and the ones at the OP and DC-8 were not washed routinely during the campaign due to difficulties on detaching the inlet. The potential bias of interactions of $Cl_2$ and $ClNO_2$ inside the inlet were not tested but the artifacts have been shown to be negligible in various field conditions (Riedel et al., 2012; Thornton et al., 2010; Liao et al., 2014)."**

> **(Ln 155-156) "The averaged sensitivity of $Cl_2$ was 31.5 ± 11.2 Hz/ppt and $ClNO_2$ was 19.7 ± 1.5 Hz/ppt. The 2 sigma detection limits of $Cl_2$ and $ClNO_2$ were 2.9 and 1.5 ppt, respectively, over 30 min."**

Another concern is on the attribution of second morning peak. I am still not convinced that it is caused by horizontal transport.

From the chemical point of view, the ClNO2 is formed after sunset, and such formation process continues throughout the night until sunrise. Starting from sunrise, the production of N2O5 and then ClNO2 should be negligible (due to the short lifetime of NO3 after sunrise), but the loss of ClNO2 from photolysis starts to govern the ClNO2 levels which results in decreasing ClNO2 in a certain bulk of air mass.

From the transport point of view, ClNO2 can be produced in the west coast of Korea during the night, and the westerly wind brings ClNO2 to the middle of Korea, and such transport should happen continuously. If the wind doesn't change significantly during the second peak (as in the present study), the same bulk of air mass from the west coast should influence the observation site continuously. And according to the chemical budget of ClNO2 (in the previous point), the ClNO2 level in this bulk of air mass should decrease after the sunrise, and no second peak should be observed.

However, the second peaks were always observed between 7-8 a.m., not before nor after. Such phenomenon is most likely linked to the break-up of the nocturnal boundary layer and the entrainment process after the sunrise. In fact, there is evidence for this downward transport. In figure 6(b), for the second morning peak, ClNO2 increases concurrently with ozone and SO2, which can be interpreted as transport of ozone and SO2 (from elevated point source) from the nocturnal layer. The aircraft measurements at 8:00 did not record elevated ClNO2, maybe because the aircraft did not sample the ClNO2 rich airmass at right location/time.

Therefore, I advise the author to carefully re-consider the explanation of the second morning peak.

We agree on the comments by referee #1 on the possibility of vertical transport of $ClNO_2$ rich air from the residual layer from the west part of the ground sites. We have revised the manuscript and the conclusion of our study accordingly.

Manuscript revised as below:

(Ln 11-12) "...these high levels of $ClNO_2$ in the morning are likely from vertical or horizontal transport of air masses from the west."

(Ln 299-300) "Therefore, horizontal or vertical transport from local sources would be the most likely explanation for the high $ClNO_2$ in the morning."

(Ln 320-325) "However, it is possible that the air mass that was measured by the DC-8 was not representative of the air mass aloft at the west side of the ground observation sites. Backtrajectory analysis initialized at 9:00 local time showed that the TRF site was affected by both the residual layer and below (Figure S10). The enhancement of $O_3$ and $SO_2$ concurrent to elevation of $ClNO_2$ could be due to the transport from the residual layer where pollution from high point sources from the other day was trapped within. From the current dataset, it would be difficult to derive a clear conclusion on whether the cause of the significant $ClNO_2$ in the morning was dominantly of transport from horizontal, vertical, or both."

(Ln 355-357) "However, there is still a possibility of the contribution of vertical transport from the residual layer. Although the current data set is limited for us to

**pinpoint on the vertical locations (i.e., boundary layer v.s. residual layer), it is clear that the CINO$_2$ rich air masses were mostly transported from the west coast where there are significant sources of precursors."**

Additional analysis has been added in the supporting as below:

[Figure]

**Figure S10.** FLEXPART backtrajectories of the selected days when a second CINO$_2$ peak was observed at TRF. Each run was initialized at 9:00 local time and each marker is an hour backward of its previous. The red line represents the center of the mass-weighted particles and the clusters are factional contributions of airmasses in percentage.

Minor points:

1. Response (1) and the corresponding changes in the manuscript, "during a severe have event" should be "during a severe haze event".
Changed

2. Response (20) and the corresponding changes in manuscript, "ClONO2 and HOCl were set to 0.06". Do you mean "the uptake coefficients of ClONO2 and HOCl were set to 0.06"?
Yes, the corresponding part has been written as below:
"...$\gamma_{ClONO2}$ and $\gamma_{HOCl}$ were set to 0.06"

**Anonymous Referee #2**

The authors have provided a thoughtful response to the original review. While the manuscript has been greatly improved, there are a few remaining issues that have not been fully addressed in the updates. The primary concerns are (1) the description of each type of box model simulation and (2) the role of vertical vs. horizontal mixing as a primary source of surface-level ClNO2.

Lines 112 – 114 – The authors have improved the manuscript with a description of the meteorological events observed during the study. However, neither Figure 2 nor 3 appear to explicitly show the meteorological conditions as described in this section. The only indicator seems to be the source contribution from the ocean. Please clarify how the different meteorological conditions are represented in these time series. In addition, please define or explain a 'Rex block pattern'.
Line 232 – Same comment here. It is not clear how Figure 3 represents the different meteorological conditions experienced at the sites. Please update the figure or add clarification in the text.

We were referring to the arrows at the bottom of Figures 3a and 3b for showing the time frames of when the different meteorological conditions governed during the observation period. Reference to 'Figure 3' has been removed from the sentence for less confusion and the caption for Figure 3 has been revised as below:

**(Figure 3 caption) "...The time frames for each meteorological condition that dominated during the observation period are classified in black arrows at the bottom of the Figures 3a and 3b."**

An additional explanation on the 'Rex block pattern' has been added as below:
**(Ln 116-117) "Rex block patterns were observed during the blocking period (June 1st - 6th). During this period, a high pressure system was adjacent to a low pressure over the Korean peninsula resulting in more local influence with occasional stagnation."**

Line 232 only mentions about source contribution analysis and not about different meteorological conditions. Therefore, no changes have been made for Line 232.

Line 238 – Please specify that the stagnation events reported in Baasandorj et al., 2017 are during the winter season.

The related sentence has been re-written as below:
> **(Ln 250-252) "Previous studies have shown that these stagnant conditions can result in enhanced levels of $N_2O_5$ driven by high ozone and $NO_2$, as shown in Baasandorj et al. (2017) during a wintertime observation in the Salt Lake Valley, Utah."**

Section 2.3 – While the authors have provided many details in their update, it remains unclear exactly how the simulations were setup, how many different types of simulations were run (and what the similarities and differences were), as well as important missing details about the model treatment of boundary layer dynamics and deposition.

First, the authors need to state explicitly in the Section 2.3 of the main text what the duration of each model simulation was (i.e., 72 hours?) and the frequency at which the model was constrained with observations (i.e. every 100 minutes?). Currently, the supplement indicates a discrepancy where it states that the model duration is 1 day. However, it becomes apparent later in the main text that (at least) two different types of simulations were conducted (both 72 hour and 1 day simulations). The setup details (e.g., duration, initialization time, constraint frequency, any chemical/physical differences, number of simulations, etc. ) are needed for each type of simulation in order to avoid confusion later. For instance, on Line 219, it is unclear what the 'end points' of 72-hour simulations are. Are these final mixing ratios at the end of different simulations? What time of day are the simulations set to end? How many individual simulations were run and what were the differences between them? At the moment, Section 2.3 reads as if there was only one type of simulation used.

Second, the authors need to include additional information about how boundary layer dynamics are treated in both the 72 and 24-hour simulations. For instance, how does the model treat the separation of the surface and residual layer at night and the entrainment of O3 in the morning? Does the model include deposition? These details are missing from both the main text and the supplement and are important for the accurate simulation of O3 production.

Third, on line 34 in the supplement, it is unclear what the authors mean by 'all the measured parameters'. Does this mean that all concentrations are held constant? All rate constants? All photolysis frequencies? Please clarify.

As stated in the first sentence of the 2.3 modeling section, there were 3 types of simulations carried out each represented in Figure 5, 8, and 10.

We revised it as below for clarification:
> **(Ln 158-160) "We used Framework for 0-D Atmospheric Modeling (F0AM v3.1) for simulating three types of simulations: 1) daytime $Cl_2$ production (Figure 5), 2) in-situ**

CINO$_2$ production in the morning (Figure 8), and 3) testing the impact of measured CINO$_2$ on the regional tropospheric chemistry (Figure 10)."

(Ln 182-183) "More details on the setup of the box model and the main differences between the three types of simulations are in the supplement material (S3)."

The duration of each of the model simulations are in the supplementary. The supplementary does not indicate that the model duration was 1 day. As stated in the supplementary, simulations for Figure 5 and Figure 8 are not runs simulating (or constrained with) a diurnal variation. Each step was constrained with a constant meteorology (e.g, j value) and observation at that time point to simulate Cl$_2$ or CINO$_2$ at the corresponding time. These information can be found in the supplementary. For clarification, the sentences has been re-written as below:

(Supporting Information, Line 31-34) "The meteorology and the constrained trace gas observations corresponding to each time point were kept constant throughout the 72 hours of integration time and the final points of each of these 72 hour runs are shown in Figure 5."

(Line 165-167) "For simulations presented in Figures 5 and 8, a constant meteorology and trace gas observation set, collected at the corresponding time point, were constrained throughout the model run. For Figure 10, the model was constrained with a diurnal variation of the parameters."

(Line 181-182) "Boundary layer height, emissions, and depositions were not considered in the model."

(Line 335-336) "Since the box model simulations in our study did not take into consideration boundary layer height dynamics, emission, and deposition, this net production rate is the result of just chemical production and loss."

Paragraph on line 246 –In this section, instead of referring to the boundary layer at night, please clarify that the boundary layer splits into a nocturnal surface layer and a residual layer. This distinction will help clarify that the authors are discussing N2O5 and CINO2 profiles at night in this section. In addition, since this is the first discussion of elevated N2O5/CINO2 aloft, it seems to make more sense to move lines 278-281 to line 255.

The paragraph on line 246 wasn't necessarily discussing about the residual layer but rather the possibility of higher CINO$_2$ levels being present in the upper part of the nocturnal surface layer. On the other hand, the paragraph starting from lines 278-281 discusses specifically on CINO$_2$ in the residual layer in the early morning. Therefore, to the best of our judgement, we would like to keep the sentences as they are.

Line 256 – Suggest changing this sentence to, "According to Baasandorj et al. (2017), O3 was completely titrated at the surface in Salt Lake Valley, Utah, while elevated mixing ratios of N2O5 were observed at 155m AGL, at a site along the valley wall." These changes are meant to clarify that Baasandorj study reported ground measurements only, whereas the current text indicates that aircraft or tower measurements were collected.

The sentence has been re-written as suggested.

Line 258 – Specify the altitude range of data reported by Young et al. (2012).

The information on the altitude has been included as suggested:

**(Line 271-273) "...showed a relatively uniform $ClNO_2$ profile throughout the nocturnal boundary layer as $O_3$ did not change significantly within the observed altitude range ( < 600 m)."**

Lines 277 & 294 – Please clarify, when the authors state, 'transport', do they mean only horizontal transport or both vertical and horizontal transport? It seems that the authors are arguing that high ClNO2 levels are being transported horizontally to the ground sites from other (non-measured) sources to the west of the sites. It also seems possible, however, that ClNO2 is being produced aloft at night in the west and these concentrations are being transported both horizontally and vertically in the morning as the air parcel moves. Please clarify and discuss any evidence for the role of horizontal vs. vertical transport from the western region. In light of this possibility, it seems that the role of vertical mixing cannot be completely ruled out, as is currently suggested in the discussion section. I would suggest that the paragraph starting on line 278 clarify that while aircraft measurements collected to the south/east of the sites provided limited evidence of the role of vertical mixing contributions to elevated surface-level ClNO2, there may be enhanced vertical contributions from air parcels arriving from the west

We agree to the referee #2 comment. We have revised the manuscript and included additional FLEXPART analysis in the supporting information as below:

Manuscript revised as below:

**(Ln 11-12) "...these high levels of $ClNO_2$ in the morning are likely from vertical or horizontal transport of air masses from the west."**

**(Ln 299-300) "Therefore, horizontal or vertical transport from local sources would be the most likely explanation for the high $ClNO_2$ in the morning."**

**(Ln 320-325) "However, it is possible that the air mass that was measured by the DC-8 was not representative of the air mass aloft at the west side of the ground observation sites. Backtrajectory analysis initialized at 9:00 local time showed that the TRF site was affected by both the residual layer and below (Figure S10). The enhancement of $O_3$ and $SO_2$ concurrent to elevation of $ClNO_2$ could be due to the**

transport from the residual layer where pollution from high point sources from the other day was trapped within. From the current dataset, it would be difficult to derive a clear conclusion on whether the cause of the significant $ClNO_2$ in the morning was dominantly of transport from horizontal, vertical, or both."

(Ln 355-357) "However, there is still a possibility of the contribution of vertical transport from the residual layer. Although the current data set is limited for us to pinpoint on the vertical locations (i.e., boundary layer v.s. residual layer), it is clear that the $ClNO_2$ rich air masses were mostly transported from the west coast where there are significant sources of precursors."

[Figure]

**Figure S10.** FLEXPART backtrajectories of the selected days when a second ClNO$_2$ peak was observed at TRF. Each run was initialized at 9:00 local time and each marker is an hour backward of its previous. The red line represents the center of the mass-weighted particles and the clusters are factional contributions of airmasses in percentage.

Figure 3 – With the author updates, it is now clear why the Cl2 data appear to have an offset in Figure 3. The below LOD data points, however, appear to be included in later figures, such as Figure 4. Please clarify why below LOD data were removed in Figure 3 and not others.

We removed data below the detection limit in Figure S7. In Figure 4, we included the Cl$_2$ data that were below the detection limit to allow comparison to ClNO$_2$ levels when there was limited O$_3$ and Cl$_2$ but significant ClNO$_2$. The caption of Figure 4 has been revised for clarification:

[Figure]

**Figure 4.** Scatter plot of daytime (11:00 - 18:00 local time) ClNO$_2$ and Cl$_2$ at (a) OP and (b)TRF, color coded with measured O$_3$. 5 min averaged data for the whole campaign were used for both sites. Data points of Cl$_2$ below detection limit (2.9 ppt, 2$\sigma$, over 30 min) are shown for the purpose of comparison to observed ClNO$_2$ levels.

Supp. Line 38 – Please clarify why dry surface area was used to calculate gamma(N2O5). This assumption will artificially increase the N2O5 rate constant as gamma(N2O5) is typically calculated using the wet aerosol surface area.

In our study, dry surface area from observations were not used in deriving the gamma(N$_2$O$_5$). As described in the paper, it was derived based on the Bertram and Thornton (2009) study where they used an empirical pre-factor in the equation. Other parameters used in deriving gamma are described in the Supporting Information section 3. The estimation by Bertram and Thornton (2009) can be an overestimation as mentioned in the first round of reviewer comments. However, the purpose of the box model simulations was to make sense whether the observed ClNO$_2$ levels were in the range that could be explained by in-situ chemistry. The box model results significantly underestimated the observations (~50 times), therefore leading to a conclusion that the second ClNO$_2$ peak in the morning is likely not from in-situ production.

Equation 1 – Please check the formatting of this equation in the final manuscript.
Will check the formatting.

Additional Minor Comments:
Line 6 – Change to, '…sites, the slope of which were dependent on O3 levels."
Changed

Line 29 – The authors have appropriately cited the discrepancies between field and laboratory results. It remains unclear, however, where the 'recommended' value comes from. Could the authors please specify. I only ask because both field and lab measurements have reported a wide range of values from < 1x10-4 to > 0.04.
The sentence regarding the 'recommended' value has been removed.

Line 62 – Change to, '…reported the highest-recorded mixing ratio of ClNO2 (8.3 ppbv)…'
Paragraph staring on line 65 – The authors might also want to include the recent modeling paper by X. Wang (2019) ACP, who simulated ClNO2 production in GEOS-Chem and found large contributions of this chemistry to daytime radical production.

Line 62 has been changed accordingly and Wang et al., (2019) has been added as below:

**(Line 84-87) "A recent study by Wang et al. (2019) updated the standard version of the GEOS-Chem (Chen et al., 2017; Sherwen et al., 2016) to better track partitioning between aerosol chloride and gas-phase chlorine species. Comparison between their model simulations with and without ClNO$_2$ production showed enhanced O$_3$ up to 8 ppb during the winter season in Europe due to prolonged nighttime."**

Line 112 – Figure 3 is referenced before Figure 2.
Reference to 'Figure 3' has been removed.

Line 122 – Change, 'However' to 'In addition'
Changed

Line 129 – What is the 'blower' that the authors refer to? This isn't indicated in Figure S1.

The information of the blower has been added as below:

**(Line 132-134) "A total of ~ 3000 standard liters per minute (slpm) was drawn in with a blower with an additional flow of 4 slpm drawn at the end of the inlet to reduce the residence time and 1 slpm was sampled into the CIMS."**

Line 149 – There are NOy species in addition to HONO and NO2 that would need to be subtracted to quantify the ambient ClNO2. These include, but are not limited to N2O5, NO3, NO, and HNO3. Please clarify whether these species were included in the subtraction or note that they were not and state the resulting bias.

As shown in Thaler et al., (2011), this method has been shown to not produce any other gas-phase NO$_y$ byproducts. Therefore, we did not subtract the compounds mentioned in the comment from the total NO$_y$.

Line 165 – Just to clarify, ClNO2, Cl2, and ClONO2 photolysis frequencies were not included in TUV, but the authors incorporated the measured values into the F0AM model, correct? If so, please adjust the text accordingly.

Re-written as below:

**(Line 174) "...taken directly from the DC-8 measurements to be used in the model runs in this study."**

Line 188 – Please report observations averaged over the same time period (i.e. 10 minutes) so that mixing ratios at different sites can be compared directly. Check for consistency throughout the manuscript when mixing ratio from the aircraft and ground sites are reported and compared to each other.

The $ClNO_2$ and $Cl_2$ observations from OP has been replaced with a 5 min averaged data.

Line 191 – Different nighttime correlations between Cl2 and ClNO2 may not only indicate different sources, but potentially different loss processes as well.

Re-written as below:

**(Line 203-204) "This implies that the sources or potentially loss processes of $Cl_2$ and $ClNO_2$ were not consistent at night."**

Line 216 – Change, '1229 runs…' 'to '1229 points…'. Unless the authors did run 1229 simulations that were each 72-hours in duration.

We ran 1229 simulations, each for a 72 hour duration.

Line 310, 335 and abstract – The authors should specify that the percent increases only account for the chemical production of O3. This is an important point because net O3 production observed from the ground will also be influenced by the morning entrainment of O3 from aloft (which doesn't seem to be included in the model here).

Changes in boundary layer height and entrainment of $O_3$ from the residual layer in the early morning are not considered in the model. The following has been added as below for clarification:

**(Line 12-13) "Box model results show that chlorine radical initiated chemistry can impact the regional photochemistry by elevating net chemical production rates of ozone by ~ 25 % in the morning."**

**(Line 335-336) "Since the box model simulations in our study did not take into consideration boundary layer height dynamics, emission, and deposition, this net production rate is the result of just chemical production and loss."**

Line 324 – Change 'on the' to 'of the'
Changed

Figure 4 – It is unclear why modeled Cl2 is compared to observed and not modeled ClNO2. This comparison could help provide additional validation of the model.

The purpose of the model was to validate whether the current understanding of the heterogeneous reactions can explain the daytime observed positive correlation between $Cl_2$ and $ClNO_2$ and its dependency on $O_3$ rather than reproducing the observed $ClNO_2$ levels from model simulations. The initial chlorine radicals can be produced from $ClNO_2$ photolysis, which were constrained with observations in the model. Other gas-phase precursors of chlorine radicals that are generated in the model include $HCl$, $ClONO_2$, $Cl_2$, and $HOCl$.

Supplement:
Section S3. – When the authors state that 'average' values were used, does that mean that a single value was used throughout the entire model simulation? Please clarify. Also, in this section, please change Figure references 'Figure 4' and 'Figure 5' to 'Figure S4' and 'Figure S5'.

Since we did not have any aerosol size distribution data collected at the ground sites, we used airborne data measured over the sites at altitude below 1 km. Since the aerosol data collected airborne did not have a significant variation within 1 km, we used the average value for the box model simulations. Figures 4 and 5 in section S3 is referring to the figures in the main manuscript not the supporting information.

Revised as below:

**(Supporting Information, Line 46-49) "Since we did not have any aerosol size distribution data collected at the ground sites, aerosol surface area was taken from airborne measurements. An averaged value was used from data retrieved below 1 km over the SMA. The airborne data did not show a significant vertical dependence within the daytime boundary layer."**

**References**

[revised manuscript text omitted]

---

## Author Response (AR3)

*We appreciate the comments by the reviewer. Reviewer comments are in Black, responses to these are in blue, and parts that have been added or revised in the manuscript are in **bold blue**.*

I misspoke in my original question, I meant to ask why the authors had used dry aerosol SA in the calculation of the 1st order loss rate coefficient of N2O5, not gamma(N2O5). The authors state on line 39 in the supplement that they do not consider aerosol hygroscopic growth when they calculate the 1st order loss rate coefficient for N2O5 (k = 0.25*c*SA*gamma). Using dry SA will result in a smaller loss rate constant and could lead to an under-prediction in ClNO2 production (as is observed by the model). Please clarify why you did not use hygroscopic growth in the calculation of kN2O5 as is stated on line 39 in the supplemental and what implications this may have on your results.

The following has been added:

(Ln 301-304) **"Using the dry surface area for the first order loss of $N_2O_5$ on aerosols certainly could result in an underestimation of $ClNO_2$ production in the model. Kim et al. (2017, 2018) observed hygroscopic growth factor of less than 1.5 in the SMA region for particles below 150 nm during the KORUS campaign period. In other words, the discrepancy between observed and modeled $ClNO_2$ of more than 50-fold cannot be reconciled by this underestimation."**

Thank you, it is now clear that the authors conducted 3 different types of simulations. I will reiterate, however, that it is important to move some of the simulation setup details from the supplement to the main text here. The reader should be able to understand the basic differences between these three simulations without having to refer to the supplement, which is not currently the case.
For instance, the authors added: (Line 165-167) "For simulations presented in figures 5 and 8, a constant meteorology and trace gas observation set, collected at the corresponding time point, were constrained throughout the model run. For figure 10, the model was constrained with a diurnal variation of the parameters" and (Ln 182-183) "More details on the setup of the box model and the main differences between the three types of simulations are in the supplement material (S3)

Please add a few more basic simulation details to lines 165-167. For example, change to something like…"For simulations presented in Figure 5, a constant meteorology and trace gas observation set, collected at the corresponding time point, were constrained throughout the 72-hour model simulation. The Cl2 concentrations at the end of the 72-hour simulation are compared to simultaneously observed mixing ratios of ClNO2 in Figure 5. Simulations in Figure 8 were similarly constrained as those in Figure 5 but allow ClNO2 concentrations to vary with time in order to assess ClNO2 production predicted by the model. For Figure 10, the model was constrained with a diurnal variation of the parameters for the [72-hour?] model simulation. To assess the impact of ClNO2 chemistry on net O3 production, all species were constrained except for NO2 and O3, which were initialized with observed values and allowed to vary in time" These are just suggestions, but information like this would give the reader enough to be able to understand how and why the different simulations were conducted. The reader can then go to the supplement if they want more information

The following has been added as suggested:

(Ln 165-173) "**A constant meteorology and trace gas observation set, collected at the corresponding time period, were constrained throughout the 72-hour model simulation presented in Figure 5. Then, the $Cl_2$ concentrations at the end of the 72-hour simulation are compared to simultaneously observed mixing ratios of $ClNO_2$ in Figure 5. Simulations in Figure 8 were similarly constrained as those in Figure 5 but allow $ClNO_2$ concentrations to vary with time in order to assess $ClNO_2$ production predicted by the model. The model simulation presented in Figure 10 was constrained with a diurnal variation of the parameters. A full diurnal cycle of the model was for 24 hours consisting a total of 864 steps and each step was integrated for 100 seconds. Each step of the model was constrained with observations measured at that time of day. To assess the impact of**

**CINO$_2$ chemistry on net O$_3$ production, all species were constrained except for NO$_2$ and O$_3$, which were initialized with observed values and allowed to vary in time."**

I'm still confused about the duration of the 3$^{rd}$ set of simulations (shown in Figure 10). The authors state on line 55 in the supplemental: "The integration time in the model was 100 sec and the model was run for 1 day" Please clarify

The following has been added for clarification:

(Ln 169-171) **"The model simulation presented in Figure 10 was constrained with a diurnal variation of the parameters. A full diurnal cycle of the model was for 24 hours consisting a total of 864 steps and each step was integrated for 100 seconds. Each step of the model was constrained with observations measured at that time of day.**

**Additional Minor Comments:**
Line 71-The authors use CMAQ here but don't define it until later on line 80.

The definition of the 'CMAQ' has been added earlier in line 71.

Line 89 – Remove 'due to prolonged nighttime'

We removed the 'due to prolonged nighttime' in the sentence.

Line 255 – remove the word 'these'. This makes it sound as though Bassandorj (2018) was studying the same stagnation events in Korea, which is not the case.

The word 'these' has been removed.

Line 256 – It looks like there is an extra part of a sentence included here. Remove the text between 'NO2' on line 255 and 'However' on line 257.

It has been removed.

Line 364 – _The authors clearly show in Section 3.1 that removing HCl production reduces Cl2 production. However, in order to conclude here that ClONO2 and HOCl uptake are responsible for the positive Cl2/ClNO2 correlations and dependency of this correlation on O3, the authors should show how changes in uptake, not HCl, impact the correlation. While it may be the case that ClONO2 and HOCl uptake are responsible for this the result, I suggest either a sensitivity test with the ClONO2 and HOCl uptake coefficients set to 0 or changing this concluding sentence to state that HCl production from VOCs may be responsible for the observed trends (which is the test that was actually conducted).

In the box model, the only reactions producing gas-phase Cl$_2$ are heterogeneous reactions of gas phase HOCl and ClONO$_2$ on aerosols. Therefore, when these reactions are switched off, simulated Cl$_2$ will be zero. As shown in Figure S4 in the supporting information, when HCl production from VOC oxidation by Cl was switched off, there were still significant amount (30-55 %) of Cl$_2$ produced in the model. The precursor of gas-phase ClONO$_2$ and HOCl is ClO, which is produced from Cl +O$_3$. As mentioned in the text, Cl can come from multiple sources (e.g., ClNO2, Cl$_2$, HOCl, ClONO$_2$ etc..).

[revised manuscript text omitted]